ecology/biogeochemistry/physiology

fasting, compound specific, temporal, migration

**Author for correspondence:**
Philip M. Riekenberg
e-mail: phrieken@gmail.com

†Present address: Department of Bird Ecology, Bureau Waardenburg, 4101 CK Culemborg, The Netherlands.

# Reconstructing the diet, trophic level and migration pattern of mysticete whales based on baleen isotopic composition

Philip M. Riekenberg[1], Jaime Camalich[1], Elisabeth Svensson[1], Lonneke L. IJsseldijk[2], Sophie M. J. M. Brasseur[3], Rob Witbaard[4], Mardik F. Leopold[3], Elisa Bravo Rebolledo[3,†], Jack J. Middelburg[5], Marcel T. J. van der Meer[1], Jaap S. Sinninghe Damsté[1,5] and Stefan Schouten[1,5]

[1]Department of Marine Microbiology and Biogeochemistry, NIOZ Royal Netherlands Institute for Sea Research, PO Box 59, Den Hoorn 1790AB, The Netherlands
[2]Division of Pathology, Department of Biomolecular Health Sciences, Faculty of Veterinary Medicine, Utrecht University, Yalelaan 1, 3854 CL Utrecht, The Netherlands
[3]Wageningen Marine Research, Wageningen University and Research, PO Box 57, 1780 AB Den Helder, The Netherlands
[4]Department of Estuarine and Delta Systems, NIOZ Royal Netherlands Institute for Sea Research, and Utrecht University, PO Box 140, 4400 AC Yerseke, The Netherlands
[5]Department of Earth Sciences, Faculty of Geosciences, Utrecht University, Princetonlaan 8a, 3584 CB Utrecht, The Netherlands

PMR, 0000-0002-6275-5762; JC, 0000-0002-7104-3476; ES, 0000-0002-5565-3266; LLI, 0000-0001-7288-9118; SMJMB, 0000-0002-9245-6990; RW, 0000-0002-7062-7920; MFL, 0000-0002-4540-9841; EBR, 0000-0002-1234-0251; JJM, 0000-0003-3601-9072; MTJv, 0000-0001-6454-1752; JSS, 0000-0002-8683-1854; SS, 0000-0001-9200-8269

Baleen from mysticete whales is a well-preserved proteinaceous material that can be used to identify migrations and feeding habits for species whose migration pathways are unknown. Analysis of $\delta^{13}C$ and $\delta^{15}N$ values from bulk baleen have been used to infer migration patterns for individuals. However, this approach has fallen short of identifying migrations between regions as it is difficult to determine variations in isotopic shifts without temporal sampling of prey items. Here, we apply analysis of $\delta^{15}N$ values of amino acids to five baleen plates belonging to three species, revealing novel insights on trophic position, metabolic state and migration between regions. Humpback and minke whales had higher reconstructed trophic levels than fin whales (3.7–3.8 versus 3–3.2, respectively) as

expected due to different feeding specialization. Isotopic niche areas between baleen minima and maxima were well separated, indicating regional resource use for individuals during migration that aligned with isotopic gradients in Atlantic Ocean particulate organic matter. Phenylanine $\delta^{15}N$ values confirmed regional separation between the niche areas for two fin whales as migrations occurred and elevated glycine and threonine $\delta^{15}N$ values suggested physiological changes due to fasting. Simultaneous resolution of trophic level and physiological changes allow for identification of regional migrations in mysticetes.

## 1. Introduction

Mysticete whales are a concern for ecosystem-based management after their populations were decimated by whaling [1,2]. Despite their current protected status, many details of the migratory patterns and feeding ecology of individual mysticetes remain uncertain and limited to a broader understanding of population-scale patterns. This is especially true at the level of metapopulations and any additional information on ecological niche separation may help to inform policy makers to protect key habitats [3]. New tools are now used to unravel migrations of mysticete whales, such as retrospective analysis of historical landing data [1,4] and satellite tags to track both migratory paths and feeding strategies [5–9]. However, tags provide only a small observational time window and are difficult to successfully deploy due to the mobility and lifestyle of these animals [10].

Feeding strategy and trophic ecology of mysticetes have primarily been identified through visual confirmation during feeding or stomach content analysis from strandings and historic catch data [11,12]. However, prey in the stomach of a deceased animal represents the animal's last meal and can be biased [13], reflecting (i) only the most recent feedings, (ii) the health status of the animal prior to death (i.e. trophic downgrading due to sickness), and (iii) the undigestible portions of the prey. By contrast, proteinaceous materials that are continually produced across an animal's lifetime (e.g. baleen, earplugs) provide a continuous record of metabolic processes [14,15] and dietary composition across the time period that they have been produced. These materials are thus useful in identifying prey composition and feeding strategies over a long period prior to death [7,16]. As a metabolically inert tissue, the incremental deposition of baleen faithfully records the dietary composition of the animal from when it is deposited until it is either lost or worn away. Baleen is composed almost entirely of keratin derived from metabolites from the bloodstream [17] and captures a continuous, long-term record of the animal's blood protein during keratin synthesis [18]. This is in contrast to erythrocytes or skin tissue that provide a single integrated snapshot of isotopic composition (e.g. approx. one–two weeks or four months, respectively) depending on turnover within the pool of carbon (C) or nitrogen (N) being examined and the size of the animal [19]. Whole lengths of baleen often reflect the dietary conditions from several months to several years depending on their growth rate and sampling distance from the gums.

The isotopic composition of C and N in baleen protein (expressed as $\delta^{13}C$ and $\delta^{15}N$ values, respectively) provides insights into the diet composition and habitat [10,20–22]. Typically, consumers have higher $\delta^{13}C$ and $\delta^{15}N$ values by 0.5–2‰ and 0.5–5‰, respectively, compared with their diet [23,24]. These enrichments, or trophic discrimination factors (TDFs), vary with species, tissue type, metabolism and diet quality and, therefore, require consideration of the available ecological context when assigning a TDF to a species within an ecosystem [25,26]. Dietary estimates reconstructed using bulk isotopic values reflect a mixture of influencing factors and often provide results that are inconclusive or muddled [27]. This is especially true in systems where the isotopic baseline supporting production changes or the trophic level that animals feed at have shifted [28,29]. Both the $\delta^{13}C$ and $\delta^{15}N$ values of resources supporting primary production shift substantially both spatially and temporally depending on the balance of biogeochemical processes affecting available inorganic and organic C and N sources within an ecosystem [30–34]. This occurs, for example, with wide-ranging mysticetes in the north Atlantic Ocean where considerably higher $\delta^{15}N$ values are observed for particulate organic matter (POM) in the higher latitudes (greater than 70°N, 6–10‰) compared with lower mid-Atlantic areas (10°N, −1 to 1‰; [35]). In these same areas, $\delta^{13}C$ values range from −28 to −30‰ and −20 to −24‰, respectively. These shifts in baseline $\delta^{15}N$ values are expected to interfere with estimates of trophic level for North Atlantic mysticetes as they migrate between mid-Atlantic breeding grounds and high-latitude feeding grounds. Correction for this baseline shift would usually require extensive sampling of primary consumers across both regions [23,36].

The issue depicted above may be resolved by the application of compound-specific analysis of $\delta^{15}N$ values from amino acids contained in baleen. Isotopic differences that arise due to metabolic pathway differences between amino acid types can help to address regional shifts caused by the underlying

$\delta^{15}$N baseline by providing simultaneous temporal information on trophic level and baseline $\delta^{15}$N values supporting the consumer [37]. Baseline isotopic values are provided by source amino acids that undergo little change as they are metabolized (e.g. phenylalanine (Phe), methionine, tyrosine and lysine; [38,39]). By contrast, trophic amino acids undergo considerable fractionation as they are metabolized (glutamic acid (Glu), aspartic acid, alanine, isoleucine, leucine, proline and valine [39]) and so-called 'metabolic' amino acids that undergo variable fractionations depending on the animal's physiology or dietary composition (glycine (Gly), threonine (Thr)). Through the utilization of both source and trophic amino acids, a TDF [40] and $\beta$, the difference between trophic and source amino acids in underlying primary producers, trophic level estimates for individuals can be calculated [38]. Trophic level estimates inherently integrate underlying baseline shifts that have occurred during migrations between breeding and feeding grounds. Uncertainties remain about the effects of diet quality and metabolic effects associated with routing of compounds (i.e. higher fractionation associated with poorer assimilation efficiency) or excretion pathways (i.e. excretion of urea versus ammonia), but these are incorporated into the high level of uncertainty assigned to the TDF (e.g. 7.6 ± 1.5‰; [40]) and resulting trophic level estimates. Metabolic amino acids may fractionate differently under fasting conditions as the whales migrate from their high-latitude feeding grounds and fasting or even starvation affects metabolism as feeding becomes limited to incidental encounters [41,42].

In this study, we are among the first to combine bulk stable isotope with analysis of $\delta^{15}$N in amino acids from baleen sourced from stranded or bow-caught fin whales ($n = 3$, *Balaenoptera physalus*), a stranded humpback whale (*Megaptera novaeangliae*) and a minke whale (*Balaenoptera acutorostrata*), all opportunistically sampled in The Netherlands. We applied $\delta^{15}$N values of amino acids to baleen to: (i) resolve trophic levels for these species, (ii) identify changes from regional biogeochemical source $\delta^{15}$N values being used during migrations, and (iii) characterize any potential metabolic effects from fasting and episodic feeding during migration. This work serves as a proof of concept that amino acids from baleen can be used as a continuous ecological indicator in mysticete whales. Application and further development of this novel method to materials derived from marine mammal strandings have the potential to inform ecological knowledge about marine mammals, particularly in mysticete whales. Once established, this method will help to reconstruct life histories and further identify the ecological overlap between species. We also provide evidence of feeding in different regions due to migration between mid-Atlantic breeding and high-latitude feeding areas through application of isotopic niche areas from trophic level-corrected bulk isotope data from baleen.

# 2. Methods

## 2.1. Sample collection

Baleen plates were cut from below the gumline of dead mysticetes that stranded on the Dutch coast or were brought into Dutch harbours caught on a ship's bow in 2012 and 2013. Details about sex and estimates of animal's maturity are located in table 1. Baleen from three fin whales were acquired from the faculty of Veterinary Medicine of Utrecht University, the baleen from the humpback whale (*M. novaeangliae*) was acquired from Naturalis Biodiversity Center, Leiden, and the minke whale (*B. acutorostrata*) baleen was acquired from a private collection.

## 2.2. Sample preparation

One baleen plate per individual whale was air-dried (greater than 2 days), cleaned with bidistilled water, then dichloromethane, and dried at 40°C for approximately 10 h. Powdered keratin was collected using a hand drill (3 mm bit) along the leg (labial side) of the plate at either 0.5 cm or 1 cm intervals (representing a range of approx. two to four weeks of growth between samplings, calculated from this study) depending on the relative size of the plate, from the gingiva along the full length of the main plate. Powdered baleen was collected and stored at −20°C until further processing. Since baleen grows continuously throughout a whale's life the material closest to the gingiva reflects the most recently produced layer with the material farther away from the gums reflecting increasingly older periods of the whale's foraging history. Subsampled material (3–5 mg) for analysis of $\delta^{15}$N of amino acids was selected based on the variation within the bulk $\delta^{15}$N values observed for each individual and was used to target minimum and maximum values observed across the lengths of plate.

**Table 1.** Information on study specimens and baleen.

| species | ID | gender | whale length (m) | baleen length (cm) | observations |
|---|---|---|---|---|---|
| fin whale | 1 | male | 18.5[a] | 28 (fragment) | juvenile bow-caught in June 2012 and entered port of Rotterdam |
| | 2 | female | 12.5[a] | 36.2 | juvenile bow-caught in August 2013 and entered port of Rotterdam |
| | 3 | male | 16.5 | 49 | juvenile stranded in September 2013 at 's Gravenzande |
| humpback whale (*Johanna*[b]) | | female | 10.5 | 26.6 | adult female, stranded alive and later died at the Razende Bol, sandbank between Den Helder and the island of Texel December 2012 |
| minke whale | | female | unknown | 18.6 | |

[a]Described in IJsseldijk *et al.* [43].
[b]Described in Besseling *et al.* [44].

## 2.3. Bulk stable isotope analysis

Approximately 0.5–0.8 mg of dry, homogenized keratin powder was weighed into tin cups in duplicates for determination of carbon ($\delta^{13}C$) and nitrogen ($\delta^{15}N$) isotopic ratios for bulk material, as well as carbon and nitrogen content (%) of bulk biomass. Samples were analysed on a Flash 200 elemental analyser coupled to a Delta V Advantage isotope ratio mass spectrometry (Thermo scientific, Bremen).

Stable isotope ratios are expressed using the δ notation in units per mil:

$$\delta(‰) = \left( \left( \frac{R_{\text{sample}}}{R_{\text{standard}}} \right) - 1 \right) \times 1000, \quad \text{where } R = \frac{^{13}C}{^{12}C} \text{ or } \frac{^{15}N}{^{14}N}, \tag{2.1}$$

and expressed versus Vienna Pee Dee Belemnite (VPDB) for $\delta^{13}C$ and atmospheric $N_2$ (air) for $\delta^{15}N$. A laboratory acetanilide standard with $\delta^{13}C$ and $\delta^{15}N$ values calibrated against NBS-22 and IAEA-N1, respectively, and known %TOC and %TN contents, was used for calibration. Analytical precision for the standards (urea, casein) for $\delta^{13}C$ and $\delta^{15}N$ analyses were 0.18‰ and 0.20‰, respectively.

## 2.4. Amino acid sample preparation

The method is a modified version of the amino acid analysis method by Chikaraishi *et al.* [37] as described in Riekenberg *et al.* [45]. In short, at the Royal Netherlands Institute for Sea Research (NIOZ) samples were hydrolysed, derivatized into N-pivaloyl/isopropyl (NPiP) derivatives and analysed in duplicate with a Trace 1310 gas chromatograph coupled to a Delta V Advantage isotope ratio mass spectrometer (Thermo Scientific, Bremen) via an IsoLink II and Conflo IV. Details about the temperature ramp, programme settings and normalization procedures are provided in Riekenberg *et al.* [45]. We report $\delta^{15}N$ values for 12 amino acids including alanine (Ala), aspartic acid (Asp), Glu, Gly, isoleucine, leucine, lysine, Phe, serine, Thr, tyrosine and valine. The precision for samples and standards was less than 0.5‰ for all amino acids in standards and samples across the 13 sequences that comprise this dataset (electronic supplementary material, table S3).

## 2.5. Estimating growth intervals

To examine the relative rates of change for the oscillations in the $\delta^{15}N$ values for bulk material along the length of the main plate, we fitted a generalized additive model (GAM) for each individual. GAM models were produced using the geom_smooth function in the ggplot2 package with model = 'gam' to apply smoothing parameters selected by data-driven methods using Akaike information criteria to time series in R (v. 4.0.0) with R Studio (v. 1.1.463) [16,46]. The marked oscillations in $\delta^{15}N$ values of

**Table 2.** Trophic position (TP) and trophic level (TL) estimates for each individual. Trophic positions were determined from stomach contents and dietary analysis in Pauly et al. [48]. TL is a unitless number calculated here using glutamic acid (Glu), phenylalanine (Phe) or the weighted average of trophic and source amino acids with $\beta$ and trophic discrimination factors indicated below each estimate. $n$ represents the number of amino acid measurements along the length of baleen for each individual and s.d. indicates the standard deviation propagated for each value.

| individual | $n$ | $TP^a_{SCA}$ | $TL^b_{Glu-Phe}$ | s.d. | $TL^b_{Glu-Phe}$ | s.d. | $TL_{Trophic-Source}$ | s.d. |
|---|---|---|---|---|---|---|---|---|
| fin whale 1 | 10 | 3.4 | 3.2 | 0.2 | 3.6 | 0.2 | 3.0 | 0.3 |
| fin whale 2 | 11 | 3.4 | 3.0 | 0.2 | 3.3 | 0.2 | 3.0 | 0.3 |
| fin whale 3 | 16 | 3.4 | 3.3 | 0.2 | 3.7 | 0.2 | 3.2 | 0.4 |
| humpback | 12 | 3.6 | 3.6 | 0.2 | 4.0 | 0.2 | 3.7 | 0.5 |
| minke | 11 | 3.4 | 3.7 | 0.4 | 4.1 | 0.5 | 3.8 | 0.4 |
| TDF | | | $3.6^c$ | 0.3 | 3.1 | 0.3 | 3.6 | 1.7 |
| $\beta$ | | | 3.6 | 0.5 | 3.6 | 0.5 | 3.0 | 0.9 |

[a]Pauly et al. [48].
[b]Ruiz-Cooley et al. [49].
[c]Equation (2.3) average for TDF.

baleen are assumed to reflect residence times in mid-Atlantic breeding grounds (minima) and high-latitude feeding grounds (maxima) with substantial differences in the underlying $\delta^{15}N$ values for POM in these regions [35]. Oscillations within $\delta^{13}C$ values for individuals were less distinct, having a smaller range than those for $\delta^{15}N$ values often due to closer similarity in prey $\delta^{13}C$ values and are known to be further confounded due to coastal foraging in areas with gradients in $\delta^{13}C$ [16]. Therefore, $\delta^{15}N$ values were used to estimate baleen growth rates for each individual (table 1) by assuming the oscillation between sequential $\delta^{15}N$ value minima (along the baleen record represented migratory annual movements between foraging grounds). Growth estimates were determined as the distance between sequential minimum $\delta^{15}N$ values and this interval was used to estimate a weekly growth rate as in Busquets-Vass et al. [16]. To further clarify the midpoint between minimum and maximum periods for $\delta^{15}N$ values we plotted a linear regression across $\delta^{15}N$ values for each baleen and binned regions of each baleen into minimum (below midpoint) and maximum (above midpoint) values depending on relative position to the conditional mean to allow for further analysis of regional differences for $\delta^{15}N$ values. Using the conditional mean to demarcate periods provided a robust and independent indicator of minimum and maximum regions, especially in baleen with less well-defined oscillations. It is more conservative than using narrowly binned regions selected in an arbitrary manner.

## 2.6. Trophic level calculations

Trophic level (TL) estimated from baleen amino acids is presented using either the individual amino acids Glu and Phe [38] or the weighted averages for both trophic and source amino acids (AAs) as presented in Richards et al. [47] using TDF and $\beta$ values appropriate to the trophic-source AA pairings or weighted averages.

$$\text{Trophic level} = \frac{(\delta^{15}N_{Trophic} - \delta^{15}N_{Source} - \beta)}{(TDF + 1)}, \tag{2.2}$$

where $\delta^{15}N_{Trophic}$ and $\delta^{15}N_{Source}$ are either the $\delta^{15}N$ values for Glu and Phe or the weighted mean values for grouped trophic (alanine, aspartic acid, Glu, isoleucine, leucine and valine) and source (lysine and Phe) amino acids. Values for $\beta$, the ‰ difference between Glu and Phe or the grouped trophic and source amino acids in the underlying phytoplankton, and TDF are presented in table 2 for each of the three TL estimates provided here and are compiled from values found in Bradley et al. [50] and McCarthy et al. [51]. The TDF of 3.6‰ has been calculated as the averaged trophic positions from stomach contents in Pauly et al. [48] by rearranging equation (2.2) as

$$TDF = \frac{(\delta^{15}N_{Trophic} - \delta^{15}N_{Source} - \beta)}{(TL - 1)}. \tag{2.3}$$

We also use the TDF value of 3.1‰ found by the statistical analysis presented in Ruiz-Cooley *et al.* [49] to allow for direct comparison across marine mammal species. Error propagation for each trophic level estimate is presented in table 2 and standard deviations throughout are calculated using the propagate package in R.

Correction for trophic enrichment to establish baseline estimates for $\delta^{15}N$ using phenylalanine was calculated as

$$\delta^{15}N_{Phe-Base} = \delta^{15}N_{Phe} - (0.4 \times (\text{trophic level-1})), \tag{2.4}$$

where $0.4 \pm 0.5‰$ is the small enrichment observed for Phe during metabolism [38] and trophic level calculated for each individual (table 1) following the method presented in Vokhshoori *et al.* [52]. Error propagation indicated a standard deviation of 1‰ for $\delta^{15}N_{Phe-Base}$ values.

Trophic level estimates were further used to estimate baseline $\delta^{13}C$ and $\delta^{15}N$ values for bulk measurements using the equations:

$$\delta^{13}C_{Base} = \delta^{13}C_{Bulk} - [2.3 + (0.5 \times (\text{trophic level-2}))], \tag{2.5}$$

and

$$\delta^{15}N_{Base} = \delta^{15}N_{Bulk} - [2.8 + (2.2 \times (\text{trophic level-2}))], \tag{2.6}$$

where $\delta^{13}C_{Bulk}$ and $\delta^{15}N_{Bulk}$ represent the C and N isotopic composition of bulk material, $2.3 \pm 0.3‰$ and $2.8 \pm 0.2‰$ represent the offset between diet and baleen for carbon and nitrogen [53], $0.5 \pm 0.3‰$ and $2.2 \pm 0.3‰$ represent the offsets for trophic enrichment for carbon and nitrogen for the trophic levels supporting the whale's prey [24,54], and trophic level is the average estimate of $TL_{trophic-source}$ for each individual (table 2). Error propagation indicated a standard deviation for $\delta^{13}C_{Base}$ and $\delta^{15}N_{Base}$ of 0.7 and 1‰, respectively. By applying trophic corrections for each species and the source amino acid Phe, we allow for direct comparison of any $\delta^{13}C$ or $\delta^{15}N$ values against the oceanic isoscape for POM presented in Trueman *et al.* [35]. Wilcoxon signed-rank *t*-tests were used to examine individual amino acid $\delta^{15}N$ values between regions of baleen.

## 2.7. Isotopic niche modelling

To analyse differences in isotopic niches within each individual baleen, standard ellipse areas corrected ($SEA_c$) for their sample size were constructed containing 70% of the variation in each group for the binned minimum and maximum values for $\delta^{13}C_{Base}$ versus $\delta^{15}N_{Base}$ for each individual using the SIBER package [55]. The overlap between groups was characterized through calculation of the Euclidean distance between the centroids for both minimum and maximum $SEA_c$, followed by a residual permutation and Hotelling $t^2$-test to evaluate statistical differences [56,57] between the areal coverage of the two niches ($\alpha = 0.05$) using the package 'Hotelling'.

# 3. Results

## 3.1. Bulk $\delta^{13}C$ and $\delta^{15}N$ values

The $\delta^{13}C$ values for all individuals fell within the range of $-17.5$ to $-20‰$ across all baleens, with oscillations of $0.5-1.5‰$ that generally mirrored changes observed in $\delta^{15}N$ values, with some deviations (figure 1; electronic supplementary material, table S1). $\delta^{13}C$ values for the fin whales were similar among individuals and higher ($-18.9$ to $-19.2‰$) than for the humpback whale ($-19.6‰$), but lower than for the minke whale ($-18.1‰$; one-way ANOVA: $F_{4,252} = 77$, $p < 0.001$). Within-individual variation in $\delta^{15}N$ values was larger than seen for $\delta^{13}C$ values (maximum within-individual range in $\delta^{15}N$ is $11.2-14.8‰$ in the humpback whale). Oscillations in $\delta^{15}N$ values also showed greater amplitude from 0.5 to approximately 3‰, with median values for the humpback and minke plates (12.8‰ and 12.2‰) higher than those for all three fin whale plates (9.3–10‰: electronic supplementary material, table S1). $\delta^{15}N$ values were higher for both the minke and humpback whale (11.8 and 12.8‰, respectively; one-way ANOVA: $F_{4,252} = 238$, $p < 0.001$) than for the fin whales (9.2–10‰).

The three fin whales (FW1, FW2, FW3; figure 2) displayed regular oscillations in $\delta^{15}N$ values that imply baleen growth rates of $2-3.5 \text{ mm week}^{-1}$, calculated based on GAM modelling (table 2), following the approach of Aguilar [58], Giménez [50]. The minke whale (MN, figure 2) showed less regular minima that corresponded to a growth rate of $2.3 \text{ mm week}^{-1}$, while the humpback whale

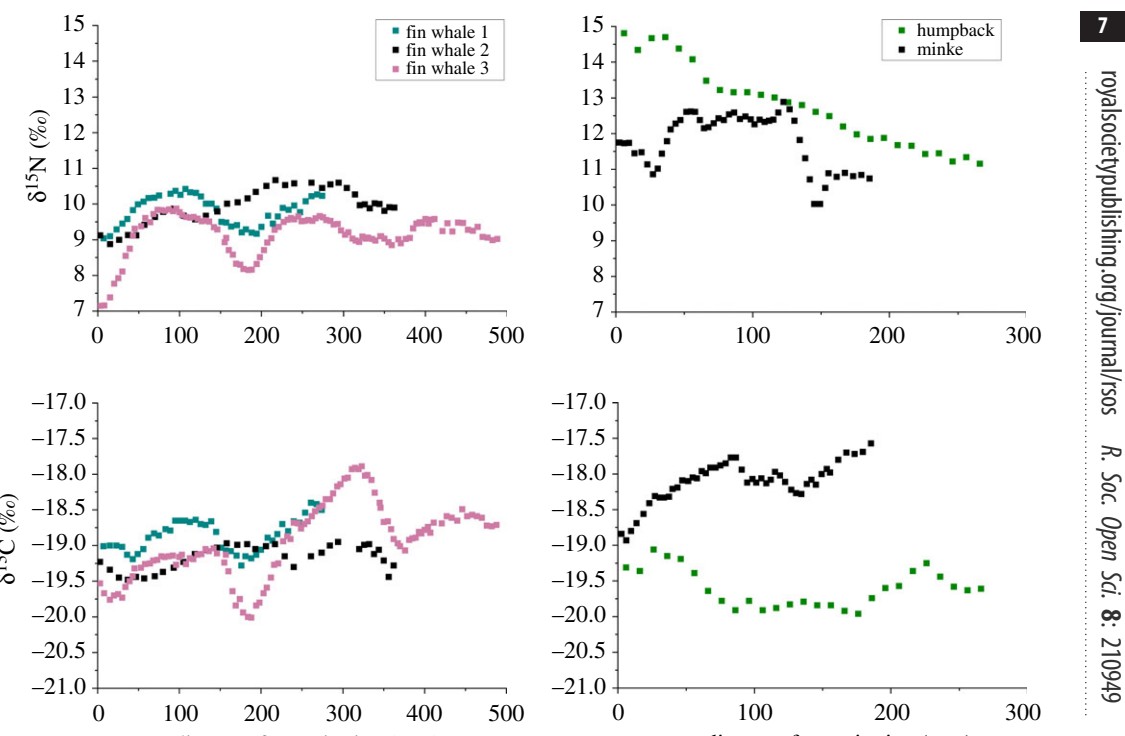

**Figure 1.** Bulk $\delta^{15}$N and $\delta^{13}$C values from incrementally sampled baleen plates of five individual mysticetes originating from the North Atlantic.

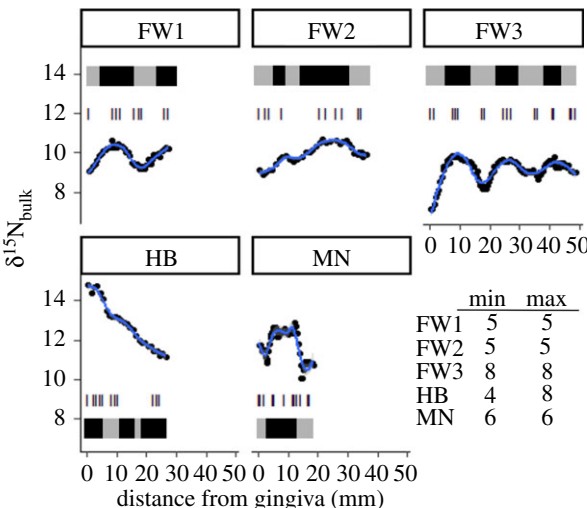

**Figure 2.** GAM models fit to baleen $\delta^{15}$N$_{bulk}$ values to identify minimum and maximum periods for $\delta^{15}$N values in baleen for each individual (blue lines). Black shaded regions indicate periods when $\delta^{15}$N values were above (max) and grey shaded regions indicate periods when $\delta^{15}$N values were below (min) the conditional mean for a linear regression applied to bulk $\delta^{15}$N values. Hash marks indicate sampling intervals for determining $\delta^{15}$N values for amino acids and the table in the bottom right panel indicates the samples located in min and max periods for each individual whale.

displayed no distinct $\delta^{15}$N minima but rather a continuous decrease in $\delta^{15}$N value from the gingiva across the full length of baleen (from 14.8 to 11.6‰; HB, figure 2) with slight oscillations from which no growth estimate could be reasonably estimated. Linear regressions applied to the $\delta^{15}$N values for bulk baleen indicated regions in the baleen that were above (black) and below (grey) the conditional mean (figure 2). The minimum and maximum periods of these oscillations reflect the net effects from metabolism, trophic position and the underlying values of the resources being used during each individual's migrations. Minimum and maximum periods for $\delta^{15}$N values (grey and black bars, figure 2) are thought to reflect residence times in different waters within the Atlantic and Arctic

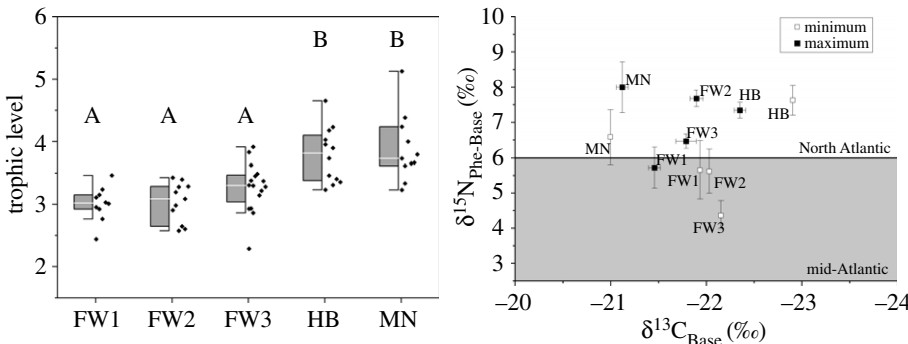

**Figure 3.** (a) Trophic level estimates for each individual and (b) baleen baseline mean $\delta^{13}C_{Base}$ and $\delta^{15}N_{Phe-Base}$ values, minimum and maximum values across the lengths of baleen for each individual; mean ± s.e. The grey shaded area indicates the $\delta^{15}N$ baseline isotope value for mid-Atlantic (2–6.5‰) versus the North Atlantic (6–10‰) oceans (Magozzi et al. [34]; Trueman et al. [60]).

Oceans, with the differences in amplitudes of oscillations reflecting the net effects from different migrations that occurred within an individual's lifetime and the seasonal decrease in the excretion of $^{15}N$ in urine as fasting and catabolism of somatic tissue for energy occurs [58,59]. These minimum and maximum periods were used to target amino acid $\delta^{15}N$ samples (hash marks figure 2) in order to maximize the potential differences between samplings in each period.

## 3.2. $\delta^{15}N$ of amino acids

Trophic levels were estimated using the weighted means of the $\delta^{15}N$ values for trophic and source amino acids (see Methods, equation (2.2)) with means ranging from 3 to 3.2 for the fin whales, 3.7 for the humpback whale and 3.8 for the minke whale (figure 3a). The trophic level estimates did not vary significantly (two-way ANOVA, all $p > 0.05$) between the minimum and maximum periods examined along the baleen. Therefore, single trophic level estimates ($TL_{Trophic-Source}$) were used for each individual whale to establish baseline–corrected $\delta^{13}C$ and $\delta^{15}N$ values ($\delta^{13}C_{Base}$ and $\delta^{15}N_{Base}$; equation (2.2); table 2). This adjustment is also applied to Phe to calculate $\delta^{15}N$ values representing the base of the food web ($\delta^{15}N_{Phe-Base}$) by accounting for fractionation due to trophic level increase (equation (2.3)). Changes in the $\delta^{15}N_{Phe-Base}$ values reflect the differences in underlying regional N source values supporting each individual during the period when the plate was formed. The $\delta^{15}N_{Phe-Base}$ values for time intervals with minimum and maximum bulk $\delta^{15}N$ values were found to be significantly different between individuals and between minimum and maximum $\delta^{15}N$ value periods (two-way ANOVA: individuals, $F_{4,54} = 5.5$, $p \leq 0.001$; min/max $F_{1,54} = 11.8$, $p = 0.001$; interaction: $F_{5,54} = 7$, $p < 0.001$; figure 3b). FW3 was found to have higher $\delta^{15}N_{Phe-Base}$ in the maximum periods for bulk $\delta^{15}N$ values than in the minimum periods (Wilcoxon ranked $t$-test, $Z_8 = -2.5$, $p = 0.008$). While for FW2 the $\delta^{15}N_{Phe-Base}$ values were also higher in the maximum bulk $^{15}N$ period than in the minimum periods ($Z_5 = -1.9$, $p = 0.06$), but not statistically significant using a threshold of $\alpha = 0.05$ (figure 3b).

Glu and Ala are trophic amino acids (figure 4a,b) whose $\delta^{15}N$ values indicate the amount of metabolic reworking occurring during metabolism [38]. Thr and Asp, Gly and Ser are metabolic amino acids that provide indications for diet composition and fasting state of the individuals, respectively [41,42]. $\delta^{15}N$ values of Phe were subtracted from all amino acids $\delta^{15}N$ values to adjust for changes in baseline values (figure 4b,c). $\delta^{15}N$ values for Ala, Asp, Glu and Ser were not significantly different between minimum and maximum periods (two-way ANOVA, all $p > 0.05$). $\delta^{15}N_{Thr}$ values were higher for the fin whales (one-way ANOVA: $F_{4,53} = 6.8$, $p < 0.001$; figure 4b) with a wider range (−29.3 to −11.2‰) than for the humpback whale (−28.4 to −25.9‰) and minke whale (−32.3 to −21.6‰). The ranges for $\delta^{15}N_{Thr}$ values in the fin (11–18‰) and minke (10.7‰) whales were considerably larger than for any of the other trophic or metabolic amino acids examined (1.4–6.2‰). $\delta^{15}N_{Gly}$ values were significantly different between minimum and maximum periods for individuals (two-way ANOVA: individuals $F_{4,50} = 112$, $p < 0.001$; min/max $F_{1,50} = 23.9$, $p < 0.001$). For all individuals, the mean $\delta^{15}N_{Gly}$ value for the bulk minimum periods was higher than for the bulk maximum, but was only statistically higher for FW3 ($Z_8 = 2$, $p = 0.04$), although FW1 ($Z_8 = 1.9$, $p = 0.06$) was close to being significant at a threshold of $\alpha = 0.05$ (figure 4c). $\delta^{15}N_{Asp}$ values were higher for both the humpback and minke than for the fin whales, and for $\delta^{15}N_{Ser}$ the humpback had higher values than all the other whales (one-way ANOVAs; Asp $F_{4,55} = 13.5$, $p < 0.001$; Ser $F_{4,55} = 14.6$, $p < 0.001$; figure 4a). $\delta^{15}N_{Ala}$ and $\delta^{15}N_{Glu}$

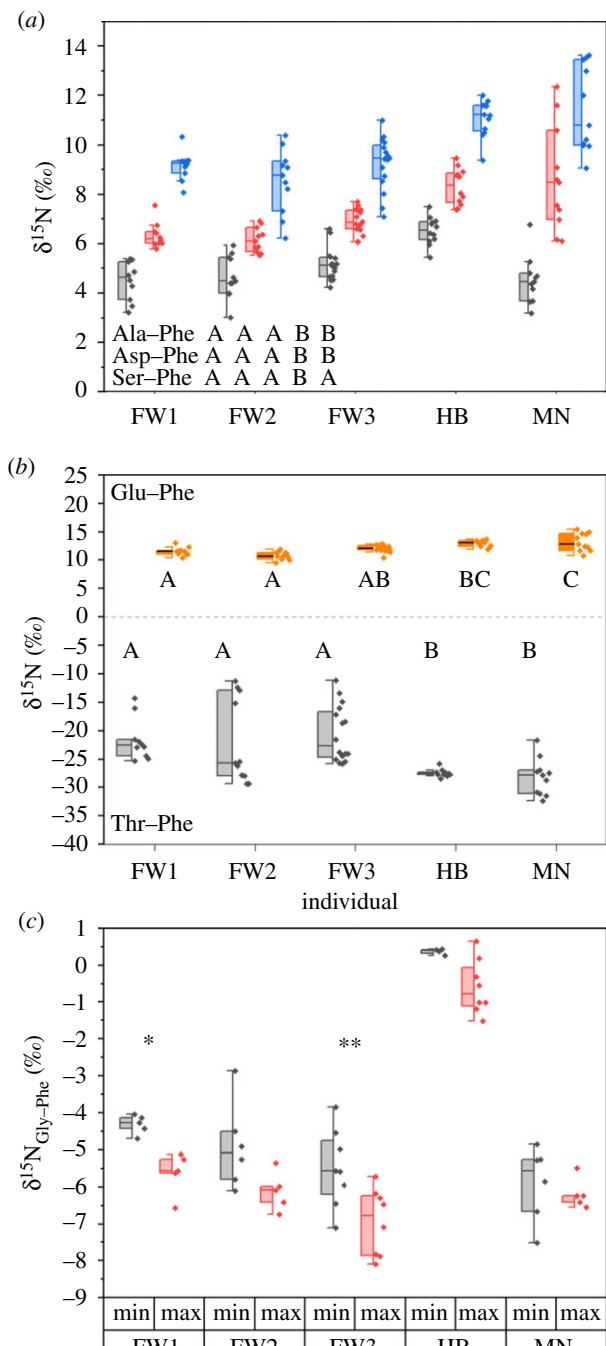

**Figure 4.** $\delta^{15}$N for (*a*) alanine, aspartic acid and serine; (*b*) glutamic acid and threonine and (*c*) glycine for the five mysticete individuals to assess trophic effects and possible starvation and fasting effects between individuals and baleen periods, respectively. All AAs have been corrected against Phe to remove underlying source AA variation. Letters indicate significant differences as indicated by a *post hoc* Tukey's test ($\alpha = 0.05$). For Gly, Wilcoxon ranked *t*-tests were used to compare between baleen regions for each individual. * indicates $p = 0.06$ and ** indicates $p < 0.05$.

values for the trophic amino acids were generally higher for the humpback and minke whales (one-way ANOVAs; Ala $F_{4,55} = 14.2$, $p < 0.001$; Glu $F_{4,55} = 14.1$, $p < 0.001$), but *post hoc* Tukey's indicated different relationships between individuals (figure 4*a,b*) with FW3 being similar to the HB for Glu.

## 3.3. Isotopic niche overlap within individuals

Isotopic niche areas (SEA$_c$) were calculated by applying a Bayesian statistical model (SIBER package) to the larger dataset of trophic level corrected $\delta^{13}C_{Base}$ and $\delta^{15}N_{Base}$ values (equations (2.4) and (2.5)).

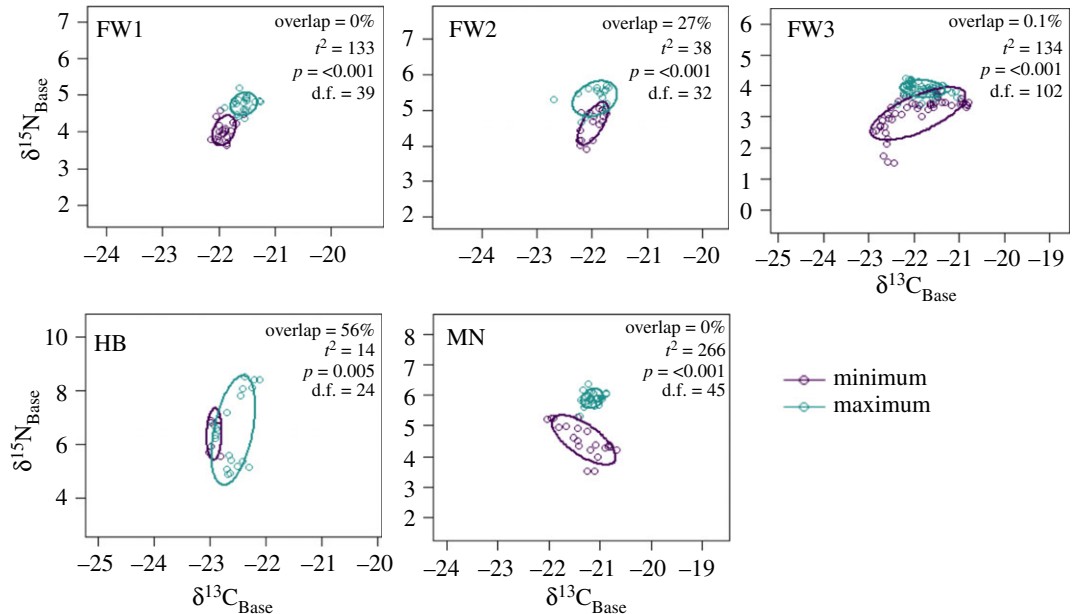

**Figure 5.** Standard ellipse areas corrected for sample size for baleen regions pooled into minimum and maximum periods as determined by GAM models on $\delta^{15}$N values for each individual. Overlap is the percentage of ‰$^2$ areal overlap between the periods. Significant $p$-values from Hotelling's $t^2$-test indicate where baleen values occupy different isotopic niches due to feeding in regions with distinct isotope resource values.

The overlap between the minimum and maximum baleen periods ranged from 0 (implying different values for basal resources) to 56% (indicating some overlap) with Hotelling's $t^2$-test, indicating significant differences between periods for all individuals (figure 5). Separation of SEA$_c$ areas for periods predominately occurred along the $y$-axis ($\delta^{15}$N$_{Base}$) for all individuals besides HB, where separation occurred across the $x$-axis ($\delta^{13}$C$_{Base}$).

# 4. Discussion

## 4.1. Growth rates for baleen

The marked oscillations in $\delta^{15}$N values of baleen are assumed to reflect residence times in mid-Atlantic breeding grounds (minima) and high-latitude feeding grounds (maxima) with substantial differences in the underlying $\delta^{15}$N values for POM in these regions (−1 to 1‰ versus 6–10‰, respectively) combined with the impacts from metabolism and excretion of N on $\delta^{15}$N values depending on fasting status during migrations [58,59]. The changes in amplitude of the oscillations in baleen reflect resource use and feeding status during migrations that occurred within an individual's lifetime. The humpback whale did not display readily apparent oscillations in $\delta^{15}$N values, but rather a distinct increase in $\delta^{15}$N value that progressed along the length of baleen, suggesting different migration behaviour than observed for the other whales in this study. The estimates for baleen growth rates for the four individuals with recurring minima in $\delta^{15}$N$_{bulk}$ values (FW1–3 and MN; 10.4–16.3 cm yr$^{-1}$; table 1) agree with previous estimates from blue [16], minke (12.9 cm yr$^{-1}$, [61]) and bowhead whales (*Balaena mysticetus*; 16–25 cm yr$^{-1}$, [18]), all of which assume continuous growth across seasons. The length of plates in this study are relatively short due to the availability of stranded and bow-caught animals examined and represent a maximum of 3 years of migration (FW3, with four $\delta^{15}$N minima) within these individuals. Other studies have examined considerably longer plates [16,60,62].

## 4.2. Trophic levels

Trophic level estimates from minimum and maximum regions of $\delta^{15}$N values along the baleen are expected to reflect the largest contrasts between food resources throughout the multi-year periods recorded in the baleen. Trophic level estimates from amino acid $\delta^{15}$N values indicated no significant changes in trophic levels across the baleen records for these five individuals using any of the three

approaches (TDF of 3.6 calculated from previous trophic level estimates using Glu and Phe [48], TDF of 3.1 following estimate for delphinid TDF using Glu and Phe [49] or TDF of 3.6 using multiple trophic and source AAs) presented. Consistent trophic levels throughout migratory periods reflect a more or less continuous utilization of the same prey, without considerable periods of specialization or switching between smaller fishes and krill during migration and no major effect of seasonal variations in $^{15}N$ excretion rates. The trophic levels for the different species showed significantly lower estimates (one-way ANOVA: $F_{4,59} = 11.6$, $p < 0.001$) for the fin whales than for the humpback and minke whales (figure 3a). This is fully consistent with the fact that fin whales preferentially consume krill in areas where these are abundant and only occasionally feeding opportunistically on schools of small fish when krill is scarce [63,64]. The higher reconstructed trophic levels for both humpback and minke whales are expected as their foraging typically includes larger number of small fishes, less than 30 cm length, such as herring (Clupea harengus) and sprat (Sprattus sprattus) [65–67]. Trophic level estimates observed in the minimum and maximum regions of $\delta^{15}N$ values along the baleen are expected to reflect the largest contrasts between food resources throughout the multi-year periods recorded in the plate. Trophic level estimates for all of the whales (ranging from 3 to 3.8) were based on a relatively small TDF (3.6‰) compared with the typically applied value of 7.6‰ used for estimation of lower trophic levels (e.g. fish and invertebrates) [39,40]. A TDF of 4‰ yielded comparable trophic levels for zooplanktivorous whales (bowhead whales, Balaena mysticetes TL of 1.9–3.0) as determined through amino acid analysis and stomach contents (TL 3) in Matthews et al. [68]. Smaller TDFs reflect increased similarity between the protein quality of the diet and the consumer, resulting in less reworking and, therefore, less fractionation of amino acid N during metabolism [40,69] and application of smaller TDFs in marine mammals has been found to be appropriate [49]. In future work, it may be useful to further account for the protein quality differences between prey types (e.g. krill, fish) using scaled TDF equations, but this is outside the scope of this study.

## 4.3. Metabolism of whales

The amino acids Gly and Thr have been found to respond to fasting and protein deficiency in mammals through variable enrichments in $\delta^{15}N$ values for each AA [41,42] as catabolism of tissue protein occurs leading to a negative nitrogen balance coinciding with metabolism of stored lipid resources. The $\delta^{15}N$ values of glucogenic amino acids (Gly, Ser, proline and Asp) are expected to increase as exogenous material becomes limited and endogenous protein starts to be metabolized [70]. Fasting has been previously thought to occur as whales migrate southward out of their northern feeding grounds toward breeding sites [71] and feeding becomes more confined to opportunistic feeding events [58,62]. We observed no fasting effects between minimum and maximum periods for baseline-corrected $\delta^{15}N_{Ser}$ or $\delta^{15}N_{Asp}$ (figure 4a) but $\delta^{15}N_{Gly}$ values were statistically higher in the minimum periods for bulk $\delta^{15}N$ values than in the maximum periods for bulk $\delta^{15}N$ in FW3 (figure 4c). However, all five individuals had higher averages for $\delta^{15}N_{Gly}$ values in minimum periods ranging from 0.3 to 1.4‰ and the difference for FW2 was nearly statistically significant ($p = 0.06$). Furthermore, patterns of significant differences between individuals for Gly and Phe did not align, probably reflecting different processes affecting the metabolic versus source amino acids. These isotopic enrichments for $\delta^{15}N$ are lower than the observed shift (2–6‰) in glucogenic amino acids for fasting southern elephant seals (Mirounga leonina, [42]). This suggests that the impacts from fasting on glucogenic $\delta^{15}N$ values for mysticetes may be more limited than during breeding and moulting events in other marine mammals. This smaller observed effect may be due to either the considerable body size and lipid stores or through subsistence with opportunistic feeding offsetting more extreme fasting effects [72]. Although limited in scale, isotopic enrichment of the $\delta^{15}N_{Gly}$ values onset in the same manner (during minimum periods) across all individuals regardless of species (figure 4c) and suggests that there is a fasting effect that occurs as fat stores are accessed during migrations toward breeding grounds when feeding becomes opportunistic.

Higher $\delta^{15}N_{Thr}$ values in mammals have been observed to coincide with reduced protein quality in their diet causing reduced reverse fractionation with higher $\delta^{15}N_{Thr}$ values indicating periods of potential starvation [41] although this mechanism is incompletely characterized. Extremely low values for $\delta^{15}N_{Thr}$ are typical of marine mammals from higher trophic levels [73]. Both humpback and minke whales have patterns of consistently low values (−27.4 ± 0.7‰ and −28.2 ± 3.2‰, respectively; figure 4a) that are expected for adequate protein availability throughout the period covered by the baleen examined. However, the fin whales (FW1–3) had higher $\delta^{15}N_{Thr}$ values (−22.1 to −20.6‰) and larger ranges (11–18‰) due to higher $\delta^{15}N$ values (−15 to −11‰) for Thr occurring intermittently throughout the

baleen records. The range in $\delta^{15}N$ values observed for Thr were 2–13× those observed for trophic (Glu, Ala) or other metabolic AAs (Gly, Ser, Asp) and Thr and Glu had different patterns (figure 4b) indicating that Thr behaves differently to the 'canonical' trophic amino acid. These higher values observed for Thr are potentially the result of protein deficiency and may mark protein deficiency or starvation events across an individual's lifetime. This finding conflicts with decreased $\delta^{15}N_{Thr}$ values observed in elephant seal whiskers during fasting [42], suggesting that there may be multiple effects impacting the metabolism of Thr during fasts that are more or less severe and warrant further investigation. There was no strong correlation between minimum and maximum periods for bulk $\delta^{15}N$ values in the baleen and differences in $\delta^{15}N_{Thr}$ suggest more episodic onset than the more regularly occurring fasting effects observed for $\delta^{15}N_{Gly}$. Given these observations, the fin whales appear to have been more regularly under food stress than either the humpback or minke whale individuals examined in this study.

## 4.4. Resource utilization

Minimum and maximum periods in baleen bulk $\delta^{15}N$ values had distinct values for both $\delta^{15}N_{Phe–Base}$ (figure 3b) and the isotope niches formed using the wider dataset of $\delta^{13}C_{Base}$ and $\delta^{15}N_{Base}$ (equations (2.4) and (2.5); figure 5). Distinct differences between the minimum and maximum bulk $\delta^{15}N$ periods probably reflect underlying isotopic differences in resource values supporting the food web between mid-Atlantic breeding grounds and high-latitude feeding grounds. Although few individuals have been tracked, all three species have been observed to make southerly winter migrations away from high-latitude feeding grounds (greater than 70°N) with North Atlantic fin whales having been observed to migrate to the Azores [8], humpback whales as far south as the Antillean islands [74] and minke whales observed off the east coast of Florida [75]. These large geographical separations between breeding and feeding grounds coincide with distinct underlying isotopic values for POM in those regions. Isoscapes, i.e. geographical maps of the underlying yearly averages for regional isotopic values of carbon and nitrogen [34,35], characterize isotopic ranges for the POM sampled from both the mid-Atlantic ($\delta^{13}C$ −20 to −24‰; $\delta^{15}N$ −1 to 1‰) and high-latitude Arctic ($\delta^{13}C$ −28 to −30‰; $\delta^{15}N$ 6–10‰). The $\delta^{13}C$ and $\delta^{15}N$ values from these regions vary depending on local biogeochemical processes (e.g. lower $\delta^{15}N$ values associated with oligotrophic conditions) and serve as variable end members for the source amino acids incorporated into baleen as it is produced from bloodstream metabolites derived from the animal's diet. The variations in those values are likely to be dampened and never reflect the end member values from the underlying isoscape depending on the relative feeding intensity (opportunistic feeding in transit), variations in seasonal values for underlying biogeochemical processes and the relative turnover of the internal source AA pool during migration and breeding. The ranges in $\delta^{15}N_{Phe–Base}$ (1.9–9.9‰), $\delta^{13}C_{Base}$ (−20.7 to −23‰) and $\delta^{15}N_{Base}$ (1.5–8.5‰) values all fall within the ranges expected for dietary intake of resources sourced from regions with distinct baseline isoscape values. Significant differences were observed between minimum and maximum regions for $\delta^{15}N_{Phe–Base}$ that did not align with those observed for $\delta^{15}N_{Gly–Phe}$ or $\delta^{15}N_{Gly}$ values, indicating that Phe fractionated differently to Gly in this dataset. Therefore, we have assumed that the fractionation associated with Phe is primarily due to changes in underlying biogeochemical values with minimal impacts from fasting, although other studies have observed considerable metabolic impacts [42].

The larger ranges observed for $\delta^{15}N_{Phe}$ and $\delta^{15}N_{Base}$ (8‰ and 7‰, respectively) versus $\delta^{13}C_{Base}$ (2.3‰) are probably due to the large amounts of lipid stores that are primarily developed with time spent on feeding grounds [60]. Under fasting conditions, lipid stores will be used as a source of C with a relatively light $\delta^{13}C$ value that reflects the fractionation of C from food resources containing the regional $\delta^{15}N$ values where they were consumed [76]. These lipid stores are expected to be mobilized during times of limited feeding and reduce the impact of C derived from incidental feeding on $\delta^{13}C_{Base}$ values (hysteresis) of the baleen during fasting conditions as metabolites from blood are incorporated into baleen. Use of C from lipid stores contributes to the dampening of variation in C values along the baleen, and although the metabolic N pool in whales is quite large, there is no comparable storage pool for N. Therefore, N from incidental feeding is expected to be more directly metabolized into animal tissues, while carbon from lipid-rich prey can be metabolically routed to either direct incorporation to tissue or to storage within large lipid stores depending on feeding status [77]. $\delta^{15}N_{Base}$ values (range of minima from 1.5 to 5‰) never reach the expected low isoscape values for the mid-Atlantic of approximately −1 to 1‰ as the lower concentrations of N from opportunistic feeding may be insufficient to fully overcome the comparatively high $\delta^{15}N$ fed on extensively at

**Table 3.** Baleen growth estimates. GAM results assessing the fluctuations of $\delta^{15}$N in baleen plates and the resulting growth estimates from these models.

| individual | $n$ | E.D.F. | $F$ | adjusted $R^2$ | $p$-value | deviance explained (%) | $\delta^{15}$N minima interval (cm) | weekly growth interval (mm) |
|---|---|---|---|---|---|---|---|---|
| fin whale 1 | 42 | 8.6 | 82 | 0.95 | <0.001 | 95.6 | 15.6 | 3 |
| fin whale 2 | 35 | 8.7 | 92 | 0.96 | <0.001 | 96.9 | 10.4 | 2 |
| fin whale 3 | 105 | 8.8 | 112 | 0.91 | <0.001 | 91.4 | 13.5, 16.3, 18.1 | 2.6–3.5 |
| humbback | 27 | 3 | 279 | 0.98 | <0.001 | 97.8 | n.a. | n.a. |
| minke | 48 | 8.4 | 39 | 0.88 | <0.001 | 90.2 | 12.2 | 2.3 |

higher latitude feeding grounds (6–10‰). Alternatively, higher than expected $\delta^{15}$N$_{Base}$ values may reflect poorly constrained TDFs and should be considered in future work aiming to anchor $\delta^{15}$N$_{Base}$ values for marine mammals to biogeochemical isoscapes.

Differentiation between periods of migration is supported in four individuals (FW1–FW3, MN), with clear separation between niche areas for basal resources that predominantly separate along the $\delta^{15}$N$_{Base}$ axis (overlap < 28%, Hotelling's $t^2$-test, $p < 0.001$; figure 5). The humpback whale appears to predominately separate along the $\delta^{13}$C$_{Base}$ axis which may reflect utilization of benthic resources within coastal margins, a lack of opportunistic feeding during migrations [16], or a relatively reduced latitudinal migration indicative of a temperate feeding style where individuals preferably feed in the temperate zone and reduce reliance on high-latitude feeding [62]. The gradual increase in $\delta^{15}$N value across the baleen record of the humpback whale from around 11.5–15‰ probably reflects increased contribution of herring and other small fish during feeding. This difference did not result from a shift in regional sources as Phe$_{Base}$ was not significantly different between these samplings ($W_6 = 1.3$, $p = 0.2$). The combination of (i) no baseline shift for Phe$_{Base}$ and (ii) no difference for $\delta^{15}$N$_{Gly}$ values between minimum and maximum periods indicating fasting effects, support the hypothesis of higher trophic level predation within the same environment (e.g. coastal margins) across time. High $\delta^{15}$N$_{Gly}$ values observed for the humpback (approx. 0‰; figure 4$b$) also mirror $\delta^{15}$N$_{Gly}$ values found in small fish migrating from estuarine into coastal waters where elevated $\delta^{15}$N$_{Gly}$ values have been previously observed, such as herring [45].

In the above discussions, it should be noted that individuals FW1 and FW3 that have statistically relevant differences between $\delta^{15}$N$_{Phe–Base}$ and $\delta^{15}$N$_{Gly}$, were males, and the remaining three individuals were female. Females are likely to display different isotopic patterns for both C and N as they reproduce, as the result of gestation and lactation altering the partitioning of resources and resultant isotopic values [78]. These effects remain unaccounted for in this analysis due to the limited knowledge of these individual's life histories as baleen was sampled from beaching and bow-catch events. Additionally, the use of isoscapes to provide yearly averages for underlying isotope values for POM ignores variability that is expected during seasonal changes, although examinations of this variance are increasingly common [60,79]. Seasonal and temporal isotopic variations can be quite large [34,35], especially for arctic or near arctic waters where the bulk of mysticete feeding occurs, and should be further considered in future work (table 3).

# 5. Conclusion

This study used isotope analysis of bulk material and amino acids to provide metabolic and trophic context to the isotopic oscillations observed within whale baleen. Trophic level estimates from source and trophic amino acids were higher for the humpback and minke whales than for the fin whales, which corresponds to previous observations. Trophic level estimates using amino acids allow for the correction of bulk stable isotope values from consumers to the underlying baseline values of the primary producers supporting regional foodwebs. Isotopic niche areas constructed from these baseline values using periods of minimum and maximum bulk $\delta^{15}$N values further confirm distinct differences between maximum periods that reflect feeding in high-latitude feeding grounds and minimum periods that reflect resources used from mid-Atlantic breeding grounds for these individuals. $\delta^{15}$N values from Gly and Thr provided useful metabolic indicators across the baleen sequences. Gly had

higher values that aligned with considerable time spent in breeding grounds where feeding is expected to become incidental and fasting is likely to occur. Differences in Thr occurred more episodically and indicate that food stress or starvation occurred more often in the fin whales in this study relative to the other whales examined. Analysis of $\delta^{15}$N values for amino acids provided further context into the movements and metabolic conditions for mysticete whales, information that is especially important in wide ranging, difficult to track, animals with threatened conservation status.

Ethics. All applicable international, national and/or institutional guidelines for the care and use of animals were followed. Transport and handling of material from protected indigenous and exotic animal species was extended to P.M.R. from Utrecht University's Licence Wnb/2018/039 (valid through 1 February 2024) by the Faculty of Veterinary Medicine, referred to in Article 3.6, second paragraph, of the Nature Conservation Act.

Data accessibility. Data available from 4TU.Research Data: Amino acid isotope dataset http://dx.doi.org/10.4121/16578503 and Bulk isotope dataset http://dx.doi.org/10.4121/16578512. The data are provided in electronic supplementary material [80].

Competing interests. We declare we have no competing interests.

Funding. Funding was provided through the project 'Waddensleutels' funded by 'Waddenfonds' (grant no. WF203930).

Acknowledgements. We thank Jort Ossebaar, Monique Verweij and Ronald von Bommel for technical support for this project. We thank Kees Camphuysen for the input provided during the conceptual stages of this work. Clive Trueman provided considerable food for thought on the first draft of this manuscript and we are grateful for his considered input and conversation during lockdown. We thank two anonymous reviewers for their comments which improved the manuscript.

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
