## [Peer Review File · Royal Society Open Science]

Review History

RSOS-201704.R0 (Original submission)

Review form: Reviewer 1

Is the manuscript scientifically sound in its present form?

Yes

Are the interpretations and conclusions justified by the results?

Yes

Is the language acceptable?

Yes

Do you have any ethical concerns with this paper?

No

Have you any concerns about statistical analyses in this paper?

Yes

Recommendation?

Major revision is needed (please make suggestions in comments)

Comments to the Author(s)

See attached (Appendix A).

Review form: Reviewer 2**Is the manuscript scientifically sound in its present form?**

No

Are the interpretations and conclusions justified by the results?

No

Is the language acceptable?

No

Do you have any ethical concerns with this paper?

No

Have you any concerns about statistical analyses in this paper?

No

Recommendation?

Reject

Comments to the Author(s)

Riekenberg et al. present a bulk tissue (d13C and d15N) and amino acid (d15N) isotope dataset to generate movement, foraging, and physiological information for limited number (n=1-2) of serially sub-sampled baleen plates from 3 whale species. The isotopic patterns are compelling, but the paper does not do a good job of exploring (or constraining) alternative explanations for the data presented. This issue is most obvious in the Discussion, where several sections (e.g., Metabolism of Whales and Resource Utilization) do not do into great depth in providing explanations for the patterns observed, especially those potentially driven by physiological or biochemical mechanisms. For example, the authors should lean on a rich literature of isotope turnover data derived from experimental dietary manipulations. Because isotopic incorporation is influenced by organism size, scaling turnover data from smaller mammals may not be possible, but at least patterns from experiments could inform and potentially constrain explanations provided by the authors regarding isotopic (C versus N) turnover. Furthermore, the assessment of trophic level based on AA d15N values is too complicated, difficult to follow, and fraught with assumptions regarding trophic-source AA TDF values. The scaling exercise used to estimate TDFs based on differences between expected and observed bulk tissue trophic discrimination is unprecedented and problematic. Lastly, the paper is difficult to follow and could benefit from careful editing to improve clarity. There are several examples of sentences being too long (e.g., Lines 30-33) and/or have unnecessarily complicated structure (e.g., Lines 37-40). The paper may be suitable for RSOP, but requires significant revision that includes alternative explanations for the bulk tissue and amino acid isotope patterns. Without a more in depth discussion of alternative explanations, the paper reads as a "just so" story that doesn't carefully consider the ecological and physiological complexity of either the bulk tissue or amino acid isotope data.

Introduction

Lines 48-49. The latter half of this sentence is a bit ambiguous in regards to “mixture of influencing factors and often provide results that are inconclusive or muddled”, perhaps be more specific about these caveats so readers better understand motivations.

Lines 65-69. Seems like this statement requires a citation, perhaps O’Connell 2017?

Methods

Amino Acid Measurement (Lines 113-121). I realize that some information may be in Reference 42, but readers shouldn’t have to look up another paper for critical methodological information that will aid future studies. Some important details are missing from this section including where the measurements were made and what reference materials (standards) were used to assess analytical precision, and how were those reference materials prepared and analyzed in relation to unknown samples, and how were raw measurements corrected/reduced with data from reference materials? Were samples derivatized alongside unknown samples in batches and run accordingly? Like other types of isotopic measurements at the bulk tissue level (e.g., d2H analysis), these kinds of measurements are complicated and so analytical protocols need to be clear for readers to push the science forward.

Trophic Level (Lines 138-165). I’m reasonably familiar with trophic level calculations based on amino acid d15N data, however, I found the section in the Methods on this topic to be very confusing and full of assumptions. For example, why use a beta value for the Glu-Phe trophic-source pair when Equation 2 utilizes weighted mean values for grouped trophic and source amino acids? Furthermore, why was bulk tissue discrimination (1.8 per mil, Busquets Vass et al. 2019) used to derive a trophic-source TDF for grouped trophic and source amino acids? I’ve never seen this kind of sliding scale approach to estimate AA specific TDFs? Furthermore, why assume that whales discriminate amino acids similar to invertebrates and use a 7.6 per mil TDF as an end-member in the scaling approach. There are data for AA-specific TDFs for marine mammals (see review by McMahon and McCarthy 2016) which show that TDFs for the canonical Glu-Phe pair is lower (5-6 per mil) than reported for invertebrates and fish, but also note that estimates vary among fish species with different feeding modes and the overall average is still significantly smaller than 7.6 per mil (Bradley et al. 2015). Lastly, there are no error bars on any of the parameters in Equations 2-5, and as such error is not properly propagated to estimates of trophic level or the subsequent baseline estimates for carbon and nitrogen isotope values. Overall, the myriad assumptions made and lack of error in the calculation of trophic and dependent estimates of baseline isotopic composition is a major flaw with the current version of the manuscript.

Lines 159-161. Here and elsewhere, isotope data should be reported to one decimal point because that is the degree of analytical precision on bulk tissue and amino acid isotope measurements.

Discussion

Lines 242-243. What is meant by “differences in excretion of d15N”? Animals don’t excrete isotope values, they excrete isotopes. Perhaps provide a bit more descriptive information about the mechanism(s) by why animals discriminate isotopes when assimilated resources and synthesizing tissues.

Lines 273-276. Is a TL of 2.7 realistic for a krill specialist? Assuming primary producers (phytoplankton) are TL 1, are krill strict herbivores with a TL of <2? I believe they are more indiscriminant omnivorous filter feeders with a TL of somewhere between 2 and 2.5, which would put a krill specialist like a fin or blue whale at TL 3 or slightly higher. The whole point of

this is that the TL estimates, specifically that TDF used in Equation 2 is not very well constrained. Instead of trying to estimate TL, the authors should consider using their data to constrain a TDF value for whales by mining the literature (e.g., reference 60) for estimates of TL for these species and then back-calculating a TDF with Equation 2. Or perhaps using a Glu-Phe specific equation (e.g., McMahon and McCarthy 2016) to calculate a TDF_{Glu-Phe} that would be extremely useful for future studies that use amino acid nitrogen isotope data to study whale ecology. In general, beta values for phytoplankton-based pelagic food webs are better constrained than trophic-source TDF values (even for the canonical pair Glu-Phe), so using published literature to constrain the latter is a really good use of the data that will benefit future studies.

Line 280-282. It must be more than lipid catabolism that drives variation in amino acid d¹⁵N during fasting. Fix to better tie with changes in nitrogen balance.

Line 282. Do you mean glucogenic (not glycogenic) amino acids here that are catabolized during gluconeogenesis (reference 63)? If so, the list is much longer than the 4 AAs listed so why were enrichments only observed in Gly? At the very least I would expect Ser to also show enrichments because Gly is synthesized from Ser (via one step) and both of these (simple) AAs share the same nitrogen atom (no amination occurs during that step). Were any fasting-related patterns observed in Phe, which is another glucogenic AA?

Lines 303-305. Why would protein catabolism lead to increases in Thr values when the general pattern is that Thr values decrease with increasing trophic level. In other words, if starvation induced catabolism has the same isotopic effect as an increase in trophic level (as the organism consumes itself), then why would Thr values get higher and not lower? I believe the work on fasting elephant seals clearly shows that Thr d¹⁵N values get lower (not higher) during nutritional stress that yields protein catabolism. Given the strange pattern reported here, the fact that (1) high Thr values were observed “intermittently” and were not associated with periods of the annual life cycle when fasting may occur (e.g., migration), and (2) Thr is a difficult AA to measure because it typically elutes close to other AAs, I suggest that authors revisit these outlier values and if they are sound please provide a possible mechanism for Thr enrichment.

Line 332. I agree that incorporation must play a big role in the dampened patterns in baleen relative to the end-member baseline values associated with the southern breeding and northern foraging grounds. I think this section could be strengthened by directly comparing the bulk baleen data to published isoscapes for the region to assess whether the whales analyzed here ever equilibrate with the end-members? This also has implications for how AA isotope patterns are interpreted because individual AAs likely have variable isotopic incorporation rates, and thus those that turnover more quickly might be more faithful recorders of local isotopic conditions than ones that turnover more slowly.

Lines 336-339. I'm curious if there is any experimental data in other mammals to show that nitrogen actually turns over faster than carbon? The Trueman et al. 2019 paper (which is not cited correctly) is suggestive of differences in turnover rates for carbon versus nitrogen, but it presents data from a single animal from a single species. There is a very rich literature on C and N isotope turnover rates for smaller mammals that the authors could lean on here to better strength these claims, as there are biochemical conditions that could promote greater turnover in C versus N isotope turnover and vice versa.

Lines 342-344. Yes, but in order for lipid carbon to be deposited in baleen, it has to be used to synthesize non-essential amino acids, and in order to do that it requires nitrogen from a central N pool. So its not clear to me why this would promote faster turnover in C in comparison to N? Also keep in mind that ~1/3 of the amino acids in baleen are essential, which can't be synthesized de novo and must be sourced from exogenous (diet) or endogenous (tissue

catabolism or gut microbes) sources; as such, carbon and nitrogen isotope turnover is probably similar for this essential pool of AAs. I also don't understand why lower concentration of nitrogen in prey would contribute to lower turnover because N concentrations of whale tissues (e.g., baleen) mirror that of their diet; the easily assimilate portions of krill or fish (e.g., muscle or internal organs) have the same C/N ratios as whale baleen (or muscle). So for a carnivore like a mysticete whale, I doubt that variation in the C versus N concentrations of food sources has any impact on consumer tissue turnover. Again, are there any experimental studies the authors can rely on here to support these claims?

Lines 356-358. Why is this increase related to trophic level, couldn't it also be related to a change in foraging location? What do the AA $\delta^{15}\text{N}$ data say about this trend?

Lines 358-360. This interpretation of $\delta^{15}\text{N}$ Gly assumes that Gly is largely routed directly from diet, however, Gly is the simplest amino acid and as such is very easy to synthesize *de novo*. Couldn't the enriched Gly $\delta^{15}\text{N}$ values be driven by other intrinsic (physiological) or extrinsic (ecological) factors? I suggest other potential explanations are evaluated here. For example, Gly $\delta^{15}\text{N}$ has also been shown to be very sensitive to fasting in marine mammals.

Decision letter (RSOS-201704.R0)

Dear Dr Riekenberg

The Editors assigned to your paper RSOS-201704 "Reconstructing the diet, trophic level, and migration pattern of Mysticete whales based on baleen isotopic composition." have made a decision based on their reading of the paper and any comments received from reviewers.

Regrettably, in view of the reports received, the manuscript has been rejected in its current form. However, a new manuscript may be submitted which takes into consideration these comments.

We invite you to respond to the comments supplied below and prepare a resubmission of your manuscript. Below the referees' and Editors' comments (where applicable) we provide additional requirements. We provide guidance below to help you prepare your revision.

Please note that resubmitting your manuscript does not guarantee eventual acceptance, and we do not generally allow multiple rounds of revision and resubmission, so we urge you to make every effort to fully address all of the comments at this stage. If deemed necessary by the Editors, your manuscript will be sent back to one or more of the original reviewers for assessment. If the original reviewers are not available, we may invite new reviewers.

Please resubmit your revised manuscript and required files (see below) no later than 27-Jul-2021. Note: the ScholarOne system will 'lock' if resubmission is attempted on or after this deadline. If you do not think you will be able to meet this deadline, please contact the editorial office immediately.

Please note article processing charges apply to papers accepted for publication in Royal Society Open Science (<https://royalsocietypublishing.org/rsos/charges>). Charges will also apply to

papers transferred to the journal from other Royal Society Publishing journals, as well as papers submitted as part of our collaboration with the Royal Society of Chemistry (<https://royalsocietypublishing.org/rsos/chemistry>). Fee waivers are available but must be requested when you submit your manuscript (<https://royalsocietypublishing.org/rsos/waivers>).

Thank you for submitting your manuscript to Royal Society Open Science and we look forward to receiving your resubmission. If you have any questions at all, please do not hesitate to get in touch.

on behalf of Dr Asha de Vos (Associate Editor) and Pete Smith (Subject Editor)
openscience@royalsociety.org

Associate Editor Comments to Author (Dr Asha de Vos):

The reviewers have provided extensive commentary, which we hope will be useful for the authors. Unfortunately, we do not feel able to recommend a decision other than 'reject/resubmit' on this occasion: while both referees see merit in the work, it seems clear that the authors have more work to do to get the paper to a publishable standard. We would encourage the authors to engage closely with the commentary of the reviewers and provide a thorough revision of their paper before resubmitting for further consideration. Good luck!

Reviewer comments to Author:

Reviewer: 1
Comments to the Author(s)
See attached file

Reviewer: 2

Comments to the Author(s)
Riekenberg et al. present a bulk tissue (d13C and d15N) and amino acid (d15N) isotope dataset to generate movement, foraging, and physiological information for limited number (n=1-2) of serially sub-sampled baleen plates from 3 whale species. The isotopic patterns are compelling, but the paper does not do a good job of exploring (or constraining) alternative explanations for the data presented. This issue is most obvious in the Discussion, where several sections (e.g., Metabolism of Whales and Resource Utilization) do not do into great depth in providing explanations for the patterns observed, especially those potentially driven by physiological or biochemical mechanisms. For example, the authors should lean on a rich literature of isotope turnover data derived from experimental dietary manipulations. Because isotopic incorporation is influenced by organism size, scaling turnover data from smaller mammals may not be possible, but at least patterns from experiments could inform and potentially constrain explanations provided by the authors regarding isotopic (C versus N) turnover. Furthermore, the assessment of trophic level based on AA d15N values is too complicated, difficult to follow, and fraught with assumptions regarding trophic-source AA TDF values. The scaling exercise used to estimate TDFs based on differences between expected and observed bulk tissue trophic discrimination is unprecedented and problematic. Lastly, the paper is difficult to follow and could benefit from careful editing to improve clarity. There are several examples of sentences being too long (e.g.,

Lines 30-33) and/or have unnecessarily complicated structure (e.g., Lines 37-40). The paper may be suitable for RSOP, but requires significant revision that includes alternative explanations for the bulk tissue and amino acid isotope patterns. Without a more in depth discussion of alternative explanations, the paper reads as a “just so” story that doesn’t carefully consider the ecological and physiological complexity of either the bulk tissue or amino acid isotope data.

Introduction

Lines 48-49. The latter half of this sentence is a bit ambiguous in regards to “mixture of influencing factors and often provide results that are inconclusive or muddled”, perhaps be more specific about these caveats so readers better understand motivations.

Lines 65-69. Seems like this statement requires a citation, perhaps O’Connell 2017?

Methods

Amino Acid Measurement (Lines 113-121). I realize that some information may be in Reference 42, but readers shouldn’t have to look up another paper for critical methodological information that will aid future studies. Some important details are missing from this section including where the measurements were made and what reference materials (standards) were used to assess analytical precision, and how were those reference materials prepared and analyzed in relation to unknown samples, and how were raw measurements corrected/reduced with data from reference materials? Were samples derivatized alongside unknown samples in batches and run accordingly? Like other types of isotopic measurements at the bulk tissue level (e.g., d2H analysis), these kinds of measurements are complicated and so analytical protocols need to be clear for readers to push the science forward.

Trophic Level (Lines 138-165). I’m reasonably familiar with trophic level calculations based on amino acid d15N data, however, I found the section in the Methods on this topic to be very confusing and full of assumptions. For example, why use a beta value for the Glu-Phe trophic-source pair when Equation 2 utilizes weighted mean values for grouped trophic and source amino acids? Furthermore, why was bulk tissue discrimination (1.8 per mil, Busquets Vass et al. 2019) used to derive a trophic-source TDF for grouped trophic and source amino acids? I’ve never seen this kind of sliding scale approach to estimate AA specific TDFs? Furthermore, why assume that whales discriminate amino acids similar to invertebrates and use a 7.6 per mil TDF as an end-member in the scaling approach. There are data for AA-specific TDFs for marine mammals (see review by McMahon and McCarthy 2016) which show that TDFs for the canonical Glu-Phe pair is lower (5-6 per mil) than reported for invertebrates and fish, but also note that estimates vary among fish species with different feeding modes and the overall average is still significantly smaller than 7.6 per mil (Bradley et al. 2015). Lastly, there are no error bars on any of the parameters in Equations 2-5, and as such error is not properly propagated to estimates of trophic level or the subsequent baseline estimates for carbon and nitrogen isotope values. Overall, the myriad assumptions made and lack of error in the calculation of trophic and dependent estimates of baseline isotopic composition is a major flaw with the current version of the manuscript.

Lines 159-161. Here and elsewhere, isotope data should be reported to one decimal point because that is the degree of analytical precision on bulk tissue and amino acid isotope measurements.

Discussion

Lines 242-243. What is meant by “differences in excretion of d15N”? Animals don’t excrete isotope values, they excrete isotopes. Perhaps provide a bit more descriptive information about

the mechanism(s) by why animals discriminate isotopes when assimilated resources and synthesizing tissues.

Lines 273-276. Is a TL of 2.7 realistic for a krill specialist? Assuming primary producers (phytoplankton) are TL 1, are krill strict herbivores with a TL of <2? I believe they are more indiscriminant omnivorous filter feeders with a TL of somewhere between 2 and 2.5, which would put a krill specialist like a fin or blue whale at TL 3 or slightly higher. The whole point of this is that the TL estimates, specifically that TDF used in Equation 2 is not very well constrained. Instead of trying to estimate TL, the authors should consider using their data to constrain a TDF value for whales by mining the literature (e.g., reference 60) for estimates of TL for these species and then back-calculating a TDF with Equation 2. Or perhaps using a Glu-Phe specific equation (e.g., McMahon and McCarthy 2016) to calculate a TDF_{Glu-Phe} that would be extremely useful for future studies that use amino acid nitrogen isotope data to study whale ecology. In general, beta values for phytoplankton-based pelagic food webs are better constrained than trophic-source TDF values (even for the canonical pair Glu-Phe), so using published literature to constrain the latter is a really good use of the data that will benefit future studies.

Line 280-282. It must be more than lipid catabolism that drives variation in amino acid d¹⁵N during fasting. Fix to better tie with changes in nitrogen balance.

Line 282. Do you mean glucogenic (not glycogenic) amino acids here that are catabolized during gluconeogenesis (reference 63)? If so, the list is much longer than the 4 AAs listed so why were enrichments only observed in Gly? At the very least I would expect Ser to also show enrichments because Gly is synthesized from Ser (via one step) and both of these (simple) AAs share the same nitrogen atom (no amination occurs during that step). Were any fasting-related patterns observed in Phe, which is another glucogenic AA?

Lines 303-305. Why would protein catabolism lead to increases in Thr values when the general pattern is that Thr values decrease with increasing trophic level. In other words, if starvation induced catabolism has the same isotopic effect as an increase in trophic level (as the organism consumes itself), then why would Thr values get higher and not lower? I believe the work on fasting elephant seals clearly shows that Thr d¹⁵N values get lower (not higher) during nutritional stress that yields protein catabolism. Given the strange pattern reported here, the fact that (1) high Thr values were observed “intermittently” and were not associated with periods of the annual life cycle when fasting may occur (e.g., migration), and (2) Thr is a difficult AA to measure because it typically elutes close to other AAs, I suggest that authors revisit these outlier values and if they are sound please provide a possible mechanism for Thr enrichment.

Line 332. I agree that incorporation must play a big role in the dampened patterns in baleen relative to the end-member baseline values associated with the southern breeding and northern foraging grounds. I think this section could be strengthened by directly comparing the bulk baleen data to published isoscapes for the region to assess whether the whales analyzed here ever equilibrate with the end-members? This also has implications for how AA isotope patterns are interpreted because individual AAs likely have variable isotopic incorporation rates, and thus those that turnover more quickly might be more faithful recorders of local isotopic conditions than ones that turnover more slowly.

Lines 336-339. I'm curious if there is any experimental data in other mammals to show that nitrogen actually turns over faster than carbon? The Trueman et al. 2019 paper (which is not cited correctly) is suggestive of differences in turnover rates for carbon versus nitrogen, but it presents data from a single animal from a single species. There is a very rich literature on C and N isotope turnover rates for smaller mammals that the authors could lean on here to better strength these

claims, as there are biochemical conditions that could promote greater turnover in C versus N isotope turnover and vice versa.

Lines 342-344. Yes, but in order for lipid carbon to be deposited in baleen, it has to be used to synthesize non-essential amino acids, and in order to do that it requires nitrogen from a central N pool. So its not clear to me why this would promote faster turnover in C in comparison to N? Also keep in mind that ~1/3 of the amino acids in baleen are essential, which can't be synthesized de novo and must be sourced from exogenous (diet) or endogenous (tissue catabolism or gut microbes) sources; as such, carbon and nitrogen isotope turnover is probably similar for this essential pool of AAs. I also don't understand why lower concentration of nitrogen in prey would contribute to lower turnover because N concentrations of whale tissues (e.g., baleen) mirror that of their diet; the easily assimilate portions of krill or fish (e.g., muscle or internal organs) have the same C/N ratios as whale baleen (or muscle). So for a carnivore like a mysticete whale, I doubt that variation in the C versus N concentrations of food sources has any impact on consumer tissue turnover. Again, are there any experimental studies the authors can rely on here to support these claims?

Lines 356-358. Why is this increase related to trophic level, couldn't it also be related to a change in foraging location? What do the AA $\delta^{15}\text{N}$ data say about this trend?

Lines 358-360. This interpretation of $\delta^{15}\text{N}$ Gly assumes that Gly is largely routed directly from diet, however, Gly is the simplest amino acid and as such is very easy to synthesize de novo. Couldn't the enriched Gly $\delta^{15}\text{N}$ values be driven by other intrinsic (physiological) or extrinsic (ecological) factors? I suggest other potential explanations are evaluated here. For example, Gly $\delta^{15}\text{N}$ has also been shown to be very sensitive to fasting in marine mammals.

===PREPARING YOUR MANUSCRIPT===

Your revised paper should include the changes requested by the referees and Editors of your manuscript. You should provide two versions of this manuscript and both versions must be provided in an editable format:
 one version identifying all the changes that have been made (for instance, in coloured highlight, in bold text, or tracked changes);
 a 'clean' version of the new manuscript that incorporates the changes made, but does not highlight them. This version will be used for typesetting if your manuscript is accepted.

If you have been asked to revise the written English in your submission as a condition of publication, you must do so, and you are expected to provide evidence that you have received language editing support. The journal would prefer that you use a professional language editing service and provide a certificate of editing, but a signed letter from a colleague who is a native

speaker of English is acceptable. Note the journal has arranged a number of discounts for authors using professional language editing services (<https://royalsociety.org/journals/authors/benefits/language-editing/>).

===PREPARING YOUR REVISION IN SCHOLARONE===

<https://royalsociety.org/journals/authors/author-guidelines/#supplementary-material> to include a suitable title and informative caption. An example of appropriate titling and captioning may be found at https://figshare.com/articles/Table_S2_from_Is_there_a_trade-

off_between_peak_performance_and_performance_breadth_across_temperatures_for_aerobic_sc
ope_in_teleost_fishes_/3843624.

Author's Response to Decision Letter for (RSOS-201704.R0)

See Appendix B.

RSOS-210949.R0

Review form: Reviewer 1

Is the manuscript scientifically sound in its present form?

No

Are the interpretations and conclusions justified by the results?

Yes

Is the language acceptable?

Yes

Do you have any ethical concerns with this paper?

No

Have you any concerns about statistical analyses in this paper?

No

Recommendation?

Major revision is needed (please make suggestions in comments)

Comments to the Author(s)

See attached file (Appendix C).

Review form: Reviewer 2

Is the manuscript scientifically sound in its present form?

No

Are the interpretations and conclusions justified by the results?

No

Is the language acceptable?

Yes

Do you have any ethical concerns with this paper?

No

Have you any concerns about statistical analyses in this paper?

No

Recommendation?

Major revision is needed (please make suggestions in comments)

Comments to the Author(s)

This is the second round of comments I'm providing on Riekenberg et al. so I will focus my current comments on the response file furnished with the resubmitted manuscript. I've organized this review on the numbered responses to my original comments. Overall, I cannot recommend this manuscript for publication in ROS in its current form because the trophic position (TP) estimates based on amino acid (AA) $\delta^{15}\text{N}$ values are flawed, and require further consideration. The authors method for calculating TP produces estimates that are <3 (e.g., 2.7 for fin whales?), which suggests that these species are omnivores assuming phytoplankton has a TP of 1 (Equation 2). How can a baleen whale be an omnivore, especially a fin or humpback whale that are known to consume a combination of krill and schooling fish (e.g., herring, capelin)? Decades of dietary data based on stomach content analysis and observation shows that baleen whales (especially these species) are not omnivores. Lastly, the manuscript could use some grammatical editing (e.g., Lines 28-29), which was also suggested in my first round of comments. The manuscript should be carefully read and edited before resubmission to ROS or another journal.

Comment #2. Despite what the literature says (Chikaraishi et al. 2009, 2014), using beta and source-trophic TDF for Glu-Phe in an estimate of trophic position (TP) based on a mixture of trophic and source AA values is fundamentally flawed as each trophic-source pair has their own TDF and beta. For example, TDFs and beta values for Ala-Lys are very different than for Glu-Phe, so how can one justify only using betas for the latter pair when estimating TP. A lot has been done on this topic in the past ~10 years since these pioneering papers were published, and while the authors complicated method for estimating TP produces a TDF of ~4 per mil, this estimate is based on a starting value of 7.6 per mil, which might have been the canonical value a decade ago but has not stood up to further analysis and scrutiny (e.g., McMahon and McCarthy 2016), especially for the group of animals (marine mammals) under study in this manuscript (Matthews et al. 2020).

Comment #6. I appreciate the additional information regarding the AA isotope analysis and I've thoroughly read the authors methods papers in RCMS, which is very similar to the protocol we use in our lab to make these measurements. One additional detail I would like to see in the supplemental material is a table reporting the average with-run analytical precision (e.g., SD) for each AA accompanied by a short description of any patterns in these data; e.g., variation in precision likely due to coelution or other analytical challenges. This should be easy to compile for the runs used to generate the data presented in this manuscript.

Comment #8. See concerns above regarding the use of Glu-Phe beta values in Equation 2 that relies on a mixture of 8 trophic and source AAs. I agree that nowhere in the manuscript do the authors mention using a TDF for fish and invertebrates, but that is exactly what the canonical 7.6 per mil TDF for Glu-Phe represents, and while scaling the Glu-Phe TDF in the way the authors have done here tries to adjust this value to be more representative of a mammal (with a different N excretion pathway than a fish/invert), Equation #2 and the assumptions therein still produce

TP estimates that are on the cusp of being ecologically possible for these species, which consume a mixture of (large) zooplankton and fish. As such, these species should have TP estimates that are above 3.

Comment #12. Stating that the TP estimate for fin whales is better than that produced by Matthews et al. (2020) of 2.0 for bowhead whales is not reassuring. Matthews et al. reported that TP estimate to prove a point, which is that TDF for large whales is much lower than the 7.6 per mil value. Furthermore, stating that the estimate for fin whales (2.7) is close to that of bowhead whales (3.2) is simply not true, it's half a trophic level different and ecologically meaningful. If its "close", then how confident can we be that AA-derived estimates of trophic position are robust and useful?

Comment 14. Do the authors have an explanation for the apparent decoupling of nitrogen in Gly versus Ser? Is it possibly that these two AAs are derived from different pools (e.g., endogenous versus exogenous), or that one might be synthesized de novo and the other is routed directly from protein in diet? More physiology and a better explanation for the unexpected dissimilarity in these two closely related AAs is needed.

Comment 16. Does POM from the mid-Atlantic really have lower d13C values than POM from the Arctic? But more importantly, are the authors confusing isotopic turnover (some times called incorporation) with assimilation? Neither the Cherry et al. 2011 or Newsome et al. 2014 study examined isotopic turnover in AA so its not clear to me how the authors are relating these two studies to their statement regarding differential turnover of nitrogen and carbon? I think the authors are talking about macromolecular routing here and potential de novo synthesis of proteinaceous tissues from lipid carbon, which was the topic of both papers listed above in wild polar bears and lab-raised house mice.

Comment 18. Yes, but the bulk (tissue) values are a mixture of essential and non-essential AAs given that the samples being measured are largely proteinaceous in character. As such, that statement that the higher variation observed in d15N is likely driven by increased turnover in N in comparison to C is not supported by the literature, nor do the authors present a feasible mechanism for this statement. The metabolic pool of N in a whale is quite large, especially if you consider the major onboard reservoir of muscle, so while I understand that the endogenous lipid pool is larger (with lots of C, but little N), I don't understand the mechanisms by which increased isotopic variation is driven by higher isotopic turnover?

Decision letter (RSOS-210949.R0)

Dear Dr Riekenberg

The Editors assigned to your paper RSOS-210949 "Reconstructing the diet, trophic level, and migration pattern of Mysticete whales based on baleen isotopic composition." have now received comments from reviewers and would like you to revise the paper in accordance with the reviewer comments and any comments from the Editors. Please note this decision does not guarantee eventual acceptance.

Please submit your revised manuscript and required files (see below) no later than 21 days from today's (ie 28-Jul-2021) date. Note: the ScholarOne system will 'lock' if submission of the revision is attempted 21 or more days after the deadline. If you do not think you will be able to meet this deadline please contact the editorial office immediately.

on behalf of Dr Asha de Vos (Associate Editor) and Pete Smith (Subject Editor)
openscience@royalsociety.org

Associate Editor Comments to Author (Dr Asha de Vos):
Associate Editor

Comments to the Author:

This manuscript has been thoroughly reviewed by both reviewers who have highlighted some key concerns that need much work. I hope you will be able to consider their suggestions and improve this paper in order for us to consider its publication.

Reviewer comments to Author:

Reviewer: 1

Comments to the Author(s)

See attached file.

Reviewer: 2

Comments to the Author(s)

This is the second round of comments I'm providing on Riekenberg et al. so I will focus my current comments on the response file furnished with the resubmitted manuscript. I've organized this review on the numbered responses to my original comments. Overall, I cannot recommend this manuscript for publication in ROS in its current form because the trophic position (TP) estimates based on amino acid (AA) d15N values are flawed, and require further consideration.

The authors method for calculating TP produces estimates that are <3 (e.g., 2.7 for fin whales?), which suggests that these species are omnivores assuming phytoplankton has a TP of 1 (Equation 2). How can a baleen whale be an omnivore, especially a fin or humpback whale that are known to consume a combination of krill and schooling fish (e.g., herring, capelin)? Decades of dietary data based on stomach content analysis and observation shows that baleen whales (especially these species) are not omnivores. Lastly, the manuscript could use some grammatical editing (e.g., Lines 28-29), which was also suggested in my first round of comments. The manuscript should be carefully read and edited before resubmission to ROS or another journal.

Comment #2. Despite what the literature says (Chikaraishi et al. 2009, 2014), using beta and source-trophic TDF for Glu-Phe in an estimate of trophic position (TP) based on a mixture of trophic and source AA values is fundamentally flawed as each trophic-source pair has their own TDF and beta. For example, TDFs and beta values for Ala-Lys are very different than for Glu-Phe, so how can one justify only using betas for the latter pair when estimating TP. A lot has been done on this topic in the past ~10 years since these pioneering papers were published, and while the authors complicated method for estimating TP produces a TDF of ~4 per mil, this estimate is based on a starting value of 7.6 per mil, which might have been the canonical value a decade ago but has not stood up to further analysis and scrutiny (e.g., McMahon and McCarthy 2016), especially for the group of animals (marine mammals) under study in this manuscript (Matthews et al. 2020).

Comment #6. I appreciate the additional information regarding the AA isotope analysis and I've thoroughly read the authors methods papers in RCMS, which is very similar to the protocol we use in our lab to make these measurements. One additional detail I would like to see in the supplemental material is a table reporting the average with-run analytical precision (e.g., SD) for each AA accompanied by a short description of any patterns in these data; e.g., variation in precision likely due to coelution or other analytical challenges. This should be easy to compile for the runs used to generate the data presented in this manuscript.

Comment #8. See concerns above regarding the use of Glu-Phe beta values in Equation 2 that relies on a mixture of 8 trophic and source AAs. I agree that nowhere in the manuscript do the authors mention using a TDF for fish and invertebrates, but that is exactly what the canonical 7.6 per mil TDF for Glu-Phe represents, and while scaling the Glu-Phe TDF in the way the authors have done here tries to adjust this value to be more representative of a mammal (with a different N excretion pathway than a fish/invert), Equation #2 and the assumptions therein still produce TP estimates that are on the cusp of being ecologically possible for these species, which consume a mixture of (large) zooplankton and fish. As such, these species should have TP estimates that are above 3.

Comment #12. Stating that the TP estimate for fin whales is better than that produced by Matthews et al. (2020) of 2.0 for bowhead whales is not reassuring. Matthews et al. reported that TP estimate to prove a point, which is that TDF for large whales is much lower than the 7.6 per mil value. Furthermore, stating that the estimate for fin whales (2.7) is close to that of bowhead whales (3.2) is simply not true, it's half a trophic level different and ecologically meaningful. If its "close", then how confident can we be that AA-derived estimates of trophic position are robust and useful?

Comment 14. Do the authors have an explanation for the apparent decoupling of nitrogen in Gly versus Ser? Is it possibly that these two AAs are derived from different pools (e.g., endogenous versus exogenous), or that one might be synthesized de novo and the other is routed directly from protein in diet? More physiology and a better explanation for the unexpected dissimilarity in these two closely related AAs is needed.

Comment 16. Does POM from the mid-Atlantic really have lower $\delta^{13}\text{C}$ values than POM from the Arctic? But more importantly, are the authors confusing isotopic turnover (some times called incorporation) with assimilation? Neither the Cherry et al. 2011 or Newsome et al. 2014 study examined isotopic turnover in AA so its not clear to me how the authors are relating these two studies to their statement regarding differential turnover of nitrogen and carbon? I think the authors are talking about macromolecular routing here and potential de novo synthesis of proteinaceous tissues from lipid carbon, which was the topic of both papers listed above in wild polar bears and lab-raised house mice.

Comment 18. Yes, but the bulk (tissue) values are a mixture of essential and non-essential AAs given that the samples being measured are largely proteinaceous in character. As such, that statement that the higher variation observed in $\delta^{15}\text{N}$ is likely driven by increased turnover in N in comparison to C is not supported by the literature, nor do the authors present a feasible mechanism for this statement. The metabolic pool of N in a whale is quite large, especially if you consider the major onboard reservoir of muscle, so while I understand that the endogenous lipid pool is larger (with lots of C, but little N), I don't understand the mechanisms by which increased isotopic variation is driven by higher isotopic turnover?

===PREPARING YOUR MANUSCRIPT===

===PREPARING YOUR REVISION IN SCHOLARONE===

To revise your manuscript, log into <https://mc.manuscriptcentral.com/rsos> and enter your Author Centre - this may be accessed by clicking on "Author" in the dark toolbar at the top of the

page (just below the journal name). You will find your manuscript listed under "Manuscripts with Decisions". Under "Actions", click on "Create a Revision".

Author's Response to Decision Letter for (RSOS-210949.R0)

See Appendix D.

RSOS-210949.R1

Review form: Reviewer 1

Is the manuscript scientifically sound in its present form?

Yes

Are the interpretations and conclusions justified by the results?

Yes

Is the language acceptable?

Yes

Do you have any ethical concerns with this paper?

No

Have you any concerns about statistical analyses in this paper?

No

Recommendation?

Accept as is

Comments to the Author(s)

I believe the authors adequately addressed all reviewer comments and I have no further suggested changes. I commend the authors on their contribution.

Decision letter (RSOS-210949.R1)

Dear Dr Riekenberg,

It is a pleasure to accept your manuscript entitled "Reconstructing the diet, trophic level, and migration pattern of Mysticete whales based on baleen isotopic composition." in its current form for publication in Royal Society Open Science. The comments of the reviewer(s) who reviewed your manuscript are included at the foot of this letter.

If you have not already done so, please ensure that you send to the editorial office an editable version of your accepted manuscript, and individual files for each figure and table included in

your manuscript. You can send these in a zip folder if more convenient. Failure to provide these files may delay the processing of your proof.

===COVID-SPECIFIC TEXT -- WILL ONLY BE ADDED TO COVID-PAPERS BY THE EDITORIAL OFFICE===

COVID-19 rapid publication process:

We are taking steps to expedite the publication of research relevant to the pandemic. If you wish, you can opt to have your paper published as soon as it is ready, rather than waiting for it to be published the scheduled Wednesday.

This means your paper will not be included in the weekly media round-up which the Society sends to journalists ahead of publication. However, it will still appear in the COVID-19 Publishing Collection which journalists will be directed to each week (<https://royalsocietypublishing.org/topic/special-collections/novel-coronavirus-outbreak>).

If you wish to have your paper considered for immediate publication, or to discuss further, please notify openscience_proofs@royalsociety.org and press@royalsociety.org when you respond to this email.

===END OF COVID-SPECIFIC TEXT -- WILL BE REMOVED AS NECESSARY BY THE EDITORIAL OFFICE===

on behalf of Mr Andrew Dunn (Associate Editor) and Pete Smith (Subject Editor)
openscience@royalsociety.org

Associate Editor Comments to Author (Mr Andrew Dunn):

Associate Editor: 1

Comments to the Author:

(There are no comments.)

Associate Editor: 2

Comments to the Author:

(There are no comments.)

Reviewer comments to Author:

Reviewer: 1

Comments to the Author(s)

I believe the authors adequately addressed all reviewer comments and I have no further suggested changes. I commend the authors on their contribution.

Appendix A

Review for RSOS-201704

Reconstructing the diet, trophic level, and migration pattern of Mysticete whales based on baleen isotopic composition

Summary

In this manuscript the authors present a unique isotopic analysis of whale baleen to study the resource use of three Mysticete species. Based on CSIA-AA, the minke and humpback whales sampled had higher TL estimates than the fin whales sampled, following expected ecological patterns. Oscillations in $\delta^{15}\text{N}$ and comparisons of isotope data in oscillation minima and maxima suggest annual migrations throughout the Atlantic. Patterns of $\delta^{15}\text{N}$ Gly and Thr are suggested to reflect alternate periods of forage and fasting.

This is an important, well-written manuscript, being one of the first applications of CSIA-AA to whale baleen and explorations of using CSIA-AA to study fasting in marine mammals. Once revised I believe it will make a good contribution to *Royal Society Open Science*. I appreciate the inclusion of the raw data in the Supplement.

General comments

- Introduction
 - Greater ecological context is needed, specifically re: knowledge gaps in these three specific Mysticete whales and how SIA will help fill those knowledge gaps. There is a lot of ecology in the Discussion that could be moved forward that would remedy this.
 - The introduction is also lacking background re: how SIA can be used to study fasting, a central feature of the discussion. i.e. are they fasting on the breeding grounds or during travel or both? Why do we think AAs will be informative? Which AAs and why? I should note this is a subfield very much in its infancy, so the authors should make efforts not to oversell the approach. Further validation is still ultimately needed.
 - Lastly, the end of the introduction needs clearer project goals and objectives. I appreciate that this is ultimately an analysis of specimens of opportunity and that specific ecological questions may not have underpinned sample collection originally. However, without the whale ecological context and specific objectives this project takes on the feel of an analysis searching for a project.
- Methods
 - There are some issues with how TL is calculated. Because the authors use a mix of trophic and source AAs, β and TDF need to be re-calculated to include all of the same AAs—the authors use values derived from Glu-Phe only. There also needs to be much greater explanation re: the derivation of a TDF of 4 %. I appreciate the efforts to derive a TDF that is biologically realistic and provide a few suggestions below.

- It is not clear to me why all of the data corrections in Lines 154–165 are needed. I comment on this further below but great justification or explanation is likely warranted.

Specific comments

Line 4: ... $\delta^{15}\text{N}$ values from bulk baleen have used....

Line 14: replace “revealed” with “suggested” to reflect the fact that the use of CSIA-AA to study fasting is *very* new.

Lines 28-29: consider also noting that stomach content analyses are biased towards undigestible prey in addition to recent prey.

Line 36: ...and captures a *continuous*, long term record...

Line 43:....diet composition *and habitat use* (10, 21-23).

Line 44: remove “in the system being examined” as variability in TDFs is driven by physiology not location.

Line 50: it would also be worth mentioning in this paragraph that $\delta^{13}\text{C}$ values also vary latitudinally, contributing another source of uncertainty. If the desire is to focus on $\delta^{15}\text{N}$ patterns alone the authors can highlight that variation in the spatial patterns relevant to this study are greater for $\delta^{15}\text{N}$ than $\delta^{13}\text{C}$, if that is the case.

Line 65: exchange “Additionally” for “In contrast”

Line 67: remove “the” at end of this line

Line 70: “TDF” is already defined on line 45 so the acronym can be used here alone

Line 70: Ref 40 should go at the end of this sentence as Chikaraishi et al. 2009 described the classically used CSIA-AA TP equation.

Line 73-76: McMahon and McCarthy 2016 would be the most appropriate reference for this statement.

Lines 87-91: can you provide any information about where in the mouth the baleen was sampled? Does that affect ware patterns? Generally speaking, more information about how the baleen were collected would also be useful, if available. For example, was it cut at the gum line? Inside the gumline? Where do you specifically know where to start sampling?

Line 95-96: include citations for the “~2 to 4 weeks” reference.

Line 112: There are inconsistencies in the use of significant digits throughout the manuscript. I'd recommend two for percentages and one for permil. A few places I noticed this...here, line 146, 147, 176, line 182

Line 131: It would be worth clarifying here what life stage or location the minima and maxima are presumed to be associated with. These are first defined on line 241 in the Discussion so I would recommend just moving that information up or reiterating it here at this first mention.

Line 134-137: Did the authors consider narrower bins for min and max for their analyses? Perhaps the 2nd and 3rd quantiles only? This classification seems to include quite a bit of the migration (intermediate) signature in each bin, which would certainly dampen perceived differences between the endpoints. i.e. the standard ellipses presented in figure 5 might be underestimating the differences. If no changes are made, this would at least be worth mentioning in the Discussion.

Line 145: if a mix of trophic and source AAs are being used to estimate TL then β must be re-calculated using those same AAs.

Line 147-151: Please elaborate on the choice of 4 ‰ as your TDF. There are a lot of assumptions packed into this statement. For example, it is not clear where the value 1.8 ‰ comes from as Busquets-Vass et al. 2017 only presents TDFs for blue whale skin (1.6 ‰). Was this calculated by the authors? I suspect that would be possible based on the data presented by Busquets-Vass et al. 2017. Additionally, 7.6 is a TDF for Glu-Phe only. As for β , the TDF needs to reflect all of the trophic and sources AAs used to estimate TL. From what is written I cannot assess whether that was accounted for. I appreciate the effort to derive a suitable whale baleen TDF given that one is not reported in the literature. However, this analysis may be better served by using a TDF derived from a closely related biological structure such as collagen or cartilage. Below are two controlled feeding studies with AA data for those tissues. Alternatively, the authors could simply use data from Chikaraishi et al. 2009 to calculate a TDF that reflects all of the trophic and source AAs used to calculate TL. I should note that all of these changes would simply affect mean values, so will ultimately not have an impact on comparisons between minima and maxima areas of baleen.

Iain P. Kendall, Michael R.F. Lee & Richard P. Evershed (2017) The effect of trophic level on individual amino acid $\delta^{15}\text{N}$ values in a Terrestrial ruminant food web, STAR: Science & Technology of Archaeological Research, 3:1, 135-145

Wyatt, A. S. J., S. Matsumoto, Y. Chikaraishi, Y. Miyairi, Y. Yokoyama, K. Sato, N. Ohkouchi, and T. Nagata. 2019. Enhancing insights into foraging specialization in the world's largest fish using a multi-tissue, multi-isotope approach. Ecol Monogr 89(1):e01339

Line 150: all that I just mentioned may also influence the values used for error propagation. Please elaborate on the program used to propagate error—I assume the *propagate* package in R?

Line 152-165: Perhaps I am missing something, but I don't really get the point of all of these corrections. It seems to be unnecessarily complicating the analysis. Perhaps there just needs to be greater explanation for why these approaches were used. The authors don't explicitly use these data to link migration patterns to specific habitats with unique SIA values—most of the analysis

relies on comparing data during minima and maxima. Therefore, I am not sure what value these corrections ultimately add to this analysis.

Line 154: Two issues I have with this equation are that (1) there is a bit of double dipping in terms data use since Phe is also in the TL estimate and (2) 0.4 ‰ enrichment for Phe is not universal. In fact, Matthew et al. 2020 suggest Phe possibly fractionates by much more in marine mammals. I recommend using the raw $\delta^{15}\text{N}$ Phe values instead. If this approach was used to standardize across species (due to different diets), the authors could just analyze the species-specific data separately.

Matthews CJD, Ruiz-Cooley RI, Pomerleau C, Ferguson SH. Amino acid $\delta^{15}\text{N}$ underestimation of cetacean trophic positions highlights limited understanding of isotopic fractionation in higher marine consumers. *Ecology and Evolution*. 2020;n/a(n/a).

Lines 157-165: I am also not sure why this is necessary. In terms of niche modeling its standard practice to use the animal $\delta^{15}\text{N}$ and $\delta^{13}\text{C}$ (Suess effect corrected) to reflect overall resource use (diet and habitat). This correction seems to remove some of that variation (the diet component). I imagine the results of the SEA would ultimately be very similar using the bulk SIA data. I also find this approach curious because a major justification for CSIA-AA is that it can overcome all the issues with SIA to better reflect the system baseline. And again, a lot of assumptions go into the mean values used in these equations, adding extra uncertainty.

Line 199: ...and catabolism of somatic tissue for energy.

Line 243: it seems like breeding status is also an important physiological state that has not really been touched on and that could very well have strong impacts on AA biosynthesis.

Line 276: it would be worth listing out ranges of TP estimates for each species from these stomach content studies. Or, better yet, including that information as an additional column in Table 2 for direct comparison.

Line 276: It would be worth noting that species-specific TDFs may in fact be different due to the protein quality factors the authors mentioned. For example, krill-eaters would be expected to have slightly higher TDFs than fish eaters. The trophic transfers within the first two TLs tend to have higher AA TDFs relative to all other TLs, hence the need for multi-TDF equations (McMahon and McCarthy 2016).

Line 282: I suggest expanding on the biogeochemical underpinnings of these patterns, perhaps at the end of this section. Also, were these patterns explored for Ala, Asp, and Ser? The methods indicated these were measured (though no Ser data are in the Supplement). Lübcker et al. 2020 identified fasting related changes in these AAs as well. And, given that they are specifically mentioned here, it is conspicuous they were not included in the analysis as additional support for the fasting component of this study.

Line 298: The patterns observed of decreasing Thr with increasing estimate TP, combined with no difference between minima and maxima (fasting states), suggest Thr is a better TL indicator

than fasting indicator. As outlined by Lübcker et al. 2020, and mentioned here by the authors, there is not very strong biogeochemical justification, as of yet, for using Thr as a fasting indicator. So, I would suggest examining the other glycogenic AAs in this section and moving the topic of Thr to the *Trophic Levels* or *Resource Utilization* sections.

Line 337: Lübcker et al. 2020 also observed significant changes in Phe during fasting, so catabolism of a whale's own tissues during fasting/migration is another source of variability. Parsing the relative effect of this vs. incidental feeding on Phe would be difficult. This Phe fractionation is likely also important to consider given that the author normalize Gly and Thr to Phe.

Line 346: That the $\delta^{15}\text{N}_{\text{base}}$ values do not reach those expected of the system could either mean the whale didn't travel to where we think they should or that the calculation is incorrect. Again, two things difficult to parse. But this might be further justification for removing these corrections and focusing on differences between minima and maxima only.

Line 358: add "relative to the other whales examined" so the reader knows you are comparing to the other species. Otherwise, this could get confused with the use of Gly to examine fasting.

Line 365: Did body size estimates or any other ecological information accompany these samples to make an educated guess on maturity? Identifying reproductive rates and strategies would be pertinent as well. i.e., are females always either pregnant or lactating and thus under some constant physiological burden?

Table 1: This answers my previous statement. I encourage the authors to mention some of this pertinent specimen information at the beginning of the methods.

Appendix B

We thank the two anonymous reviewers that have provided critical feedback that has helped to improve this manuscript.

Reviewer 1

Summary

In this manuscript the authors present a unique isotopic analysis of whale baleen to study the resource use of three Mysticete species. Based on CSIA-AA, the minke and humpback whales sampled had higher TL estimates than the fin whales sampled, following expected ecological patterns. Oscillations in $\delta^{15}\text{N}$ and comparisons of isotope data in oscillation minima and maxima suggest annual migrations throughout the Atlantic. Patterns of $\delta^{15}\text{N}$ Gly and Thr are suggested to reflect alternate periods of forage and fasting.

This is an important, well-written manuscript, being one of the first applications of CSIA-AA to whale baleen and explorations of using CSIA-AA to study fasting in marine mammals. Once revised I believe it will make a good contribution to *Royal Society Open Science*. I appreciate the inclusion of the raw data in the Supplement.

General comments

☑ Introduction

1) Greater ecological context is needed, specifically re: knowledge gaps in these three specific Mysticete whales and how SIA will help fill those knowledge gaps. There is a lot of ecology in the Discussion that could be moved forward that would remedy this.

We have primarily presented an introduction that is focused on the potential to utilize amino acid analysis of baleen to identify migratory patterns and physiological states of the whales being examined. Currently, discussion of the ecology of individual species is primarily limited to the discussion section due to the ecology only becoming necessary to explain isotope differences observed in the baleen between species were revealed through baleen analysis. In this sense, this paper is first a proof of concept about the potential of baleen to provide ecological data, then discussion follows about potential ecological differences between species that drive the observed differences in the data set. We did not go into this work assuming ecological differences drive the isotope patterns, but rather observed statistically significant differences and then looked for potential ecological and physiological differences that could explain these differences. To provide some additional context as requested, we have included relevant references identifying dietary work and the differences between fin whales versus other mysticete species that are more reliant on fish rather than krill.

We now include more ecological context on LNs 49-51: “Previous work has identified that fin whales are predominately krill feeders while humpback and Minke whales feed on small fish such as sprat or herring resulting in comparatively lower $\delta^{15}\text{N}$ values in fin whales when compared to the other two species (29, 30).”

We now clarify that the application of this method is primarily aiming to provide more ecological context once it is more established on LN 93-98: “This work serves as a further proof of concept that amino acids from baleen can be used as a continuous ecological indicator in mysticete whales. Application and further development of this novel method to materials derived from marine mammal strandings have the potential to inform ecological knowledge about marine mammals, particularly in mysticete whales. Once established, this method will help to reconstruct life histories and further identify the ecological overlap between species..”

2) The introduction is also lacking background re: how SIA can be used to study fasting, a central feature of the discussion. i.e. are they fasting on the breeding

grounds or during travel or both? Why do we think AAs will be informative? Which AAs and why? I should note this is a subfield very much in its infancy, so the authors should make efforts not to oversell the approach. Further validation is still ultimately needed.

We agree that a brief mention of the effect of fasting on metabolic amino acids is appropriate to include in the section describing the different amino acid types.

We have now included this description on LN 82: "Metabolic amino acids may fractionate differently under fasting conditions as the whales migrate from their high latitude feeding grounds and fasting or even starvation affects metabolism as feeding becomes limited to incidental encounters (43, 44)."

3) Lastly, the end of the introduction needs clearer project goals and objectives. I appreciate that this is ultimately an analysis of specimens of opportunity and that specific ecological questions may not have underpinned sample collection originally. However, without the whale ecological context and specific objectives this project takes on the feel of an analysis searching for a project.

We have further clarified the goals of this study, primarily to apply CSIA to identify trophic levels using trophic AAs, identify regional use of biogeochemical resources with source AAs, and to examine any potential indicators for metabolism (e.g. starvation or fasting) using metabolic AAs. We now provide a concluding statement that highlights that this study provides evidence that differences in amino acids in baleen can help to provide ecological context across individual's life history and between species.

*LNs 87-100 now reads: "In this study we are among the first to combined bulk stable isotope with analysis of $\delta^{15}\text{N}$ in amino acids from baleen sourced from stranded or bowcaught fin whales ($n=3$, *Balaenoptera physalus*), a stranded humpback whale (*Megaptera novaeangliae*), and a minke whale (*Balaenoptera acutorostrata*), all opportunistically sampled in the Netherlands. We applied $\delta^{15}\text{N}$ values of amino acids to baleen to: 1) resolve trophic levels for these species, 2) identify changes from regional biogeochemical source $\delta^{15}\text{N}$ values being utilized during migrations and 3) characterize any potential metabolic effects from fasting and episodic feeding during migration. This work serves as a further proof of concept that amino acids from baleen can be used as a continuous ecological indicator in mysticete whales. Application and further development of this novel method to materials derived from marine mammal strandings have the potential to inform ecological knowledge about marine mammals, particularly in mysticete whales. Once established, this method will help to reconstruct life histories and further identify the ecological overlap between species. Additionally, we provide evidence of feeding in different regions due to migration between Mid-Atlantic breeding and high-latitude feeding areas through application of isotopic niche areas from trophic level corrected bulk isotope data from baleen."*

Methods

4) There are some issues with how TL is calculated. Because the authors use a mix of trophic and source AAs, β and TDF need to be re-calculated to include all of the same AAs—the authors use values derived from Glu-Phe only. There also needs to be much greater explanation re: the derivation of a TDF of 4 ‰. I appreciate the efforts to derive a TDF that is biologically realistic and provide a few suggestions below.

We have addressed the reviewer's concerns in comments 23 and 24 where we explicitly lay out why we used the set of equations that were applied and further expand upon the reasoning behind the

values for each of the variables that we have applied to this data set. Changes to the text can be found within the respective comments.

23) Line 163: if a mix of trophic and source AAs are being used to estimate TL then β must be recalculated using those same AAs.

The necessity of recalculating β indicated by reviewer 1 is not reflected in the current literature: Chikaraishi et al 2009 presents a calculation that includes the use of the weighted averages from trophic and source AAs along with a TDF of 7.6‰ and the typical β of 3.4, both from Glu-Phe relationships. This form of equation is also applied in Matthews et al. 2020, equation 2 with the explanation of TDF as: “Equations (1) and (2), based on the work of McClelland and Montoya (2002) and Chikaraishi et al. (2007, 2009), apply a single TDF_{Glx-Phe} of 7.6‰ to all trophic transfers, while equations (3) and (4) apply a dual TDF_{Glx-Phe} to account for variation among trophic transfers.”

And the explanation for beta as: “In all equations, $\delta^{15}\text{N}_{\text{Glx}}$ and $\delta^{15}\text{N}_{\text{Phe}}$ are the measured consumer $\delta^{15}\text{N}$ values of those AAs, β is the $\delta^{15}\text{N}$ difference between $\delta^{15}\text{N}_{\text{Glx}}$ and $\delta^{15}\text{N}_{\text{Phe}}$ in primary producers (3.4‰ for algae and cyanobacteria; McClelland & Montoya, 2002)...”

We chose to apply a more compressed TDF to attempt to account for the compression of TPs that occurs in multiple whale species that is well described in Matthews et al. 2020. Additionally, we chose to apply the least controversial β value of 3.4‰ reflecting algae and cyanobacteria because there remains a lack of appropriate phytoplankton producer measurements of β from open ocean, oligotrophic systems. Using lysine (Lys) for constraining β is problematic, as most laboratories do not measure Lys, with only a handful of references (Vander Zanden et al. 2012, McMahan and McCarthy 2016) including the necessary estimates for this variable from very different ecosystems (seagrass, and across a meta-analysis, respectively). Rather than applying a more speculative or mismatched primary producer β value, we felt it was more appropriate to apply the most often used value of 3.4‰. These considerations point towards the near-term necessity for the CSIA community to obtain estimates for β values that are appropriate for the primary producers within the ecosystems that they are being applied to.

LN 174 now reads: “The calculated TDF (4.0‰) is similar to those presented for amino acids in marine mammals (5-6‰) in McMahan and McCarthy (2016).”

24) Line 167-173: Please elaborate on the choice of 4 ‰ as your TDF. There are a lot of assumptions packed into this statement. For example, it is not clear where the value 1.8 ‰ comes from as Busquets-Vass et al. 2017 only presents TDFs for blue whale skin (1.6 ‰). Was this calculated by the authors? I suspect that would be possible based on the data presented by Busquets-Vass et al. 2017. Additionally, 7.6 is a TDF for Glu-Phe only. As for β , the TDF needs to reflect all of the trophic and sources AAs used to estimate TL. From what is written I cannot assess whether that was accounted for. I appreciate the effort to derive a suitable whale baleen TDF given that one is not reported in the literature. However, this analysis may be better served by using a TDF derived from a closely related biological structure such as collagen or cartilage. Below are two controlled feeding studies with AA data for those tissues. Alternatively, the authors could simply use data from Chikaraishi et al. 2009 to calculate a TDF that reflects all of the trophic and source AAs used to calculate TL. I should note that all of these changes would simply affect mean values, so will ultimately not have an impact on comparisons between minima and maxima areas of baleen.

Iain P. Kendall, Michael R.F. Lee & Richard P. Evershed (2017) The effect of trophic level on individual amino acid $\delta^{15}\text{N}$ values in a Terrestrial ruminant food web, STAR: Science & Technology of Archaeological Research, 3:1, 135-145

Wyatt, A. S. J., S. Matsumoto, Y. Chikaraishi, Y. Miyairi, Y. Yokoyama, K. Sato, N. Ohkouchi, and T. Nagata. 2019. Enhancing insights into foraging specialization in the world's largest fish using a multi-tissue, multi-isotope approach. *Ecol Monogr* 89(1):e01339

The value of 1.8‰ comes from page 17 of Busquets-Vass et al 2017 describing baleen from blue whales as we have stated in LNs 147-151 in our manuscript: “However, the mean (\pm SD) baleen $\delta^{15}\text{N}$ value of the three male blue whales (D:12.2 \pm 0.3; E: 12.4 \pm 0.3; F: 12.3 \pm 0.4; S7 Table, Fig 4D \pm 4F) that presumably remained within the CCS for \sim 2 \pm 3 years prior to death, and by extension were isotopically equilibrated with local food sources, were enriched by only 1.7-1.9‰ relative to local prey sources (10.5 \pm 0.2; S2 Table), and is similar to estimates for skin of captive bottlenose dolphins (1.6 \pm 0.5‰) [47,48].”

We felt that it was more appropriate to utilize the fractionation presented from the baleen values from a blue whale study (Busquets-Vass et al 2017) that included extensive analysis of their underlying prey items in the environment than to utilize more speculative estimates from alternative tissues from less closely related species such as terrestrial ruminants or whale sharks as the reviewer suggests.

Regarding the reviewer's comment about needing a TDF and β that include all of the trophic and source AAs, we have chosen to follow the approach presented in Chikaraishi et al 2009 and Matthews et al. 2020 where the trophic equation includes Σ trophicAA and Σ sourceAA terms, but utilize TDF and β estimates appropriate for the system. We believe this is appropriate due to the lack of system and tissue specific estimates for these variables that are available for mysticetes, especially populations from the North Atlantic. Determination of a TDF or β for mysticetes from these data seems to us an overinterpretation due to the lack of ecological context associated with the samples from five stranded animals.

5) It is not clear to me why all of the data corrections in Lines 177–183 are needed. I comment on this further below but great justification or explanation is likely warranted.

We have now expanded upon our use of adjustments in response to Reviewer 1's comments 26 and 28 below emphasizing our attempts to allow for direct comparison of our baleen isotopic values with the isotopic values from the underlying isoscape for particulate organic matter from the Atlantic Ocean and higher latitudes.

Specific comments

6) Line 4: ... $\delta^{15}\text{N}$ values from bulk baleen have used....

Corrected as indicated.

7) Line 14: replace “revealed” with “suggested” to reflect the fact that the use of CSIA-AA to study fasting is very new.

Corrected as indicated.

8) Lines 28-29: consider also noting that stomach content analyses are biased towards undigestible prey in addition to recent prey.

Corrected as indicated.

LN 28 now reads: “However, prey in the stomach of deceased animal represent the animal’s last meal presence and stomach content (13) is biased and reflects 1) only the most recent feedings, 2) the health status of the animal prior to death (i.e. trophic downgrading due to sickness), and 3) the undigestible portions of the prey.”

9) Line 37: ...and captures a *continuous*, long term record...

Corrected as indicated.

10) Line 45:....diet composition *and habitat use* (10, 21-23).

Corrected as indicated.

11) Line 44: remove “in the system being examined” as variability in TDFs is driven by physiology not location.

In this case, system was referring to species, not location, so I have changed system to species to further clarify.

12) Line 54: it would also be worth mentioning in this paragraph that $\delta^{13}\text{C}$ values also vary latitudinally, contributing another source of uncertainty. If the desire is to focus on $\delta^{15}\text{N}$ patterns alone the authors can highlight that variation in the spatial patterns relevant to this study are greater for $\delta^{15}\text{N}$ than $\delta^{13}\text{C}$, if that is the case.

We now mention the carbon isotope values observed in POM for these areas, but we refrain from discussing the ranges of these values observed in this study as discussion is included on this topic in LNs 322 to 342.

LN 54 now reads: “Both the $\delta^{13}\text{C}$ and $\delta^{15}\text{N}$ values of resources supporting primary production shift substantially both spatially and temporally depending on the balance of biogeochemical processes affecting available inorganic and organic C and N sources within an ecosystem (31-35). This occurs, for example, with wide ranging mysticetes in the north Atlantic Ocean where considerably higher $\delta^{15}\text{N}$ values are observed for particulate organic matter (POM) in the higher latitudes ($>70^\circ\text{N}$, 6 to 10‰) compared to lower mid-Atlantic areas (10°N , -1 to 1‰; (36)). In these same areas $\delta^{13}\text{C}$ values partially overlap, with values from -20 to -25‰ and -19 to -22‰, respectively.”

13) Line 72: exchange “Additionally” for “In contrast”

Corrected as indicated.

14) Line 77: remove “the” at end of this line

Corrected as indicated.

15) Line 75: “TDF” is already defined on line 45 so the acronym can be used here alone

Corrected as indicated.

16) Line 77: Ref 40 should go at the end of this sentence as Chikaraishi et al. 2009 described the classically used CSIA-AA TP equation.

Chikaraishi et al. 2009 is now included here as requested.

17) Line 78-82: McMahon and McCarthy 2016 would be the most appropriate reference for this statement.

McMahon and McCarthy 2016 is now referenced here.

18) Lines 99-104: can you provide any information about where in the mouth the baleen was sampled? Does that affect wear patterns? Generally speaking, more information about how the

baleen were collected would also be useful, if available. For example, was it cut at the gum line? Inside the gumline? Where do you specifically know where to start sampling?

We now specify that baleen plates were cut from below the gumline from each whale on LN 99, but LNs 101 to 108 already stated that the samples were acquired from the gingiva across the entire plate for each baleen. Unfortunately, we do not have any additional context as to where the plates were situated in each animal's mouth and patterns for wear, and we feel that this is well outside the context of what was being examined in this study.

LN 101 now reads: "Baleen plates were cut from below the gumline of dead mysticetes that stranded on the Dutch coast or brought into Dutch harbors caught on a ship's bow in 2012 and 2013 (Table 1)."

LN 110-117 reads: "One baleen plate per individual whale was air dried (>2 days), cleaned with bidistilled water, then dichloromethane, and dried at 40°C for ~10 hours. Powdered keratin was collected using a hand drill (3 mm bit) along the leg (labial side) of the plate at either 0.5 cm or 1 cm intervals (representing a range of ~2 to 4 weeks of growth between samplings, calculated from this study) depending on the relative size of the plate, from the gingiva along the full length of the main plate. Powdered baleen was collected and stored at -20°C until further processing. Since baleen grows continuously throughout a whale's life the material closest to the gingiva reflects the most recently produced layer with the material farther away from the gums reflecting increasingly older periods of the whale's foraging history."

19) Line 113: include citations for the "~2 to 4 weeks" reference.

The two to four weeks estimate here is provided from the GAM sampling presented in this study (Table 3) to provide temporal context as to the range of time that the sampling intervals represent. We have included an indication that this calculation is from the current study.

*We have now clarified this on LN 111: "Powdered keratin was collected using a hand drill (3 mm bit) along the leg (labial side) of the plate at either 0.5 cm or 1 cm intervals (representing a range of ~2 to 4 weeks of growth between samplings, **calculated from this study**) depending on the relative size of the plate, from the gingiva along the full length of the main plate."*

20) Line 114: There are inconsistencies in the use of significant digits throughout the manuscript. I'd recommend two for percentages and one for permil. A few places I noticed this...here, line 146, 147, 176, line 182

Values were corrected to two significant digits for percentages and one for per mil values throughout the manuscript as requested.

21) Line 146: It would be worth clarifying here what life stage or location the minima and maxima are presumed to be associated with. These are first defined on line 241 in the Discussion so I would recommend just moving that information up or reiterating it here at this first mention.

We have now included a brief statement assigning the areas associated with minima and maxima here in addition to the more detailed description on LN 250 in the discussion.

LN 146 now reads: "The marked oscillations in $\delta^{15}\text{N}$ values of baleen are assumed to reflect residence times in Mid-Atlantic breeding grounds (minima) and high-latitude feeding grounds (maxima) with substantial differences in the underlying $\delta^{15}\text{N}$ values for POM in these regions {Trueman, 2019 #578}."

22) Line 155-158: Did the authors consider narrower bins for min and max for their analyses? Perhaps the 2nd and 3rd quantiles only? This classification seems to include quite a bit of the migration (intermediate) signature in each bin, which would certainly dampen perceived differences between the endpoints. i.e. the standard ellipses presented in figure 5 might be

underestimating the differences. If no changes are made, this would at least be worth mentioning in the Discussion.

We did consider more narrow bins for the min and max samples, but as this technique has not been previously applied to baleen we wanted both a statistically defensible method to apply (above and below conditional mean) and a balanced and well-replicated sampling between the periods. This required including more marginal samplings that were closer to the conditional mean than others, especially in animals with less well-defined oscillations (HB, MN). We made the decision to present this more conservative analysis to demonstrate that significant differences were still present in spite of the less targeted approach that could have been applied. We feel that this makes a stronger argument than if the sample numbers would have been reduced and applied only to the extreme values for min and max regions.

We have now included text to describe our reasoning for using the conditional mean as an independent indicator instead of more narrowly defined regions for binning. LN 159 now reads: "Using the conditional mean to demarcate periods provided a robust and independent indicator of minimum and maximum regions, especially in baleen with less well-defined oscillations. It is more conservative than using narrowly binned regions selected in an arbitrary manner."

23) Line 163: if a mix of trophic and source AAs are being used to estimate TL then β must be recalculated using those same AAs.

The necessity indicated by reviewer 1 is not reflected in the current literature: Chikaraishi et al 2009 presents a calculation that includes the use of the weighted averages from trophic and source AAs along with a TDF of 7.6‰ and the typical β of 3.4, both from Glu-Phe relationships. This form of equation is also applied in Matthews et al. 2020, equation 2 with the explanation of TDF as: "Equations (1) and (2), based on the work of McClelland and Montoya (2002) and Chikaraishi et al. (2007, 2009), apply a single TDF_{Glx-Phe} of 7.6‰ to all trophic transfers, while equations (3) and (4) apply a dual TDF_{Glx-Phe} to account for variation among trophic transfers."

And the explanation for beta as: "In all equations, $\delta^{15}\text{N}_{\text{Glx}}$ and $\delta^{15}\text{N}_{\text{Phe}}$ are the measured consumer $\delta^{15}\text{N}$ values of those AAs, β is the $\delta^{15}\text{N}$ difference between $\delta^{15}\text{N}_{\text{Glx}}$ and $\delta^{15}\text{N}_{\text{Phe}}$ in primary producers (3.4‰ for algae and cyanobacteria; McClelland & Montoya, 2002)..."

We chose to apply a more compressed TDF to attempt to account for the compression of TPs that occurs in multiple whale species that is well described in Matthews et al. 2020. Additionally, we chose to apply the least controversial β value of 3.4‰ reflecting algae and cyanobacteria because there remains a lack of appropriate phytoplankton producer measurements of β from open ocean, oligotrophic systems. Incorporating Lys into a measurement of β is problematic, as most laboratories do not measure Lys, with only a handful of references (Vander Zanden et al. 2012, McMahon and McCarthy 2016) including the necessary estimates for this variable from very different ecosystems (seagrass, and across a meta-analysis, respectively). Rather than applying a more speculative or mismatched primary producer β value, we felt it was more appropriate to apply the most often used value of 3.4‰. These considerations point towards the near-term necessity for the CSIA community to produce β values that are appropriate for the primary producers within the ecosystems that they are being applied to.

LN 174 now reads: "The calculated TDF (4.0‰) is similar to those presented for amino acids in marine mammals (5-6‰) in McMahon and McCarthy (2016)."

24) Line 164-170: Please elaborate on the choice of 4 ‰ as your TDF. There are a lot of assumptions packed into this statement. For example, it is not clear where the value 1.8 ‰ comes from as Busquets-Vass et al. 2017 only presents TDFs for blue whale skin (1.6 ‰). Was this calculated

by the authors? I suspect that would be possible based on the data presented by Busquets-Vass et al. 2017. Additionally, 7.6 is a TDF for Glu-Phe only. As for β , the TDF needs to reflect all of the trophic and sources AAs used to estimate TL. From what is written I cannot assess whether that was accounted for. I appreciate the effort to derive a suitable whale baleen TDF given that one is not reported in the literature. However, this analysis may be better served by using a TDF derived from a closely related biological structure such as collagen or cartilage. Below are two controlled feeding studies with AA data for those tissues. Alternatively, the authors could simply use data from Chikaraishi et al. 2009 to calculate a TDF that reflects all of the trophic and source AAs used to calculate TL. I should note that all of these changes would simply affect mean values, so will ultimately not have an impact on comparisons between minima and maxima areas of baleen.

Iain P. Kendall, Michael R.F. Lee & Richard P. Evershed (2017) The effect of trophic level on individual amino acid $\delta^{15}\text{N}$ values in a Terrestrial ruminant food web, STAR: Science & Technology of Archaeological Research, 3:1, 135-145

Wyatt, A. S. J., S. Matsumoto, Y. Chikaraishi, Y. Miyairi, Y. Yokoyama, K. Sato, N. Ohkouchi, and T. Nagata. 2019. Enhancing insights into foraging specialization in the world's largest fish using a multi-tissue, multi-isotope approach. Ecol Monogr 89(1):e01339

The value of 1.8‰ comes from this passage of Busquets-Vass et al 2017; pg 17 of 25 describing baleen from blue whales as stated in LNs 147-151: "However, the mean (\pm SD) baleen $\delta^{15}\text{N}$ value of the three male blue whales (D:12.2 \pm 0.3; E: 12.4 \pm 0.3; F: 12.3 \pm 0.4; S7 Table, Fig 4D \pm 4F) that presumably remained within the CCS for \sim 2 \pm 3 years prior to death, and by extension were isotopically equilibrated with local food sources, were enriched by only 1.7-1.9‰ relative to local prey sources (10.5 \pm 0.2; S2 Table), and is similar to estimates for skin of captive bottlenose dolphins (1.6 \pm 0.5‰ [47,48])."

We felt that it was more appropriate to utilize the fractionation presented from the baleen values from a blue whale study (Busquets-Vass et al 2017) that included extensive analysis of their underlying prey items in the environment than to utilize more speculative estimates from alternative tissues from less closely related species such as terrestrial ruminants or whale sharks as the reviewer suggests.

Regarding the reviewer's comment about needing a TDF and β that include all of the trophic and source AAs included, we have chosen to follow the approach presented in Chikaraishi et al 2009 and Matthews et al. 2020 where the trophic equation includes Σ trophicAA and Σ sourceAA terms, but utilize TDF and β estimates appropriate for the system. We believe this is appropriate due to the lack of system and tissue specific estimates for these variables that are available for mysticetes, especially from populations in the North Atlantic. Determination of a TDF or β for mysticetes from these data points seems like overinterpretation due to the lack of ecological context associated with the samples from five stranded animals.

No changes were made in response to this comment.

25) Line 172: all that I just mentioned may also influence the values used for error propagation. Please elaborate on the program used to propagate error—I assume the *propagate* package in R? *We did not propagate the error in this equation using R as the equation for this propagation has already been quite helpfully presented in Ohkouchi et al. 2018. We incorporated that equation into an excel spreadsheet to propagate the errors associated with the measurements in this study and we have cited this source appropriately.*

LN 175 states: "Error propagation for trophic level gives a standard deviation of ± 0.8 using the equation presented in Ohkouchi, Chikaraishi (39)."

We have now included similar propagations for the baseline corrections for bulk values on LN 182 and 190:

For Phe: "Error propagation indicated a standard deviation of 1‰ for $\delta^{15}\text{N}_{\text{Phe-Base}}$ values." and for baseline bulk $\delta^{15}\text{N}$ and $\delta^{13}\text{C}$: "Error propagation indicated a standard deviation for $\delta^{13}\text{C}_{\text{Base}}$ and $\delta^{15}\text{N}_{\text{Base}}$ of 0.8 and 0.9‰, respectively."

26) Line 176-183: Perhaps I am missing something, but I don't really get the point of all of these corrections. It seems to be unnecessarily complicating the analysis. Perhaps there just needs to be greater explanation for why these approaches were used. The authors don't explicitly use these data to link migration patterns to specific habitats with unique SIA values—most of the analysis relies on comparing data during minima and maxima. Therefore, I am not sure what value these corrections ultimately add to this analysis.

These corrections are required to correct for the small enrichment that occurs in source amino acids during metabolism in order to be able to directly compare the corrected Phe value and the bulk carbon and nitrogen values to the isoscapes previously created from measured carbon and nitrogen isotope values from particulate organic matter. The removal of 0.7 to 1 per mil for Phe is required in order to better anchor these values to the underlying biogeochemical resources that are supporting these animals. This is similar to making sure that all animals and resources have been corrected to the same trophic level (typically that of primary consumer) before applying the mixing models SIAR or SIMMR to account for resource utilization. By including these corrections, we are allowing for the direct comparison of our corrected values to the baseline POM values included in the isoscape.

We have clarified this on LN 191: "By applying trophic corrections for each species and the source amino acid Phe, we allow for direct comparison of any $\delta^{13}\text{C}$ or $\delta^{15}\text{N}$ values against the oceanic isoscape for POM presented in Trueman and St. John Glew 2019."

27) Line 179: Two issues I have with this equation are that (1) there is a bit of double dipping in terms data use since Phe is also in the TL estimate and (2) 0.4 ‰ enrichment for Phe is not mammals. I recommend using the raw $\delta^{15}\text{N}$ Phe values instead. If this approach was used to standardize across species (due to different diets), the authors could just analyze the species-specific data separately.

Matthews CJD, Ruiz-Cooley RI, Pomerleau C, Ferguson SH. Amino acid $\delta^{15}\text{N}$ underestimation of cetacean trophic positions highlights limited understanding of isotopic fractionation in higher marine consumers. *Ecology and Evolution*. 2020;n/a(n/a).

To the author's knowledge, there is no clear way to avoid that source amino acids are used in the accounting for trophic level estimates. Vokshoori et al 2019 demonstrates the necessity of these corrections when trying to tie animal values to the underlying biogeochemical resources that are supporting them within the ecosystem. Without these corrections, trophic level differences will impact species differently and interfere with interpretations of the use of underlying biogeochemical sources within the environments.

Regarding the commentary in Matthews et al 2020 about Phe values, the differences in values may very well reflect partitioning of predatory resources relying on different N values within those environments as a result of partitioning of predation within upwelling, coastal, or pelagic zones. Without the simultaneous inclusion of underlying biogeochemical measurements or the prey items these species are feeding on, it remains quite speculative to assign these differences in Phe to trophic causation. The authors do not feel that it is appropriate to apply a larger and considerably more

speculative fractionation for Phe but believe that is necessary to correct for TL across the species being examined. We have therefore applied the more conservative value and well-established value of 0.4‰ as it reflects the trophic dynamics presented in Chikaraishi et al. 2009 and what has been previously applied in other ecosystems (Vokshoori et al 2019).

Vokshoori, N. L., McCarthy, M. D., Collins, P. W., Etnier, M. A., Rick, T., Eda, M., Beck, J., and Newsome, S. D. 2019. Broader foraging range of ancient short-tailed albatross populations into California coastal waters based on bulk tissue and amino acid isotope analysis. MEPS. 610:1-13

LN 180 now further clarifies that we are applying a similar correction as presented in Vokshoori et al. 2019: "...where 0.4‰ is the small enrichment observed for Phe during metabolism (40) and trophic level calculated for each individual (Table 1) following the method presented in Vokshoori, et al. 2019).

28) Lines 183-188: I am also not sure why this is necessary. In terms of niche modeling its standard practice to use the animal $\delta^{15}\text{N}$ and $\delta^{13}\text{C}$ (Suess effect corrected) to reflect overall resource use (diet and habitat). This correction seems to remove some of that variation (the diet component). I imagine the results of the SEA would ultimately be very similar using the bulk SIA data. I also find this approach curious because a major justification for CSIA-AA is that it can overcome all the issues with SIA to better reflect the system baseline. And again, a lot of assumptions go into the mean values used in these equations, adding extra uncertainty.

Typically, unless considerable sampling of prey items has occurred along with sampling, SIA studies have avoided applying trophic corrections prior to niche modeling. This is largely due to the nature of niche modeling applications, in that you can actually perform them to look for differences without characterizing most of the resources within the system rather than apply a mixing model such as SIMMR or SIAR and receive inconclusive results. Since we have characterized interspecific trophic structure using AAs, we felt it was more appropriate and informative for these species to correct for trophic position for this analysis.

The SEA analysis in this study utilizes the Bulk C and N data that has been corrected for trophic enrichment to the underlying primary producer level in order to allow for comparison of these values against existing isoscapes for particulate organic matter in the Atlantic Ocean basin. No Suess corrections have been applied to this data due to the fairly small time period that these relatively recent samplings encompass (2009-2013).

CSIA does not overcome ALL of the issues with bulk analysis, but the method does provide additional information (source AAs) about the baseline biogeochemical values supporting prey across oceanic basins that allows trophic levels to be more accurately estimated without considerable additional sampling of prey items and primary producers supporting the food web.

We have further clarified that these corrections allow for direct comparison of the whale bulk isotope values against the isoscape for particulate organic matter on LN 191: "By applying trophic corrections for each species and the source amino acid Phe, we allow for direct comparison of any $\delta^{13}\text{C}$ or $\delta^{15}\text{N}$ values against the oceanic isoscape for POM presented in Trueman and St. John Glew 2019."

This is also addressed in our response to comment #26 which has been reproduced here:

These corrections are required to correct for the small enrichment that occurs in source amino acids during metabolism in order to be able to directly compare the corrected Phe value and the bulk carbon and nitrogen values to the isoscapes previously created from measured carbon and nitrogen

isotope values from particulate organic matter. The removal of 0.7 to 1 per mil for Phe is required in order to better anchor these values to the underlying biogeochemical resources that are supporting these animals. This is similar to making sure that all animals and resources have been corrected to the same trophic level (typically that of primary consumer) before applying the mixing models SIAR or SIMMR to account for resource utilization. By including these corrections, we are allowing for the direct comparison of our corrected values to the baseline POM values included in the isoscape.

29) Line 225: ...and catabolism of somatic tissue for energy.
Corrected as indicated.

30) Line 245: it seems like breeding status is also an important physiological state that has not really been touched on and that could very well have strong impacts on AA biosynthesis.

The reviewer is correct, there are potential effects from gestation that likely affect AA biosynthesis, but this was well outside the scope of this study. With only baleen available from these animals and no other life history indicators, we feel that it is inappropriate to speculate on this matter in this study. Furthermore, we state that this subject should be examined more carefully in future studies on LNs 422-427: “In the above discussions it should be noted that individuals FW1 and FW3 have statistically relevant differences between $\delta^{15}\text{N}_{\text{Phe-Base}}$ and $\delta^{15}\text{N}_{\text{Gly}}$, were males, and the remaining three individuals were female. Females are likely to display different isotopic patterns for both C and N as they reproduce, as the result of gestation and lactation altering the partitioning of resources and resultant isotopic values (71). These effects remain unaccounted for in this analysis due to the limited knowledge of these individual’s life histories as baleen was sampled from beaching and bow-catch events.”

No changes resulted from this comment.

31) Line 278: it would be worth listing out ranges of TP estimates for each species from these stomach content studies. Or, better yet, including that information as an additional column in Table 2 for direct comparison.

These references do not provide trophic position estimates, but rather provide a comparative analysis between species that indicate that fin whales utilize more krill than humpback whales and therefore have niches with lower $\delta^{15}\text{N}$ values. The lack of trophic position estimates in this previous work indicates the uncertainty in underlying biogeochemical values in that work that is now accounted for using CSIA techniques. Rather than provide speculative TP values for these species, we have been more conservative and referenced the relevant work to support these author’s interpretations of the relative differences between species. Without CSIA analysis to account for biogeochemical variations in those samplings, it remains inappropriate for us to provide TPs from these references.

No changes resulted from this comment.

32) Line 317: It would be worth noting that species-specific TDFs may in fact be different due to the A protein quality factors the authors mentioned. For example, krill-eaters would be expected to have slightly higher TDFs than fish eaters. The trophic transfers within the first two TLs tend to have higher AA TDFs relative to all other TLs, hence the need for multi-TDF equations (McMahon and McCarthy 2016).

Multi-TDF equations that scale for protein quality differences may be useful for future work, but are a bit outside the scope of this study, which is assessing TP differences from baleen for the first time that the authors are aware of.

We have clarified that multi-TDF strategies may be useful for future work on LN 317: “In future work, it may be useful to further account for the protein quality differences between krill and fish using scaled TDF equations, but this is outside the scope of this study (39).”

33) Line 322: I suggest expanding on the biogeochemical underpinnings of these patterns, perhaps at the end of this section. Also, were these patterns explored for Ala, Asp, and Ser? The methods indicated these were measured (though no Ser data are in the Supplement). Lübcker et al. 2020 identified fasting related changes in these AAs as well. And, given that they are specifically mentioned here, it is conspicuous they were not included in the analysis as additional support for the fasting component of this study.

We have now included analysis for Ala, Ser, Glu, and Asp in figure 4, reproduced below. No differences between minimum and maximum periods were observed for either the trophic or additional metabolic AAs. As for the reviewer's request to examine the biogeochemical underpinnings, we have explicitly tried to remove any source effects to get at the metabolic signal in these AAs by subtracting Phe values from the examined AAs to account for any baseline shifts in $\delta^{15}\text{N}$ values due to migratory movements.

We have now included additional statistical analysis for Ala, Ser, Glu, and Asp on LNs 250-269: "Glu and Ala are trophic amino acids (Fig. 4a&b) that indicate the amount of metabolic reworking occurring during metabolism {Chikaraishi, 2009 #457}. Thr and Asp, Gly, and Ser are metabolic amino acids that provide indications for diet composition and fasting state of the individuals, respectively (50, 51). $\delta^{15}\text{N}$ values of Phe were subtracted from all amino acids $\delta^{15}\text{N}$ values to adjust for changes resulting from baseline values (Fig 4b&c). $\delta^{15}\text{N}$ values for Ala, Asp, Glu, and Ser were not significantly different when examined between minimum and maximum periods (two-way ANOVA, all $p > 0.05$). $\delta^{15}\text{N}_{\text{Thr}}$ values were higher for the fin whales (one-way ANOVA: $F_{4,53} = 6.8$, $p < 0.001$; Fig. 4b) with a wider range (-29.3 to -11.2‰) than for the humpback whale (-28.4 to -25.9‰) and minke whale (-32.3 to -21.6‰). The ranges for $\delta^{15}\text{N}_{\text{Thr}}$ values in the fin (11 to 18‰) and Minke (10.7‰) whales were considerably larger than for any of the other trophic or metabolic amino acids examined (1.4 to 6.2‰). $\delta^{15}\text{N}_{\text{Gly}}$ values were significantly different between minimum and maximum periods for individuals (two-way ANOVA: Individuals $F_{4,50} = 112$, $p < 0.001$; min/max $F_{1,50} = 23.9$, $p < 0.001$). For all individuals, the mean $\delta^{15}\text{N}_{\text{Gly}}$ value for the bulk minimum periods were higher than for the bulk maximum but was only statistically higher for FW3 ($Z_8 = 2$, $p = 0.04$), although FW1 ($Z_8 = 1.9$, $p = 0.06$) was close to being significant at a threshold of $\alpha = 0.05$ (Fig. 4c). $\delta^{15}\text{N}_{\text{Asp}}$ values were higher for both the humpback and Minke than for the fin whales and for $\delta^{15}\text{N}_{\text{Ser}}$ the humpback had higher values than all the other whales (One way ANOVAs; Asp $F_{4,55} = 13.5$, $p < 0.001$; Ser $F_{4,55} = 14.6$, $p < 0.001$; Fig. 4a). $\delta^{15}\text{N}_{\text{Ala}}$ and $\delta^{15}\text{N}_{\text{Glu}}$ values for the trophic amino acids were generally higher for the humpback and Minke whales (one-way ANOVAs; Ala $F_{4,55} = 14.2$, $p < 0.001$; Glu $F_{4,55} = 14.1$, $p < 0.001$), but post hoc Tukey's indicated different relationships between individuals (Fig. 4 a&b) with FW3 being similar to the HB for Glu.

And we comment on the lack of fasting effects observed for other glycoenic AAs on LN 328: "We observed no fasting effects between minimum and maximum periods for baseline-corrected $\delta^{15}\text{N}_{\text{Ser}}$ or $\delta^{15}\text{N}_{\text{Asp}}$ (Fig. 4a) but $\delta^{15}\text{N}_{\text{Gly}}$ values were statistically higher in the minimum periods for bulk $\delta^{15}\text{N}$ values than in the maximum periods for bulk $\delta^{15}\text{N}$ in FW3 (Fig. 4c). However, all five individuals had higher averages for minimum periods ranging from 0.3 to 1.4‰ and the difference for FW2 was nearly statistically significant ($p = 0.06$)."

Figure 4: $\delta^{15}\text{N}$ for A) alanine, aspartic acid, and serine; B) glutamic acid and threonine; and C) glycine for the five mysticete individuals to assess trophic effects and possible starvation and fasting effects between individuals and baleen periods, respectively. All AAs have had been corrected against Phe to remove underlying source AA variation. Letters indicate significant differences as indicated by a post-hoc Tukey's test ($\alpha = 0.05$). For Gly, Wilcoxon ranked t-tests were used to compare between baleen regions for each individual. * indicates $p = 0.06$ and ** indicates $p < 0.05$.

34) Line 342: The patterns observed of decreasing Thr with increasing estimate TP, combined with no difference between minima and maxima (fasting states), suggest Thr is a better TL indicator than fasting indicator. As outlined by Lübcker et al. 2020, and mentioned here by the authors, there is not very strong biogeochemical justification, as of yet, for using Thr as a fasting indicator. So, I would suggest examining the other glycoenic AAs in this section and moving the topic of Thr to the *Trophic Levels* or *Resource Utilization* sections.

The reviewer's assessment is flawed in that she/he assumes that Thr behaves as other trophic AAs do within this data set. A direct comparison to Glu acid in Fig. 4b and to Ala in Fig 4a shows that much wider ranges of $\delta^{15}\text{N}$ values in Thr and suggests that another process is occurring within Thr that results in considerably larger fractionation differences and different relationships between individuals shown by Tukey's post hoc tests for Glu. The large ranges observed for Thr were 2-13x larger than for any of the other trophic or metabolic AAs examined in this study. We have therefore not moved our discussion on Thr as suggested due to the results now presented for other trophic AAs.

See panel B in the figure included in the previous comment and in the paper as Fig. 4 for the direct comparison between Glu and Thr.

We have also included statistical analysis for the trophic AAs Ala and Glu in LNs 238 to 257 reproduced in the previous comment.

We have now provided clarification that Thr behaves differently than Glu and the other metabolic AAs on LN 351: “The large ‰ ranges observed for Thr were 2 to 13× those observed for trophic (Glu, Ala) or other metabolic AAs (Gly, Ser, Asp) and Thr and Glu had different patterns of significant differences (Fig. 4b) indicating that Thr behaves differently than the ‘canonical’ trophic amino acid. These higher values observed for Thr are potentially the result of protein deficiency and may mark protein deficiency or starvation events across an individual’s lifetime.”

35) Line 3654: Lübcker et al. 2020 also observed significant changes in Phe during fasting, so catabolism of a whale’s own tissues during fasting/migration is another source of variability. Parsing the relative effect of this vs. incidental feeding on Phe would be difficult. This Phe fractionation is likely also important to consider given that the author normalize Gly and Thr to Phe.

We performed a two-way ANOVA on Phe and found that there were significant differences between min and max periods that were different than those found for Gly on LN 243: “The $\delta^{15}\text{N}_{\text{Phe-Base}}$ values for time intervals with minimum and maximum bulk $\delta^{15}\text{N}$ values were found to be significantly different between individuals and between minimum and maximum $\delta^{15}\text{N}$ value periods (two-way ANOVA: Individuals, $F_{4,54}=5.3$, $p=0.001$; min/max $F_{1,54}=11.4$, $p=0.001$; Interaction: $F_{5,54}=6.9$, $p<0.001$; Fig 3b). FW3 was found to have higher $\delta^{15}\text{N}_{\text{Phe-Base}}$ in the maximum periods for bulk $\delta^{15}\text{N}$ values than in the minimum periods (Wilcoxon ranked t-test, $Z_8=-2.5$, $p=0.008$). While for FW2 the $\delta^{15}\text{N}_{\text{Phe-Base}}$ values were also higher in the maximum bulk $\delta^{15}\text{N}$ period than in the minimum periods ($Z_5=-1.9$, $p=0.06$), but not statistically significant using a threshold of $\alpha=0.05$ (Fig 3b).”

We observed different patterns of difference for Phe and Gly within this data set and have assumed that the primary driver for these differences are underlying source values for Phe and fasting impacts from Gly. We now include discussion describing these differences and plainly state the assumptions that we are making in light of the different statistical results between the AAs.

LN 385 now reads: “Significant differences were observed between minimum and maximum regions for $\delta^{15}\text{N}_{\text{Phe-Base}}$ that did not align with those observed for $\delta^{15}\text{N}_{\text{Gly-Phe}}$ or $\delta^{15}\text{N}_{\text{Gly}}$ values indicate that Phe fractionated differently than Gly in this data set. Therefore, we have assumed that the fractionation associated with Phe is primarily due to changes in underlying biogeochemical values with minimal impacts from fasting, although other studies have observed considerable metabolic impacts {Lübcker, 2020 #769}.”

36) Line 401: That the $\delta^{15}\text{N}_{\text{base}}$ values do not reach those expected of the system could either mean the whale didn’t travel to where we think they should or that the calculation is incorrect. Again, two things difficult to parse. But this might be further justification for removing these corrections and focusing on differences between minima and maxima only.

Without applying the corrections for TDF, we rise even further away from the $\delta^{15}\text{N}$ values of -1 to 1‰ for the Mid-Atlantic isoscape and we would not account for the difference between tertiary consumers and primary resources (e.g. POM). We are addressing this by acknowledging that future work should attempt to further constrain the TDFs used for these calculations, but we do not think that analyzing unadjusted values is the clear path to go with this data set.

LN 404 now reads: “Alternatively, higher than expected $\delta^{15}\text{N}_{\text{Base}}$ values may reflect poorly constrained TDFs and should be considered in future work aiming to anchor $\delta^{15}\text{N}_{\text{Base}}$ values for marine mammals to biogeochemical isoscapes.”

37) Line 445 add “relative to the other whales examined” so the reader knows you are comparing to

the other species. Otherwise, this could get confused with the use of Gly to examine fasting.
Corrected as indicated.

38) Line 423: Did body size estimates or any other ecological information accompany these samples to make an educated guess on maturity? Identifying reproductive rates and strategies would be pertinent as well. i.e., are females always either pregnant or lactating and thus under some constant physiological burden? Table 1: This answers my previous statement. I encourage the authors to mention some of this pertinent specimen information at the beginning of the methods.
Corrected as indicated.

LN 104 now reads: “Details about sex and estimates of animal’s maturity are located in Table 1.”

Reviewer 2

Riekenberg et al. present a bulk tissue (d13C and d15N) and amino acid (d15N) isotope dataset to generate movement, foraging, and physiological information for limited number (n=1-2) of serially sub-sampled baleen plates from 3 whale species. The isotopic patterns are compelling, but the paper does not do a good job of exploring (or constraining) alternative explanations for the data presented. This issue is most obvious in the Discussion, where several sections (e.g., Metabolism of Whales and Resource Utilization) do not do into great depth in providing explanations for the patterns observed, especially those potentially driven by physiological or biochemical mechanisms.

1) For example, the authors should lean on a rich literature of isotope turnover data derived from experimental dietary manipulations. Because isotopic incorporation is influenced by organism size, scaling turnover data from smaller mammals may not be possible, but at least patterns from experiments could inform and potentially constrain explanations provided by the authors regarding isotopic (C versus N) turnover.

We address the reviewer’s concerns about the dynamics of turnover of C and N in large mammals in comments 17 and 18 where we address the relevant studies that were originally cited in the manuscript with excerpts from those references demonstrating their relevance.

2) Furthermore, the assessment of trophic level based on AA d15N values is too complicated, difficult to follow, and fraught with assumptions regarding trophic-source AA TDF values. The scaling exercise used to estimate TDFs based on differences between expected and observed bulk tissue trophic discrimination is unprecedented and problematic.

We have discussed reviewer 2’s concerns about trophic level calculations in discussion of their comment 8 below, and further discussion on the choice of equation and the included variables is included in the response to Reviewer 1’s comment 23.

3) Lastly, the paper is difficult to follow and could benefit from careful editing to improve clarity. There are several examples of sentences being too long (e.g., Lines 30-33) and/or have unnecessarily complicated structure (e.g., Lines 37-40).

Corrected as indicated.

Introduction

4) Lines 52-53. The latter half of this sentence is a bit ambiguous in regards to “mixture of influencing factors and often provide results that are inconclusive or muddled”, perhaps be more specific about these caveats so readers better understand motivations.

The following sentence (LN 53) directly describes the two likely scenarios that could muddle results from bulk isotope mixing studies along with references for each situation: “This is especially true in

systems where the isotopic baseline supporting production changes or the trophic level that animals feed at have shifted (29, 30).”

No changes resulted from this comment.

5) Lines 71-75. Seems like this statement requires a citation, perhaps O’Connell 2017?

Corrected as indicated.

Methods

6) Amino Acid Measurement (Lines 132-140). I realize that some information may be in Reference 47, but readers shouldn’t have to look up another paper for critical methodological information that will aid future studies. Some important details are missing from this section including where the measurements were made and what reference materials (standards) were used to assess analytical precision, and how were those reference materials prepared and analyzed in relation to unknown samples, and how were raw measurements corrected/reduced with data from reference materials? Were samples derivatized alongside unknown samples in batches and run accordingly? Like other types of isotopic measurements at the bulk tissue level (e.g., d2H analysis), these kinds of measurements are complicated and so analytical protocols need to be clear for readers to push the science forward.

Reference 47 is Riekenberg PM, van der Meer M, Schouten S. Practical considerations for improved reliability and precision during determination of $\delta^{15}\text{N}$ values in amino acids using a single combined oxidation-reduction reactor. *Rapid Communications in Mass Spectrometry*. 2020.

In this reference, we explicitly lay out the improved oxidation settings and applied corrections from Yarnes and Herzage (2017) that allowed for a precision of <0.5‰ for samples and standards on this data set run with that method using the same analysis. I agree with the reviewer that these analytical methods and procedures are complicated and need to have the necessary level of detail provided in order to replicate them. Providing an increased level of detail is why we published a separate paper in an analytically focused specialist journal that goes into extensive detail about this method in 12 pages rather than the 150-200 words that is appropriate in the method section of a related article.

We now indicate where the analysis took place. LN 133 now reads: “In short, at the Royal Netherlands Institute for Sea Research (NIOZ) samples were hydrolyzed, derivatized into N-pivaloyl/isopropyl (NPIP) derivatives and analyzed in duplicate with a Trace 1310 gas chromatograph coupled to a Delta V Advantage isotope ratio mass spectrometer (Thermo Scientific, Bremen) via an IsoLink II and Conflo IV. Details about the temperature ramp, program settings, and normalization procedures are provided in (45).”

7) Trophic Level (Lines 163-195). I’m reasonably familiar with trophic level calculations based on amino acid d15N data, however, I found the section in the Methods on this topic to be very confusing and full of assumptions. For example, why use a beta value for the Glu-Phe trophic-source pair when Equation 2 utilizes weighted mean values for grouped trophic and source amino acids? *This comment is addressed in Reviewer 1’s comment 23 where we go into extensive detail about both the equation that we used for trophic level and the values we used for the variables that are included in that calculation.*

8) Furthermore, why was bulk tissue discrimination (1.8 per mil, Busquets Vass et al. 2019) used to derive a trophic-source TDF for grouped trophic and source amino acids? I’ve never seen this kind of sliding scale approach to estimate AA specific TDFs? Furthermore, why assume that whales discriminate amino acids similar to invertebrates and use a 7.6 per mil TDF as an end-member in the scaling approach. There are data for AA-specific TDFs for marine mammals (see review by McMahon and McCarthy 2016) which show that TDFs for the canonical Glu-Phe pair is lower (5-6 per mil) than

reported for invertebrates and fish, but also note that estimates vary among fish species with different feeding modes and the overall average is still significantly smaller than 7.6 per mil (Bradley et al. 2015).

We used the value of 7.6‰ as a TDF as it has often been applied as a ‘canonical’ TDF for consumer TDFs and has been cited as such in Chikaraishi et al. (2009), McCarthy et al. (2016), and Ohkouchi et al. (2017), amongst others. By using Σ trophicAA and Σ sourceAA values along with scaled values from bulk estimates, we were attempting to minimize uncertainty from using single AA calculations while utilizing a compressed TDF (4‰) and the most often quoted δ value of 3.4‰. As the reviewer states, this allows us to apply a lower TDF value of 4‰ calculated from the difference between bulk (3.4‰) and amino acid (7.6‰) TDFs. Nowhere is there mention of applying invertebrate or fish AA TDFs to whales and the calculated AA TDF of 4‰ reflects the compressed nature of TDFs in higher trophic level marine mammal consumers as the reviewer states.

We have now added values to allow for comparison to the TDFs for AAs presented in McMahon and McCarthy (2016) on LN 174: “The calculated TDF (4.0‰) is similar to those presented for amino acids in marine mammals (5-6‰) in McMahon and McCarthy (2016).”

9) Lastly, there are no error bars on any of the parameters in Equations 2-5, and as such error is not properly propagated to estimates of trophic level or the subsequent baseline estimates for carbon and nitrogen isotope values. Overall, the myriad assumptions made and lack of error in the calculation of trophic and dependent estimates of baseline isotopic composition is a major flaw with the current version of the manuscript.

We have provided the propagated error estimate for trophic level in the original manuscript from the equation presented in Ohkouchi et al 2017 on LN 175: “Error propagation for trophic level gives a standard deviation of $\pm 0.8‰$ using the equation presented in Ohkouchi, Chikaraishi (39).”

We have included the errors necessary to calculate and the propagated error for $\delta^{15}N_{Phe-Base}$ values on LN 180-182: “... where $0.4 \pm 0.5‰$ is the small enrichment observed for Phe during metabolism (40) and trophic level calculated for each individual (Table 1) following the method presented in Vokhshoori, McCarthy (47). Error propagation indicated a standard deviation of $1‰$.”

We have included the errors necessary to calculate and the propagated errors for $\delta^{13}C_{Base}$ and $\delta^{15}N_{Base}$ on LNs 187-191: “... where $\delta^{13}C_{Bulk}$ and $\delta^{15}N_{Bulk}$ represent the C and N isotopic composition of bulk material, $2.3‰$ and $2.8‰$ represent the offset between diet and baleen for carbon and nitrogen (44), $0.5 \pm 0.3‰$ and $2.2 \pm 0.3‰$ represent the offsets for trophic enrichment for carbon and nitrogen for the trophic levels supporting the whales prey {McCutchan, 2003 #590; Connell, 2010 #184}, and trophic level is the average estimate for each individual (Table 1). Error propagation indicated a standard deviation for $\delta^{13}C_{Base}$ and $\delta^{15}N_{Base}$ of 0.8 and $0.9‰$, respectively.”

10) Lines 182-184. Here and elsewhere, isotope data should be reported to one decimal point because that is the degree of analytical precision on bulk tissue and amino acid isotope measurements.

Corrected as indicated throughout the manuscript.

Discussion

11) Lines 278-279. What is meant by “differences in excretion of $\delta^{15}N$ ”? Animals don’t excrete isotope values, they excrete isotopes. Perhaps provide a bit more descriptive information about the mechanism(s) by why animals discriminate isotopes when assimilated resources and synthesizing tissues.

We have clarified this statement to highlight that the changes in bulk $\delta^{15}\text{N}$ values result from the combined effects of excretion, metabolism, and underlying biogeochemical source values of N across migrations.

LN 279 now reads: "The marked oscillations in $\delta^{15}\text{N}$ values of baleen are assumed to reflect residence times in Mid-Atlantic breeding grounds (minima) and high-latitude feeding grounds (maxima) with substantial differences in the underlying $\delta^{15}\text{N}$ values for POM in these regions (-1 to 1‰ versus 6-10‰, respectively) combined with the impacts from metabolism and excretion of N on $\delta^{15}\text{N}$ values depending on fasting status during migrations (48, 49)."

12) Lines 313-316. Is a TL of 2.7 realistic for a krill specialist? Assuming primary producers (phytoplankton) are TL 1, are krill strict herbivores with a TL of <2? I believe they are more indiscriminant omnivorous filter feeders with a TL of somewhere between 2 and 2.5, which would put a krill specialist like a fin or blue whale at TL 3 or slightly higher. The whole point of this is that the TL estimates, specifically that TDF used in Equation 2 is not very well constrained. Instead of trying to estimate TL, the authors should consider using their data to constrain a TDF value for whales by mining the literature (e.g., reference 63) for estimates of TL for these species and then back-calculating a TDF with Equation 2. Or perhaps using a Glu-Phe specific equation (e.g., McMahon and McCarthy 2016) to calculate a TDF_{Glu-Phe} that would be extremely useful for future studies that use amino acid nitrogen isotope data to study whale ecology. In general, beta values for phytoplankton-based pelagic food webs are better constrained than trophic-source TDF values (even for the canonical pair Glu-Phe), so using published literature to constrain the latter is a really good use of the data that will benefit future studies.

We feel that defining the differences between TDFs for individual species is outside the scope of this study. This study is trying to identify differences for trophic level, metabolism, and source values across the lifetime of the whales utilizing the continuous record of baleen. We have chosen to use a relatively small AA TDF of 4‰ for all species examined largely due to the poor replication across individuals in this study and the uncertainties surrounding marine mammal TDFs.

Currently, Matthews & Ruiz-Cooley 2020 (reference 63) is the best between species comparison for AA defined TLs and clearly highlights the discrepancy between AA TP estimates and stomach content studies using vastly more replication than we have within this study. Incidentally, our TL estimate of 2.7 for fin whales is higher than the corresponding estimate for Bowhead whales (2.0) using a similar equation in Matthews et al (2020) and is closer to the stomach content estimate for Bowhead whales (3.2) than the Matthews et al. estimate. This comparison is relevant due to both species being zooplanktivorous.

*We have clarified that our results yielded similar outcomes as those applied in Matthews et al (2020) on LN 312: "A TDF of 4‰ yielded comparable trophic levels for zooplanktivorous whales (Bowhead whales, *Balaena mysticetes* TL of 1.9-3.0) as determined through amino acid analysis and stomach contents (TL 3) in Matthews, Ruiz-Cooley (63)."*

13) Line 322-324. It must be more than lipid catabolism that drives variation in amino acid $\delta^{15}\text{N}$ during fasting. Fix to better tie with changes in nitrogen balance.

We have clarified that catabolism of tissue protein leads to a negative nitrogen balance in addition to utilization of lipid stores.

LN 322-324 reads: "The amino acids Gly and Thr have been found to respond to fasting and protein deficiency in mammals through variable enrichments in $\delta^{15}\text{N}$ values for each AA (53, 54) as catabolism of tissue protein occurs leading to a negative nitrogen balance coinciding with metabolism of stored lipid resources."

14) Line 324. Do you mean glucogenic (not glycogenic) amino acids here that are catabolized during gluconeogenesis (reference 63)? If so, the list is much longer than the 4 AAs listed so why were

enrichments only observed in Gly? At the very least I would expect Ser to also show enrichments because Gly is synthesized from Ser (via one step) and both of these (simple) AAs share the same nitrogen atom (no amination occurs during that step). Were any fasting-related patterns observed in Phe, which is another glucogenic AA?

We found that Ser had no significant differences for minimum and maximum periods in the baleen using a two-way ANOVA, which differed from Gly which had differences (Fig. 4 A&C, reproduced below).

We have now included analysis of Ser on LN 264: “ $\delta^{15}\text{N}_{\text{Asp}}$ values were higher for both the humpback and Minke than for the fin whales and for $\delta^{15}\text{N}_{\text{Ser}}$ the humpback had higher values than all the other whales (One way ANOVAs; Asp $F_{4,55}=13.5$, $p<0.001$; Ser $F_{4,55}=14.6$, $p<0.001$; Fig. 4a).”

We also found different patterns of significant differences between Gly and Phe, which is now mentioned on LN 331: “Furthermore, patterns of significant differences between individuals for Gly and Phe did not align, likely reflecting different processes affecting the metabolic and source amino acids.”

Furthermore, we acknowledge that differences in Phe have been observed with fasting impacts before and more clearly state our assumptions as a result of the statistical analyses in this study on LN 385: “Significant differences were observed between minimum and maximum regions for $\delta^{15}\text{N}_{\text{Phe-Base}}$ that did not align with those observed for $\delta^{15}\text{N}_{\text{Gly-Phe}}$ or $\delta^{15}\text{N}_{\text{Gly}}$ values indicate that Phe fractionated differently than Gly in this data set. Therefore we have assumed that the fractionation associated with Phe is primarily due to changes in underlying biogeochemical values with minimal impacts from fasting, although other studies have observed considerable metabolic impacts (53).”

Figure 4: $\delta^{15}\text{N}$ for A) alanine, aspartic acid, and serine; B) glutamic acid and threonine; and C) glycine for the five mysticete individuals to assess trophic effects and possible starvation and fasting effects between individuals and baleen periods, respectively. All AAs have had been corrected against Phe to remove underlying source AA variation. Letters indicate significant differences as indicated by a post-hoc Tukey’s test ($\alpha=0.05$). For Gly, Wilcoxon ranked t-tests were used to compare between baleen regions for each individual. * indicates $p = 0.06$ and ** indicates $p < 0.05$.

A**B****C**
15) Lines 344-347. Why would protein catabolism lead to increases in Thr values when the general pattern is that Thr values decrease with increasing trophic level. In other words, if starvation induced catabolism has the same isotopic effect as an increase in trophic level (as the organism consumes itself), then why would Thr values get higher and not lower? I believe the work on fasting elephant seals clearly shows that Thr $\delta^{15}\text{N}$ values get lower (not higher) during nutritional stress that yields protein catabolism. Given the strange pattern reported here, the fact that (1) high Thr values were observed “intermittently” and were not associated with periods of the annual life cycle when fasting may occur (e.g., migration), and (2) Thr is a difficult AA to measure because it typically elutes close to other AAs, I suggest that authors revisit these outlier values and if they are sound please provide a possible mechanism for Thr enrichment.

Thr typically elutes close to Asp when using a DB-5 column and the NPip derivitization procedure. Riekenberg et al 2020 (Reference 47) demonstrates that using a 60 m column allows for clear separation of both peaks with no evidence of carryover despite marginal co-elution (<20%) if concentrations are allowed to be too high. Through the combination of targeted loading rates and routine removal of the dirtied precolumn/retention gap between sequences to optimize chromatography, it is possible to routinely examine Thr values. This is another reason that we published a dedicated paper to highlight the tricks to optimizing this analysis rather than relying on 150-200 word descriptions in associated works to adequately describe this method.

The mechanism that has been proposed for Thr fractionation is a strong reverse fractionation Fuller and Petzke (2017) RCMS 31:8 (reference 45 in this MS). They state: “We hypothesize that the inverse nitrogen equilibrium isotope effects of Schiff base formation, between AAs and pyridoxal-5’-phosphate cofactor enzymes, play a key role in the bioaccumulation and depletion of ^{15}N in the biomolecules of living organisms and contributes to the variability in the nitrogen trophic level effect.” Additionally, they found evidence for decreased fractionation with increased protein deficiency in Fig 1B with a 5‰ increase between mice fed with high protein and those fed with adequate protein.

The work presented in Lubcker et al. 2019 (Reference 46) indeed found that elephant seals did not show a decreased fractionation with fasting, although they did see a considerable effect on Gly. This is why we state that our findings conflict with those in Ref 44 on LN 356: “This finding conflicts with decreased $\delta^{15}\text{N}_{\text{Thr}}$ values observed in elephant seal whiskers during fasting (44), suggesting that there may be multiple effects impacting the metabolism of Thr during fasts that are more or less severe and warrant further investigation.”

Given that Fuller and Petzke (2017) clearly demonstrate the patterns described in this manuscript and describe the mechanism for Thr depletion and are cited as such on LN 344: “Higher $\delta^{15}\text{N}_{\text{Thr}}$ values in mammals have been observed to coincide with reduced protein quality in their diet causing reduced reverse fractionation with higher $\delta^{15}\text{N}_{\text{Thr}}$ values indicating periods of potential starvation (43) although this mechanism is incompletely characterized.”

No changes have resulted from this comment.

16) Line 377. I agree that incorporation must play a big role in the dampened patterns in baleen relative to the end-member baseline values associated with the southern breeding and northern foraging grounds. I think this section could be strengthened by directly comparing the bulk baleen data to published isoscapes for the region to assess whether the whales analyzed here ever equilibrate with the end-members? This also has implications for how AA isotope patterns are interpreted because individual AAs likely have variable isotopic incorporation rates, and thus those that turnover more quickly might be more faithful recorders of local isotopic conditions than ones that turnover more slowly.

The end member values being compared against are presented in the original manuscript on LN 373 to allow for the type of direct comparison that the reviewer is requesting: “Isoscapes, i.e. geographic maps of the underlying yearly averages for regional isotopic values of carbon and nitrogen (35, 36), characterize isotopic ranges for the POM sampled from both the mid-Atlantic ($\delta^{13}\text{C}$ -28 to -30‰; $\delta^{15}\text{N}$ -1 to 1‰) and high-latitude Arctic ($\delta^{13}\text{C}$ -24 to -20‰; $\delta^{15}\text{N}$ 6 to 10‰).”

We have clarified that baleen values are unlikely to reflect the end member baseline values on LN 379: “The variations in those values are likely to be dampened and never reflect the end member values from the underlying isoscape depending on the relative feeding intensity (opportunistic feeding in transit), variations in seasonal values for underlying biogeochemical processes, and the relative turnover of the internal source AA pool during migration and breeding.”

17) Lines 390-392. I’m curious if there is any experimental data in other mammals to show that nitrogen actually turns over faster than carbon? The Trueman et al. 2019 paper (which is not cited correctly) is suggestive of differences in turnover rates for carbon versus nitrogen, but it presents data from a single animal from a single species. There is a very rich literature on C and N isotope turnover rates for smaller mammals that the authors could lean on here to better strength these claims, as there are biochemical conditions that could promote greater turnover in C versus N isotope turnover and vice versa.

We have corrected the Trueman et al. (2019) as reference 59.

What is relevant here is the decoupling that can occur between C and N due to relative density of C in lipids. This has been observed previously and is discussed in the two references in this section. Cherry et al. (2011) (reference 71) and Newsome et al. (2014) (reference 72) are both excellent resources for looking into routing effects on C/N dynamics for bulk mixing in large mammals and how routing impacts amino acids in rats, respectively. Both of these references were originally included in the section immediately prior to the sentence the reviewer referenced for this comment.

No changes have occurred as a result of this comment, but further discussion and changes in this paragraph do result from the following comment that continues the discussion of routing in marine mammals.

18) Lines 396-398. Yes, but in order for lipid carbon to be deposited in baleen, it has to be used to synthesize non-essential amino acids, and in order to do that it requires nitrogen from a central N pool. So its not clear to me why this would promote faster turnover in C in comparison to N? Also keep in mind that ~1/3 of the amino acids in baleen are essential, which can’t be synthesized de novo and must be sourced from exogenous (diet) or endogenous (tissue catabolism or gut microbes) sources; as such, carbon and nitrogen isotope turnover is probably similar for this essential pool of AAs. I also don’t understand why lower concentration of nitrogen in prey would contribute to lower turnover because N concentrations of whale tissues (e.g., baleen) mirror that of their diet; the easily assimilate portions of krill or fish (e.g., muscle or internal organs) have the same C/N ratios as whale baleen (or muscle). So for a carnivore like a mysticete whale, I doubt that variation in the C versus N concentrations of food sources has any impact on consumer tissue turnover. Again, are there any experimental studies the authors can rely on here to support these claims?

The reviewer is most likely correct about similar turnover in essential AAs for C and N, but the highlighted statement is examining baseline bulk $\delta^{13}\text{C}$ values and says nothing about essential amino acids. We state in the original manuscript that the wider range of $\delta^{15}\text{N}$ values is like due to “...increased turnover in N when compared to C” which does not reflect the reviewer’s question to us about “So its not clear to me why this would promote faster turnover in C in comparison to N?”. The metabolic N pool is considerably smaller than the C pool contained in lipid stores in mysticete whales, so it’s quite feasible that a negative N balance will disproportionately affect the smaller pool of

material resulting in larger fractionations apparent within the pool of N. As to the reviewer's doubt that C/N concentrations having no impacts on consumer tissue turnover, Newsome et al. (2014) has this to say: "The large degree of variation in $\delta^{13}\text{C}$ values that we observed (1) in the major non-essential amino-acid components of mouse muscle and (2) in the calculated trophic-discrimination factors among diettreatments shows that the extraction of dietary lipids could influence interpretations gleaned from stable-isotope data in studies of omnivores and carnivores. Several studies provide empirical evidence that this pathway may be important for mammalian consumers, especially carnivores that frequently consume lipid-rich diets. ...The utilization of lipid-derived carbon to synthesize non-essential amino acids may be especially pertinent for marine animals that consistently consume lipid-rich prey (Rosen and Trites 2000; Copeman and Laurel 2010; Cherry et al. 2011; Whiteman et al. 2012; Parrish 2013). For example, many species of marine forage fish--herring, capelin, sand lance, and eulachon--that provide an abundant and energy-dense resource for marine mammals, seabirds, and fish in many coastal and pelagic ecosystems may contain 25-50% lipids (Hislop et al. 1991; Lawson et al. 1998; Robards et al. 1999; Anthony et al. 2000; Wanless et al. 2005)."

As for the reviewer's request for relevant studies about routing of C, Cherry et al. (2011) (reference 71) and Newsome et al. (2014) (reference 72) are both excellent resources for looking into routing effects on C/N dynamics for bulk mixing in large mammals and how routing impacts amino acids in rats, respectively. Both of these references were originally included in the section immediately prior to the sentence the reviewer referenced for this comment.

We have edited to clarify that we are referring to predation on lipid rich prey and the disproportionate impacts to bulk $\delta^{15}\text{N}_{\text{Base}}$ values due to fractionation caused by small pool size and not amino acid values in this section. LN 390-401 now reads: " The larger ranges observed for $\delta^{15}\text{N}_{\text{Phe}}$ and $\delta^{15}\text{N}_{\text{Base}}$ (8.1 and 6.6, respectively) versus $\delta^{13}\text{C}_{\text{Base}}$ (2.3‰) are likely due to increased turnover and fractionation occurring in N when compared to C within tissues as incidental feeding occurs (57). N from incidental feeding is likely to be directly metabolized into animal tissues, while carbon from lipid rich prey can be metabolically routed to either direct incorporation to tissue or to storage within large lipid stores depending on feeding status (71). Under fasting conditions, lipid stores will be utilized as a source of C with a light $\delta^{13}\text{C}$ value that reflects the fractionation of C from food resources containing the regional $\delta^{15}\text{N}$ values where they were consumed (72). These lipid stores are expected to be mobilized during times of limited feeding and reduce the impact of C derived from incidental feeding on $\delta^{13}\text{C}_{\text{Base}}$ values of the baleen during fasting conditions as metabolites from blood are incorporated into baleen. Metabolic routing of C could contribute to dampening variation in C values along the baleen, especially if incidental feeding lipid-rich prey such as sprat or herring is occurring during fasting periods (71)."

19) Lines 413-415. Why is this increase related to trophic level, couldn't it also be related to a change in foraging location? What do the AA $\delta^{15}\text{N}$ data say about this trend?

A change in foraging location would be expected to coincide with changed Phe_{Base} values for nitrogen, which does not occur (Fig. 3B). We have now included a statistical comparison between these two periods for Phe_{Base} , that indicated no change in underlying resource values. This leaves an upward trophic shift that is not large enough to qualify as a changed trophic level as the evident solution.

LN 415 now reads: "This difference did not result from a shift in regional sources as Phe_{Base} was not significantly different between these samplings ($W_6=1.3$, $p=0.2$)."

20) Lines 419-421. This interpretation of $\delta^{15}\text{N}$ Gly assumes that Gly is largely routed directly from diet, however, Gly is the simplest amino acid and as such is very easy to synthesize de novo. Couldn't the enriched Gly $\delta^{15}\text{N}$ values be driven by other intrinsic (physiological) or extrinsic (ecological)

factors? I suggest other potential explanations are evaluated here. For example, Gly $\delta^{15}\text{N}$ has also been shown to be very sensitive to fasting in marine mammals.

We examined Gly $\delta^{15}\text{N}$ for this humpback whale and observed no differences between minimum and maximum periods that indicates a considerable fasting effect in this individual across the baleen record. Gly $\delta^{15}\text{N}$ values were consistently higher with a fairly small range, and as stated in the previous comment, no baseline shift was evident for Phe_{Base} during this time period. These separate indicators build up to an indication of predation on higher trophic level prey in similar environments across time.

LN 415 now reads: “The combination of 1) no baseline shift for Phe_{Base} and 2) no difference for $\delta^{15}\text{N}_{\text{Gly}}$ values between minimum and maximum periods indicating fasting effects, support the hypothesis of higher trophic level predation within the same environment (e.g. coastal margins) across time. High $\delta^{15}\text{N}_{\text{Gly}}$ values observed for the humpback (~0‰; Fig. 4b) also mirror $\delta^{15}\text{N}_{\text{Gly}}$ values found in small fish migrating from estuarine into coastal waters where elevated $\delta^{15}\text{N}_{\text{Gly}}$ values have been previously observed such as herring (43).”

Appendix C

Review for RSOS-210949

Reconstructing the diet, trophic level, and migration pattern of Mysticete whales based on baleen isotopic composition

Summary

I believe the authors adequately addressed most of the reviewers' original concerns with the manuscript. The additional ecological context and detailed project objectives in the introduction are helpful, as are the greater clarification of error propagation and analysis of the Ala, Ser, Glu, and Asp data.

However, the authors' justifications for their β and TDF values are fundamentally flawed and call into question the validity of the TP estimates from this study. My issue is not with the TP, β , nor TDF estimates themselves, which by and large seem reasonable for the given species and systems, but with the path that led to them. Recalculated β and TDFs may very well end up close to the 3.4‰ and 4.0‰ values used by the authors, but how they are derived matters and the authors' current approach does not match how these are derived in the CSIA literature.

The authors use Chikaraishi et al. 2009 and Matthews et al. 2020 as justification for not recalculating β and TDFs when averaging AA $\delta^{15}\text{N}$ values within the TP equation, asserting "*The necessity of recalculating β indicated by reviewer 1 is not reflected in the current literature: Chikaraishi et al. 2009 presents a calculation that includes the use of weighted averages from trophic and source AAs along with a TDF of 7.6‰ and the typical β of 3.4*". This is simply not true. Chikaraishi et al. (2009) did not present equations to calculate TPs that use weighted averages of AAs. They presented alternate TP_{CSIA} equations for individual combinations of trophic AAs and Phe, not weighted averages, and in fact advance the need to use different β and TDF values for each combination of trophic and source AA (Equations 3-7). So, it is unclear to me why this source is used as justification for not using weighted β and TDF values. It is similarly unclear why Matthews et al. (2020) cited Chikaraishi et al. (2007, 2009) at all for their weighted mean approach, as neither of these sources discuss the issue of weighted means. Matthews et al. (2020) either forgot to include the clarifier "modified from" or cited the wrong literature. It was McCarthy et al. (2007) that first proposed using weighted means of trophic and source AAs in TP_{CSIA} equations, but it was not until Vander Zanden et al. (2013) that this framework was extended to the canonical TP_{CSIA} equation presented by Chikaraishi et al. (2009). Moreover, Matthews et al. (2020) tested four TP_{CSIA} equations and found that the multi-TDF equations presented by Germain et al. (2013) or McMahan et al. (2015) produced bowhead whale—the only mysticete sampled— TP_{CSIA} estimates (2.4–3.0) closest to TP calculated from stomach contents (3.2). In fact, the weighted averages equation the authors use in the present study produced the most inaccurate TP_{CSIA} estimates (1.9–2.0). There is thus no scientific justification for using weighted averages of trophic and sources AA $\delta^{15}\text{N}$ in combination with Glu-Phe β and TDF values within Chikaraishi et al. 2009 and Matthews et al. 2020.

Moreover, a review of the CSIA literature makes clear that when using weighted averages of trophic and sources AA $\delta^{15}\text{N}$ the convention within the field is to recalculate β and TDF values as was suggested by both reviewers (see the following 18 studies Vander Zanden et al. 2013,

Downs et al. 2014, Choy et al. 2015, Bradley et al. 2015, Geringer et al. 2017, Williams et al. 2017, Houssard et al. 2017, Décima et al. 2017, Peavey et al. 2017, Gagne et al. 2018a b, Vane et al. 2018, Sahm et al. 2020, Glass et al. 2020, Ledesma et al. 2020, Lemons et al. 2020, Tokuda et al. 2020, Richards et al. 2020). The only cases where this is not the norm is when β and TDF are removed from the equation entirely and a relative TP is calculated using weighed means of trophic and source AAs only (e.g., McCarthy et al. 2007, Sherwood et al. 2011, Batista et al. 2014, Sackett et al. 2015, Bradley et al. 2016). As a result, this manuscripts combination of weighted consumer AA values and Glu-Phe β values and TDFs is unprecedented in the CSIA literature. At a fundamental level, if the authors believe that the weighted means approach improves TP_{CSIA} estimation then it is illogical not to extend that approach to all parameters in the TP_{CSIA} equation. Otherwise, one is not comparing like to like and not accurately propagating error.

The authors make a valid point that primary producer $\delta^{15}N$ Lysine data are underrepresented in the CSIA literature. However, McClelland & Montoya (2002), McClelland et al. (2003), McCarthy et al. (2007, 2013), and Gutiérrez-Rodríguez et al. (2014) collectively provide sufficient data from which all necessary β values for this manuscript could be calculated. Indeed, Table 4 in Bradley et al. (2015) distills much of this information into such β values (including XX-Lys). These data are hardly speculative and have been used in a number of subsequent TP_{CSIA} studies (included in list above).

Lastly, the sliding scale approach for estimating a TDF of 4.0‰ is similarly unprecedented in the CSIA literature. I am aware of no CSIA study that has used this approach. The convention in this scenario would be to use a TDF from another marine mammal or urea producing secondary consumer. The lack of species-specific AA $\delta^{15}N$ TDFs is a common issue for all marine megafauna. Again, I believe 4.0‰ is likely a reasonable TDF for this manuscript, but the path to that value matters. At minimum, the authors need to further detail their sliding scale approach as I still do not understand how they arrived at 4.0‰. However, issues outlined above with only using Glu-Phe aside, using a scaling factor of 7.6‰ here for AAs is problematic. This value is indeed considered a ‘canonical’ TDF, but a rich body of literature clearly demonstrates that in marine systems it is only accurate for primary and secondary ammonia-producing consumers (e.g., invertebrate, zooplankton, planktivorous fish). As reviewed by Nielsen et al. (2015), Bradley et al. (2015), and McMahon and McCarthy (2016), higher order consumers (secondary+), particularly urea-producing consumers, typically have much lower TDFs, which the authors also note in this manuscript. As a result, if the authors are keen on using a sliding scale approach, they should instead use a lower scaling factor for AAs. A value of 7.6‰ is especially inappropriate to use for humpback and minke whales given that they feed on fish.

As I see it the authors have 3 options for estimating TP_{CSIA} estimates moving forward that are supported by the prevailing CSIA literature:

1. Use the published literature to re-calculate weighted β and TDF values as originally suggested by the two reviewers.
2. Abandon the weighted means approach and use the Germain et al. (2013) or McMahon et al. (2015) multi-TDF equations in combination with a single trophic and single source AA, the latter of which is the most commonly applied in the literature.

3. Calculate relative TP_{CSIA} estimates using weighted averages of trophic and source AAs only as outlined by McCarthy et al. (2007) (and others).

I do not believe the results of this study will fundamentally change as a result of these recalculations, but they will yield more defensible TP_{CSIA} estimates. This is critically important given the “proof of concept” nature of this study and the fact that it presents some of the first TP_{CSIA} estimates for mysticetes, which will inevitably be cited in many future studies.

Specific comments

Line 4: add “values” after $\delta^{15}\text{N}$

Line 82: clarify $7.6\pm 1.5\%$ is an example ...TDF (*e.g.*, $7.6\pm 1.5\%$). Remove extra space after the last parenthesis.

Line 85: Remove “Additionally,”

Line 314: I recommend removing the phrase “compared to the typically applied value of 7.6‰.” As written, this sentence could be taken to mean that 7.6‰ is typical for whales, which is certainly not the case.

References

- Batista FC, Ravelo AC, Crusius J, Casso MA, McCarthy MD (2014) Compound specific amino acid $\delta^{15}\text{N}$ in marine sediments: A new approach for studies of the marine nitrogen cycle. *Geochim Cosmochim Acta* 142:553–569.
- Bradley C, Longenecker K, Pyle R, Popp B (2016) Compound-specific isotopic analysis of amino acids reveals dietary changes in mesophotic coral-reef fish. *Mar Ecol Prog Ser* 558:65–79.
- Bradley CJ, Wallsgrave NJ, Choy CA, Drazen JC, Hetherington ED, Hoen DK, Popp BN (2015) Trophic position estimates of marine teleosts using amino acid compound specific isotopic analysis: Stable isotope-derived trophic positions of teleosts. *Limnol Oceanogr Methods* 13:476–493.
- Choy CA, Popp BN, Hannides CCS, Drazen JC (2015) Trophic structure and food resources of epipelagic and mesopelagic fishes in the North Pacific Subtropical Gyre ecosystem inferred from nitrogen isotopic compositions: Trophic structure of pelagic fishes. *Limnol Oceanogr* 60:1156–1171.
- Décima M, Landry MR, Bradley CJ, Fogel ML (2017) Alanine $\delta^{15}\text{N}$ trophic fractionation in heterotrophic protists. *Limnol Oceanogr* 62:2308–2322.
- Downs E, Popp B, Holl C (2014) Nitrogen isotope fractionation and amino acid turnover rates in the Pacific white shrimp *Litopenaeus vannamei*. *Mar Ecol Prog Ser* 516:239–250.
- Gagne TO, Hyrenbach KD, Hagemann ME, Van Houtan KS (2018a) Trophic signatures of seabirds suggest shifts in oceanic ecosystems. *Sci Adv* 4:eao3946.
- Gagne TO, Hyrenbach KD, Hagemann ME, Van Houtan KS (2018b) Trophic signatures of seabirds suggest shifts in oceanic ecosystems. *Sci Adv* 4:eao3946.
- Gerringer ME, Popp BN, Linley TD, Jamieson AJ, Drazen JC (2017) Comparative feeding ecology of abyssal and hadal fishes through stomach content and amino acid isotope analysis. *Deep Sea Res Part Oceanogr Res Pap* 121:110–120.

- Glass J, Daly R, Cowley P, Post D (2020) Spatial trophic variability of a coastal apex predator, the giant trevally *Caranx ignobilis*, in the western Indian Ocean. *Mar Ecol Prog Ser* 641:195–208.
- Gutiérrez-Rodríguez A, Décima M, Popp BN, Landry MR (2014) Isotopic invisibility of protozoan trophic steps in marine food webs. *Limnol Oceanogr* 59:1590–1598.
- Houssard P, Lorrain A, Tremblay-Boyer L, Allain V, Graham BS, Menkes CE, Pethybridge H, Couturier LIE, Point D, Leroy B, Receveur A, Hunt BPV, Vourey E, Bonnet S, Rodier M, Raimbault P, Feunteun E, Kuhnert PM, Munaron J-M, Lebreton B, Otake T, Letourneur Y (2017) Trophic position increases with thermocline depth in yellowfin and bigeye tuna across the Western and Central Pacific Ocean. *Prog Oceanogr* 154:49–63.
- Ledesma M, Gorokhova E, Holmstrand H, Garbaras A, Karlson AML (2020) Nitrogen isotope composition of amino acids reveals trophic partitioning in two sympatric amphipods. *Ecol Evol* 10:10773–10784.
- Lemons GE, Lewison RL, Seminoff JA, Coppentrath CM, Popp BN (2020) Nitrogen isotope fractionation of amino acids from a controlled study on the green turtle (*Chelonia mydas*): expanding beyond Glx/Phe for trophic position. *Mar Biol* 167:149.
- McCarthy MD, Benner R, Lee C, Fogel ML (2007) Amino acid nitrogen isotopic fractionation patterns as indicators of heterotrophy in plankton, particulate, and dissolved organic matter. *Geochim Cosmochim Acta* 71:4727–4744.
- McCarthy MD, Lehman J, Kudela R (2013) Compound-specific amino acid $\delta^{15}\text{N}$ patterns in marine algae: Tracer potential for cyanobacterial vs. eukaryotic organic nitrogen sources in the ocean. *Geochim Cosmochim Acta* 103:104–120.
- McClelland JW, Holl CM, Montoya JP (2003) Relating low $\delta^{15}\text{N}$ values of zooplankton to N_2 -fixation in the tropical North Atlantic: insights provided by stable isotope ratios of amino acids. *Deep Sea Res Part Oceanogr Res Pap* 50:849–861.
- McClelland JW, Montoya JP (2002) Trophic relationships and the nitrogen isotopic composition of amino acids in plankton. *Ecology* 83:2173–2180.
- Peavey LE, Popp BN, Pitman RL, Gaines SD, Arthur KE, Kelez S, Seminoff JA (2017) Opportunism on the High Seas: Foraging Ecology of Olive Ridley Turtles in the Eastern Pacific Ocean. *Front Mar Sci* 4:348.
- Richards TM, Sutton TT, Wells RJD (2020) Trophic Structure and Sources of Variation Influencing the Stable Isotope Signatures of Meso- and Bathypelagic Micronekton Fishes. *Front Mar Sci* 7:507992.
- Sackett DK, Drazen JC, Choy CA, Popp B, Pitz GL (2015) Mercury Sources and Trophic Ecology for Hawaiian Bottomfish. *Environ Sci Technol* 49:6909–6918.
- Sahm R, Sünger E, Burmann L, Zubrod JP, Schulz R, Fink P (2020) Compound-specific $\delta^{15}\text{N}$ analyses of amino acids for trophic level estimation from indigenous and invasive freshwater amphipods. 7.
- Sherwood OA, Lehmann MF, Schubert CJ, Scott DB, McCarthy MD (2011) Nutrient regime shift in the western North Atlantic indicated by compound-specific $\delta^{15}\text{N}$ of deep-sea gorgonian corals. *Proc Natl Acad Sci* 108:1011–1015.
- Tokuda AK, Drazen JC, Geringer ME, Popp BN, Grammatopoulou E, Mayor DJ (2020) Trophic interactions of megafauna in the Mariana and Kermadec trenches inferred from stable isotope analysis. *Deep Sea Res Part Oceanogr Res Pap* 164:103360.

- Vander Zanden HB, Arthur KE, Bolten A b., Popp B, Lagueux C, Harrison E, Campbell C, Bjorndal K (2013) Trophic ecology of a green turtle breeding population. *Mar Ecol Prog Ser* 476:237–249.
- Vane K, Wallsgrove N, Ekau W, Popp B (2018) Reconstructing lifetime nitrogen baselines and trophic position of *Cynoscion acoupa* from $\delta^{15}\text{N}$ values of amino acids in otoliths. *Mar Ecol Prog Ser* 597:1–11.
- Williams B, Thibodeau B, Chikaraishi Y, Ohkouchi N, Walnum A, Grottoli AG, Colin PL (2017) Consistency in coral skeletal amino acid composition offshore of Palau in the western Pacific warm pool indicates no impact of decadal variability in nitricline depth on primary productivity: Western Pacific variability from coral composition. *Limnol Oceanogr* 62:399–407.

Appendix D

The authors thank both reviewers for further consideration and the detailed responses with which we have aimed to improve the manuscript sufficiently to allow for publication.

Reviewer 1

I believe the authors adequately addressed most of the reviewers' original concerns with the manuscript. The additional ecological context and detailed project objectives in the introduction are helpful, as are the greater clarification of error propagation and analysis of the Ala, Ser, Glu, and Asp data. However, the authors' justifications for their β and TDF values are fundamentally flawed and call into question the validity of the TP estimates from this study. My issue is not with the TP, β , nor TDF estimates themselves, which by and large seem reasonable for the given species and systems, but with the path that led to them. Recalculated β and TDFs may very well end up close to the 3.4‰ and 4.0‰ values used by the authors, but how they are derived matters and the authors' current approach does not match how these are derived in the CSIA literature.

The authors use Chikaraishi et al. 2009 and Matthews et al. 2020 as justification for not recalculating β and TDFs when averaging AA $\delta^{15}\text{N}$ values within the TP equation, asserting "*The necessity of recalculating β indicated by reviewer 1 is not reflected in the current literature: Chikaraishi et al. 2009 presents a calculation that includes the use of weighted averages from trophic and source AAs along with a TDF of 7.6‰ and the typical β of 3.4*". This is simply not true. Chikaraishi et al. (2009) did not present equations to calculate TPs that use weighted averages of AAs. They presented alternate TPCSIA equations for individual combinations of trophic AAs and Phe, not weighted averages, and in fact advance the need to use different β and TDF values for each combination of trophic and source AA (Equations 3-7). So, it is unclear to me why this source is used as justification for not using weighted β and TDF values. It is similarly unclear why Matthews et al. (2020) cited Chikaraishi et al. (2007, 2009) at all for their weighted mean approach, as neither of these sources discuss the issue of weighted means. Matthews et al. (2020) either forgot to include the clarifier "modified from" or cited the wrong literature. It was McCarthy et al. (2007) that first proposed using weighted means of trophic and source AAs in TPCSIA equations, but it was not until Vander Zanden et al. (2013) that this framework was extended to the canonical TPCSIA equation presented by Chikaraishi et al. (2009). Moreover, Matthews et al. (2020) tested four TPCSIA equations and found that the multi-TDF equations presented by Germain et al. (2013) or McMahon et al. (2015) produced bowhead whale—the only mysticete sampled—TPCSIA estimates (2.4–3.0) closest to TP calculated from stomach contents (3.2). In fact, the weighted averages equation the authors use in the present study produced the most inaccurate TPCSIA estimates (1.9–2.0). There is thus no scientific justification for using weighted averages of trophic and sources AA $\delta^{15}\text{N}$ in combination with Glu-Phe β and TDF values within Chikaraishi et al. 2009 and Matthews et al. 2020.

Moreover, a review of the CSIA literature makes clear that when using weighted averages of trophic and sources AA $\delta^{15}\text{N}$ the convention within the field is to recalculate β and TDF values as was suggested by both reviewers (see the following 18 studies Vander Zanden et al. 2013, Downs et al. 2014, Choy et al. 2015, Bradley et al. 2015, Gerringer et al. 2017, Williams et al. 2017, Houssard et al. 2017, Décima et al. 2017, Peavey et al. 2017, Gagne et al. 2018a b, Vane et al. 2018, Sahm et al. 2020, Glass et al. 2020, Ledesma et al. 2020, Lemons et al. 2020, Tokuda et al. 2020, Richards et al. 2020). The only cases where this is not the norm is when β and TDF are removed from the equation entirely and a relative TP is calculated using weighed means of trophic and source AAs only (e.g., McCarthy et al. 2007, Sherwood et al. 2011, Batista et al. 2014, Sackett et al. 2015, Bradley et al. 2016). As a result, this manuscripts combination of weighted consumer AA values and Glu-Phe β values and TDFs is unprecedented in the CSIA literature. At a fundamental level, if the authors believe that the weighted means approach improves TPCSIA estimation then it is illogical not to extend that approach to all parameters in the TPCSIA equation. Otherwise, one is not comparing like to like and not accurately propagating error.

The authors make a valid point that primary producer $\delta^{15}\text{N}$ Lysine data are underrepresented in the CSIA literature. However, McClelland & Montoya (2002), McClelland et al. (2003), McCarthy et al. (2007, 2013), and Gutiérrez-Rodríguez et al. (2014) collectively provide sufficient data from which all necessary β values for this manuscript could be calculated. Indeed, Table 4 in Bradley et al. (2015) distills much of this information into such β values (including XX-Lys). These data are hardly speculative and have been used in a number of subsequent TPCSIA studies (included in list above).

Lastly, the sliding scale approach for estimating a TDF of 4.0‰ is similarly unprecedented in the CSIA literature. I am aware of no CSIA study that has used this approach. The convention in this scenario would be to use a TDF from another marine mammal or urea producing secondary consumer. The lack of species-specific AA $\delta^{15}\text{N}$ TDFs is a common issue for all marine megafauna. Again, I believe 4.0‰ is likely a reasonable TDF for this manuscript, but the path to that value matters. At minimum, the authors need to further detail their sliding scale approach as I still do not understand how they arrived at 4.0‰. However, issues outlined above with only using Glu-Phe aside, using a scaling factor of 7.6‰ here for AAs is problematic. This value is indeed considered a ‘canonical’ TDF, but a rich body of literature clearly demonstrates that in marine systems it is only accurate for primary and secondary ammonia-producing consumers (e.g., invertebrate, zooplankton, planktivorous fish). As reviewed by Nielsen et al. (2015), Bradley et al. (2015), and McMahan and McCarthy (2016), higher order consumers (secondary+), particularly urea-producing consumers, typically have much lower TDFs, which the authors also note in this manuscript. As a result, if the authors are keen on using a sliding scale approach, they should instead use a lower scaling factor for AAs. A value of 7.6‰ is especially inappropriate to use for humpback and minke whales given that they feed on fish.

As I see it the authors have 3 options for estimating TPCSIA estimates moving forward that are supported by the prevailing CSIA literature:

1. Use the published literature to re-calculate weighted β and TDF values as originally suggested by the two reviewers.
2. Abandon the weighted means approach and use the Germain et al. (2013) or McMahon et al. (2015) multi-TDF equations in combination with a single trophic and single source AA, the latter of which is the most commonly applied in the literature.
3. Calculate relative TPCSIA estimates using weighted averages of trophic and source AAs only as outlined by McCarthy et al. (2007) (and others).

I do not believe the results of this study will fundamentally change as a result of these recalculations, but they will yield more defensible TPCSIA estimates. This is critically important given the “proof of concept” nature of this study and the fact that it presents some of the first TPCSIA estimates for mysticetes, which will inevitably be cited in many future studies.

We agree with the reviewer’s outlook and now provide trophic level calculations using appropriately transformed β values and TDF estimates appropriately scaled to the trophic positions found by Pauly et al. 1998 and the TDF estimated from Ruiz-Cooley et al. 2021 using a variety of statistical analyses from a large data set of dolphins. We also present TL calculated using weighted averages of trophic and source amino acids using a recalculated β using the references provided by Reviewer 1. We have edited several figures and recalculated the statistical tests throughout to account for recalculation due to changed trophic level values. We now provide a table of the trophic levels and the values for TDF and β used to calculate them in Table 2, reproduced below. These revised calculations do not substantially alter the conclusions of this work.

Table 2: Trophic position and trophic level (TL) estimates for each individual. Trophic positions were determined from stomach contents and dietary analysis in Pauly et al. 1998. TL is a unitless number calculated here using glutamic acid (Glu), phenylalanine (Phe), or the weighted average of trophic and source amino acids with β and trophic discrimination factors indicated below each estimate. n represents the number of amino acid measurements along the length of baleen for each individual and SD indicates the standard deviation propagated for each value.

Individual	n	TP _{SCA} ^a	TL _{Glu-Phe} ^b	SD	TL _{Glu-Phe} ^b	SD	TL _{Trophic-Source}	SD
Fin Whale 1	10	3.4	3.2	0.2	3.6	0.2	3.0	0.3
Fin Whale 2	11	3.4	3.0	0.2	3.3	0.2	3.0	0.3
Fin Whale 3	16	3.4	3.3	0.2	3.7	0.2	3.2	0.4
Humpback	12	3.6	3.6	0.2	4.0	0.2	3.7	0.5
Minke	11	3.4	3.7	0.4	4.1	0.5	3.8	0.4
TDF			3.6*	0.3	3.1	0.3	3.6	1.7
β			3.6	0.5	3.6	0.5	3.0	0.9
^a Pauly et al. 1998								
^b Ruiz-Cooley et al. 2021								
*eq. 3 average for TDF								

Specific comments

Line 4: add “values” after δ^{15}

Corrected as indicated

Line 78: clarify $7.6 \pm 1.5\%$ is an exampleTDF (e.g., $7.6 \pm 1.5\%$). Remove extra space after the last parenthesis.

Corrected as indicated.

Line 93: Remove “Additionally,”

Corrected as indicated.

Line 309: I recommend removing the phrase “compared to the typically applied value of 7.6% .” As written, this sentence could be taken to mean that 7.6% is typical for whales, which is certainly not the case.

We now clarify further that the TDF of 7.6% is not typical of whales.

LN 309 now reads: “Trophic level estimates for all of the whales (ranging from 2.7 to 3.5) were based on a relatively small TDF (3.6%) compared to the typically applied value of 7.6% used for estimation of lower trophic levels (e.g. fish and invertebrates) (39, 42).”

References

- Batista FC, Ravelo AC, Crusius J, Casso MA, McCarthy MD (2014) Compound specific amino acid δ^{15} N in marine sediments: A new approach for studies of the marine nitrogen cycle. *Geochim Cosmochim Acta* 142:553–569.
- Bradley C, Longenecker K, Pyle R, Popp B (2016) Compound-specific isotopic analysis of amino acids reveals dietary changes in mesophotic coral-reef fish. *Mar Ecol Prog Ser* 558:65–79.
- Bradley CJ, Wallsgrove NJ, Choy CA, Drazen JC, Hetherington ED, Hoen DK, Popp BN (2015) Trophic position estimates of marine teleosts using amino acid compound specific isotopic

- analysis: Stable isotope-derived trophic positions of teleosts. *Limnol Oceanogr Methods* 13:476–493.
- Choy CA, Popp BN, Hannides CCS, Drazen JC (2015) Trophic structure and food resources of epipelagic and mesopelagic fishes in the North Pacific Subtropical Gyre ecosystem inferred from nitrogen isotopic compositions: Trophic structure of pelagic fishes. *Limnol Oceanogr* 60:1156–1171.
- Décima M, Landry MR, Bradley CJ, Fogel ML (2017) Alanine $\delta^{15}\text{N}$ trophic fractionation in heterotrophic protists. *Limnol Oceanogr* 62:2308–2322.
- Downs E, Popp B, Holl C (2014) Nitrogen isotope fractionation and amino acid turnover rates in the Pacific white shrimp *Litopenaeus vannamei*. *Mar Ecol Prog Ser* 516:239–250.
- Gagne TO, Hyrenbach KD, Hagemann ME, Van Houtan KS (2018a) Trophic signatures of seabirds suggest shifts in oceanic ecosystems. *Sci Adv* 4:eao3946.
- Gerringer ME, Popp BN, Linley TD, Jamieson AJ, Drazen JC (2017) Comparative feeding ecology of abyssal and hadal fishes through stomach content and amino acid isotope analysis. *Deep Sea Res Part Oceanogr Res Pap* 121:110–120.
- Glass J, Daly R, Cowley P, Post D (2020) Spatial trophic variability of a coastal apex predator, the giant trevally *Caranx ignobilis*, in the western Indian Ocean. *Mar Ecol Prog Ser* 641:195–208.
- Gutiérrez-Rodríguez A, Décima M, Popp BN, Landry MR (2014) Isotopic invisibility of protozoan trophic steps in marine food webs. *Limnol Oceanogr* 59:1590–1598.
- Houssard P, Lorrain A, Tremblay-Boyer L, Allain V, Graham BS, Menkes CE, Pethybridge H, Couturier LIE, Point D, Leroy B, Receveur A, Hunt BPV, Vourey E, Bonnet S, Rodier M, Raimbault P, Feunteun E, Kuhnert PM, Munaron J-M, Lebreton B, Otake T, Letourneur Y (2017) Trophic position increases with thermocline depth in yellowfin and bigeye tuna across the Western and Central Pacific Ocean. *Prog Oceanogr* 154:49–63.
- Ledesma M, Gorokhova E, Holmstrand H, Garbaras A, Karlson AML (2020) Nitrogen isotope composition of amino acids reveals trophic partitioning in two sympatric amphipods. *Ecol Evol* 10:10773–10784.
- Lemons GE, Lewison RL, Seminoff JA, Copenrath CM, Popp BN (2020) Nitrogen isotope fractionation of amino acids from a controlled study on the green turtle (*Chelonia mydas*): expanding beyond Glx/Phe for trophic position. *Mar Biol* 167:149.
- McCarthy MD, Benner R, Lee C, Fogel ML (2007) Amino acid nitrogen isotopic fractionation patterns as indicators of heterotrophy in plankton, particulate, and dissolved organic matter. *Geochim Cosmochim Acta* 71:4727–4744.
- McCarthy MD, Lehman J, Kudela R (2013) Compound-specific amino acid $\delta^{15}\text{N}$ patterns in marine algae: Tracer potential for cyanobacterial vs. eukaryotic organic nitrogen sources in the ocean. *Geochim Cosmochim Acta* 103:104–120.

- McClelland JW, Holl CM, Montoya JP (2003) Relating low $\delta^{15}\text{N}$ values of zooplankton to N_2 -fixation in the tropical North Atlantic: insights provided by stable isotope ratios of amino acids. *Deep Sea Res Part Oceanogr Res Pap* 50:849–861.
- McClelland JW, Montoya JP (2002) Trophic relationships and the nitrogen isotopic composition of amino acids in plankton. *Ecology* 83:2173–2180.
- Peavey LE, Popp BN, Pitman RL, Gaines SD, Arthur KE, Kelez S, Seminoff JA (2017) Opportunism on the High Seas: Foraging Ecology of Olive Ridley Turtles in the Eastern Pacific Ocean. *Front Mar Sci* 4:348.
- Richards TM, Sutton TT, Wells RJD (2020) Trophic Structure and Sources of Variation Influencing the Stable Isotope Signatures of Meso- and Bathypelagic Micronekton Fishes. *Front Mar Sci* 7:507992.
- Sackett DK, Drazen JC, Choy CA, Popp B, Pitz GL (2015) Mercury Sources and Trophic Ecology for Hawaiian Bottomfish. *Environ Sci Technol* 49:6909–6918.
- Sahm R, Sünger E, Burmann L, Zubrod JP, Schulz R, Fink P (2020) Compound-specific $\delta^{15}\text{N}$ analyses of amino acids for trophic level estimation from indigenous and invasive freshwater amphipods. 7.
- Sherwood OA, Lehmann MF, Schubert CJ, Scott DB, McCarthy MD (2011) Nutrient regime shift in the western North Atlantic indicated by compound-specific $\delta^{15}\text{N}$ of deep-sea gorgonian corals. *Proc Natl Acad Sci* 108:1011–1015.
- Tokuda AK, Drazen JC, Gerringer ME, Popp BN, Grammatopoulou E, Mayor DJ (2020) Trophic interactions of megafauna in the Mariana and Kermadec trenches inferred from stable isotope analysis. *Deep Sea Res Part Oceanogr Res Pap* 164:103360.
- Vander Zanden HB, Arthur KE, Bolten A b., Popp B, Lagueux C, Harrison E, Campbell C, Bjorndal K (2013) Trophic ecology of a green turtle breeding population. *Mar Ecol Prog Ser* 476:237–249.
- Vane K, Wallsgrove N, Ekau W, Popp B (2018) Reconstructing lifetime nitrogen baselines and trophic position of *Cynoscion acoupa* from $\delta^{15}\text{N}$ values of amino acids in otoliths. *Mar Ecol Prog Ser* 597:1–11.
- Williams B, Thibodeau B, Chikaraishi Y, Ohkouchi N, Walnum A, Grottoli AG, Colin PL (2017) Consistency in coral skeletal amino acid composition offshore of Palau in the western Pacific warm pool indicates no impact of decadal variability in nitricline depth on primary productivity: Western Pacific variability from coral composition. *Limnol Oceanogr* 62:399–407.

Reviewer 2

Comments to the Author(s)

This is the second round of comments I'm providing on Riekenberg et al. so I will focus my current comments on the response file furnished with the resubmitted manuscript. I've organized this review on the numbered responses to my original comments. Overall, I cannot recommend this manuscript for publication in ROS in its current form because the trophic position (TP) estimates based on amino acid (AA) $\delta^{15}\text{N}$ values are flawed and require further consideration. The authors method for calculating TP produces estimates that are <3 (e.g., 2.7 for fin whales?), which suggests that these species are omnivores assuming phytoplankton has a TP of 1 (Equation 2). How can a baleen whale be an omnivore, especially a fin or humpback whale that are known to consume a combination of krill and schooling fish (e.g., herring, capelin)? Decades of dietary data based on stomach content analysis and observation shows that baleen whales (especially these species) are not omnivores.

Please see commentary from the response to Reviewer 1 and Reviewer 2's comment #2 and #8 that thoroughly cover the actions that we have taken to address this concern.

Lastly, the manuscript could use some grammatical editing (e.g., Lines 28-29), which was also suggested in my first round of comments. The manuscript should be carefully read and edited before resubmission to ROS or another journal.

Corrected as indicated.

LN 28-29 now reads: "However, prey in the stomach of a deceased animal represents the animal's last meal and can be biased (13), reflecting 1) only the most recent feedings, 2) the health status of the animal prior to death (i.e. trophic downgrading due to sickness), and 3) the undigestible portions of the prey.

We have carefully checked the manuscript and made small edits where needed.

Comment #2. Despite what the literature says (Chikaraishi et al. 2009, 2014), using beta and source-trophic TDF for Glu-Phe in an estimate of trophic position (TP) based on a mixture of trophic and source AA values is fundamentally flawed as each trophic-source pair has their own TDF and beta. For example, TDFs and beta values for Ala-Lys are very different than for Glu-Phe, so how can one justify only using betas for the latter pair when estimating TP. A lot has been done on this topic in the past ~10 years since these pioneering papers were published, and while the authors complicated method for estimating TP produces a TDF of ~4 per mil, this estimate is based on a starting value of 7.6 per mil, which might have been the canonical value a decade ago but has not stood up to further analysis and scrutiny (e.g., McMahon and McCarthy 2016), especially for the group of animals (marine mammals) under study in this manuscript (Matthews et al. 2020).

To address this, we now appropriately pair our TDFs and β values for the amino acid pairings being examined, as presented in Table 2, see response to reviewer 1). We now present trophic level estimates using both single and weighted averages for trophic and source amino acids and have aimed to incorporate the most up to date values for marine mammals from the statistical analysis presented in Ruiz-Cooley et al 2021. They recalculated the β value used for the weighted average to appropriately incorporate all of the pairing between multiple trophic AAs and both source AAs lysine and phenylalanine. All TDF and β values are presented at the bottom of Table 2.

Comment #6. I appreciate the additional information regarding the AA isotope analysis and I've thoroughly read the authors methods papers in RCMS, which is very similar to the protocol we use in our lab to make these measurements. One additional detail I would like to see in the supplemental material is a table reporting the average with-run analytical precision (e.g., SD) for each AA accompanied by a short description of any patterns in these data; e.g., variation in precision likely due to coelution or other analytical challenges. This should be easy to compile for the runs used to generate the data presented in this manuscript.

We now provide the requested information in Supplemental table 3, reproduced here.

Supplemental Table 3: Average standard deviations for the determination of the isotopic composition of the amino acids analyzed separated into individual analysis days. Larger than normal standard deviations (>0.5%) for Thr in the initial runs are due to the combination of establishing a proper loading concentration for baleen material and getting to the end of a combustion reactor's lifetime.

Date Run	Ala	Asp	Glu	Gly	Ile	Leu	Lys	Phe	Ser	Thr	Tyr	Val
20181203	0.29	0.41	0.14	0.34	0.24	0.11	0.34	0.18	0.48	0.80	0.29	0.46
20181204	0.36	0.30	0.24	0.24	0.36	0.17	0.30	0.25	0.14	0.29	0.47	0.30
20181211	0.36	0.29	0.34	0.51	0.41	0.14	0.26	0.32	0.41	0.41	0.35	0.41
20181212	0.19	0.25	0.21	0.54	0.37	0.15	0.20	0.29	0.28	1.41	0.18	0.22
20191017	0.11	0.32	0.13	0.17	0.20	0.15	0.13	0.13	0.27	0.17	0.22	0.26
20191021	0.14	0.18	0.07	0.16	0.20	0.16	0.20	0.27	0.18	0.28	0.26	0.18
20191022	0.21	0.24	0.31	0.23	0.29	0.26	0.28	0.12	0.07	0.25	0.25	0.23
20191023	0.10	0.11	0.19	0.13	0.17	0.15	0.10	0.20	0.09	0.18	0.26	0.25
20191024	0.30	0.21	0.07	0.21	0.19	0.21	0.25	0.30	0.18	0.21	0.34	0.26
20191025	0.14	0.19	0.25	0.16	0.20	0.14	0.32	0.17	0.27	0.28	0.26	0.22
20191028	0.17	0.24	0.22	0.23	0.13	0.35	0.24	0.33	0.21	0.31	0.25	0.15
20191029	0.22	0.18	0.17	0.14	0.18	0.17	0.25	0.26	0.28	0.20	0.30	0.15
20191030	0.10	0.40	0.13	0.32	0.09	0.28	0.23	0.31	0.16	0.31	0.44	0.25

Comment #8. See concerns above regarding the use of Glu-Phe beta values in Equation 2 that relies on a mixture of 8 trophic and source AAs. I agree that nowhere in the manuscript do the authors mention using a TDF for fish and invertebrates, but that is exactly what the canonical 7.6 per mil TDF for Glu-Phe represents, and while scaling the Glu-Phe TDF in the way the authors have done here tries to adjust this value to be more representative of a mammal (with a different N excretion pathway than a fish/invert), Equation #2 and the assumptions therein still produce TP estimates that are on the cusp of being ecologically possible for these species, which consume a mixture of (large) zooplankton and fish. As such, these species should have TP estimates that are above 3.

These concerns have been addressed through the application of TDF values of 3.6 and 3.1 following the methods presented in Ruiz-Cooley et al. 2021, all of which are presented in Table 2, reproduced in response to Reviewer 1's comments. The TDF of 3.6 scales to the average trophic position for these species found by Pauly in Pauly et al, 1998. We believe that the resulting trophic levels from 3.0 to 4.1 likely align more with the reviewer's expectation that these animals are not omnivores, but we would like to point out that Pauly's original trophic position estimates ranging from 3.4 to 3.6 are not terrifically far outside of our original estimates considering the rather large errors propagated from the multi-AA approach.

Comment #12. Stating that the TP estimate for fin whales is better than that produced by Matthews et al. (2020) of 2.0 for bowhead whales is not reassuring. Matthews et al. reported that TP estimate to prove a point, which is that TDF for large whales is much lower than the 7.6 per mil value.

Furthermore, stating that the estimate for fin whales (2.7) is close to that of bowhead whales (3.2) is simply not true, it's half a trophic level different and ecologically meaningful. If its "close", then how confident can we be that AA-derived estimates of trophic position are robust and useful?

This statement was provided for context to other TL estimates that are present in the literature. We now also refer to Cooley-Ruiz et al 2021 that found a TDF of 3.1‰ to be useful for providing more realistic trophic levels in dolphins likely due to increased similarity between the protein content of the food they eat and themselves resulting in reduced fractionation. The reviewer presents this material as fully settled, yet there are still relevant papers coming out that thoroughly investigate the compression of TDFs within marine mammals, even during this review process. We have done our best to take the new TDFs into consideration and to integrate them into our response as well as revising our multi-AA calculation as requested.

LN 309 to 318 now reads: "Trophic level estimates for all of the whales (ranging from 2.7 to 3.5) were based on a relatively small TDF (3.6‰) compared to the typically applied value of 7.6‰ used

for estimation of lower trophic levels (e.g. fish and invertebrates) (39, 42). A TDF of 4‰ yielded comparable trophic levels for zooplanktivorous whales (Bowhead whales, *Balaena mysticetes* TL of 1.9-3.0) as determined through amino acid analysis and stomach contents (TL 3) in Matthews, Ruiz-Cooley (63). Smaller TDFs reflect increased similarity between the protein quality of the diet and the consumer, resulting in less reworking and therefore less fractionation of amino acid N during metabolism (39, 64) and application in marine mammals has been found to be appropriate (51). In future work, it may be useful to further account for the protein quality differences between prey types (e.g. krill, fish) using scaled TDF equations, but this is outside the scope of this study.”

Comment 14. Do the authors have an explanation for the apparent decoupling of nitrogen in Gly versus Ser? Is it possible that these two AAs are derived from different pools (e.g., endogenous versus exogenous), or that one might be synthesized de novo and the other is routed directly from protein in diet? More physiology and a better explanation for the unexpected dissimilarity in these two closely related AAs is needed.

Unfortunately, we do not have an explanation outside of pointing to McMahon and McCarthy 2016 and Nielsen et al 2015, both of which state that Gly and Ser appear to interact heavily with the central Glu pool during the production of urea and that Gly may be heavily affected by microbial reworking. That Gly was significantly different between minimum and maximum periods and Ser was not may depend on the relative intensity of the fasting that occurred, both Lubcker et al. 2020 and Fuller et al. 2017 found impacts to both AAs from extensive starvation and protein deficiency, respectively. Further extensive discussion of physiology on two less well characterized amino acids is outside of the scope of this study due to 1) limited sample size and 2) the primary focus being on the novelty of using baleen as an across lifetime indicator in whales.

Comment 16. Does POM from the mid-Atlantic really have lower $\delta^{13}\text{C}$ values than POM from the Arctic? But more importantly, are the authors confusing isotopic turnover (some times called incorporation) with assimilation? Neither the Cherry et al. 2011 or Newsome et al. 2014 study examined isotopic turnover in AA so its not clear to me how the authors are relating these two studies to their statement regarding differential turnover of nitrogen and carbon? I think the authors are talking about macromolecular routing here and potential de novo synthesis of proteinaceous tissues from lipid carbon, which was the topic of both papers listed above in wild polar bears and lab-raised house mice.

The existing isoscape for particulate matter indicates opposite of the pattern the reviewer has indicated. POM values for $\delta^{13}\text{C}$ are ~ -22 to -24 ‰ for the mid-Atlantic and ~ -28 ‰ for the Arctic. Figure comes from Trueman et al. 2019.

Further discussion of this comment is provided in the response to Comment 18.

Comment 18. Yes, but the bulk (tissue) values are a mixture of essential and non-essential AAs given that the samples being measured are largely proteinaceous in character. As such, that statement that the higher variation observed in $d15\text{N}$ is likely driven by increased turnover in N in comparison to C is not supported by the literature, nor do the authors present a feasible mechanism for this statement. The metabolic pool of N in a whale is quite large, especially if you consider the major onboard reservoir of muscle, so while I understand that the endogenous lipid pool is larger (with lots of C, but little N), I don't understand the mechanisms by which increased isotopic variation is driven by higher isotopic turnover?

The traits in the data that we are trying to explain are the increased variation in values for N vs C and how this is caused in marine mammals more so than for other animals with smaller fat stores. Amino acids contain $\sim 9\times$ more C than N and are the bulk of the material contained in baleen. The carbon forming those amino acids comes from the blood stream and reflects the C being metabolized at the time of deposit. Depending on the animal's metabolic state, this C is likely to predominately reflect exogenous sources during ample feeding, or endogenous (which is an interesting concept here, because endogenous lipids are predominately exogenous material now stored through metabolic routing and still retaining the original signal of the material that it was formed from, slightly reworked). But when these marine mammals move off their feeding grounds and begin to feed less, the C in their blood begins to reflect increased utilization of the lipid stores (previous exogenous material with $\delta^{13}\text{C}$ value reflecting the feeding ground where the material was taken on board). This subsidy from lipid stores is likely to muddle the changes observed in $d13\text{C}$ versus $d15\text{N}$, as no such comparable pool of stored metabolic N exists, leading to less change in the variation of C (hysteresis) due to a continuous contribution of previously metabolically routed lipids with a $d13\text{C}$ values closer to their feeding grounds than their current position. This concept has little to do with turnover times

per se, and more to do with the relative sizes and characteristic of the pools of C and N and physiological traits that are somewhat unique to marine mammals.

We have further clarified these points on LN 385: “The larger ranges observed for $\delta^{15}\text{N}_{\text{Phe}}$ and $\delta^{15}\text{N}_{\text{Base}}$ (8 and 7‰, respectively) versus $\delta^{13}\text{C}_{\text{Base}}$ (2.3‰) are likely due to the large amounts of lipid stores that are primarily developed with time spent on feeding grounds (62). Under fasting conditions, lipid stores will be utilized as a source of C with a light $\delta^{13}\text{C}$ value that reflects the fractionation of C from food resources containing the regional $\delta^{15}\text{N}$ values where they were consumed (76). These lipid stores are expected to be mobilized during times of limited feeding and reduce the impact of C derived from incidental feeding on $\delta^{13}\text{C}_{\text{Base}}$ values (hysteresis) of the baleen during fasting conditions as metabolites from blood are incorporated into baleen. Use of C from lipid stores contribute to the dampening of variation in C values along the baleen, and although the metabolic N pool in whales is quite large, there is no comparable storage pool for N. Therefore, N from incidental feeding is expected to be more directly metabolized into animal tissues, while carbon from lipid rich prey can be metabolically routed to either direct incorporation to tissue or to storage within large lipid stores depending on feeding status (77).”